# TEST TIME TRAINING ENHANCES IN-CONTEXT LEARNING OF NONLINEAR FUNCTIONS

## ABSTRACT

Test-time training (TTT) enhances model performance by explicitly updating designated parameters prior to each prediction to adapt to the test data. While TTT has demonstrated considerable empirical success, its theoretical underpinnings remain limited, particularly for nonlinear models. In this paper, we investigate the combination of TTT with in-context learning (ICL), where the model is given a few examples from the target distribution at inference time. We analyze this framework in the setting of single-index models $y = \sigma_*(\langle \beta, \boldsymbol{x} \rangle)$, where the feature vector $\beta$ is drawn from a hidden low-dimensional subspace. For single-layer transformers trained with gradient-based algorithms and adopting TTT, we establish an upper bound on the prediction risk. Our theory reveals that TTT enables the single-layer transformers to adapt to both the feature vector $\beta$ and the link function $\sigma_*$, which vary across tasks. This creates a sharp contrast with ICL alone, which is theoretically difficult to adapt to shifts in the link function. Moreover, we provide the convergence rate with respect to the data length, showing the predictive error can be driven arbitrarily close to the noise level as the context size and the network width grow.

## 1 INTRODUCTION

In-context learning (ICL) is a powerful capability of pretrained transformers to solve tasks using a few labeled examples provided as input, without updating their weights. This ability has gained increasing attention with the advent of models with massive context windows, as more examples lead to significantly improved performance (Agarwal et al., 2024). This approach has also proven effective for multimodal tasks (Jiang et al., 2024). From a theoretical perspective, transformers are known to implement algorithms such as linear regression. Recent studies have extended this understanding to nonlinear settings, showing that transformers can learn nonlinear single-index models (Oko et al., 2024) and that softmax attention facilitates data-efficient feature learning (Nishikawa et al., 2025). Nevertheless, ICL faces its inherent limitations: the performance of ICL is fundamentally constrained by factors like the pretraining data (Bigoulaeva et al., 2025) and model architecture (Naim & Asher, 2025).

Test-time training (TTT) has emerged as a promising strategy to overcome these barriers. TTT adapts the model by updating its parameters on the test data before each prediction. This adaptive mechanism has led to strong empirical success across various fields, including large language models (Hu et al., 2025) and video object segmentation (Bertrand et al., 2023). In the context of ICL, TTT can be seamlessly integrated by using the in-context examples as data for task-specific adaptation. For instance, Akyürek et al. (2025) demonstrated that this combination achieves notable improvements on few-shot reasoning benchmarks.

Despite its empirical success, the theoretical foundations of TTT remain underdeveloped. A notable work by Gozeten et al. (2025) established the statistical efficiency of TTT over standard ICL, but their analysis was restricted to linear regression with a linear transformer. This simplified model fails to capture the true potential of TTT's power on nonlinear, complex tasks. This gap motivates our primary research question:

*Does test-time training improve in-context learning in nonlinear settings?*

Furthermore, existing theoretical works commonly analyze performance in high-dimensional regimes, showing that the prediction loss is $o_d(1)$ and thus vanishes as the data dimension $d$ grows. In practice, however, this dimension is fixed, making it crucial to understand how the loss behaves as the number of data points $n$ increases. This raises our second key question:

*How does the test loss behave when we fix the dimension and increase the data size?*

## 1.1 OUR CONTRIBUTION

To address the questions above, we analyze the performance of transformers on learning single-index models, a simple type of nonlinear function. In our setting, each task is defined as

$$\boldsymbol{x}_1^t, \ldots, \boldsymbol{x}_N^t, \boldsymbol{x}^t \overset{\text{i.i.d.}}{\sim} \mathcal{N}(0, \boldsymbol{I}_d), \ y_i^t \approx \sigma_*^t(\langle \beta^t, \boldsymbol{x}_i^t \rangle) \ (i = 1, \cdots, n)$$

where $\sigma_*^t$ is an unknown polynomial that varies across tasks, and $\beta^t$ is drawn from a fixed $r$-dimensional subspace. Using the in-context data $\boldsymbol{x}_1, \cdots, \boldsymbol{x}_N$, our model constructs the predictor for the new query $\boldsymbol{x}$. We establish a rigorous upper-bound on the predictive risk for a transformer that utilizes TTT. Our main result is as follows:

**Theorem 1** (Informal). *Consider learning single-index polynomial $\sigma_*^t(\langle \beta^t, \boldsymbol{x} \rangle)$ with the transformer trained via Algorithm 1. Then, with probability at least 0.99, we can construct the model $f_{\text{TF}}$ for each prompt that satisfies $\mathbb{E}[\|f_{\text{TF}}(\boldsymbol{x}) - y\|] = \tau + \tilde{O}(m^{-1/2}) + \tilde{O}(\sqrt{\frac{r\sqrt{r}}{N_{test}}})$, where $N_{pt}$ and $T_{pt}$ are the context length and the number of tasks in pretraining, respectively, $N_{test}$ is the context length in test-time, $m$ is the network width and $\tau = O(1)$ is the noise level, if $N_{pt} = T_{pt} = (d^{\Omega(\text{ie}(\sigma_*^t))} r^2)$ and $N_{test} = \tilde{\Omega}(r^{\Omega(\text{ge}(\sigma_*^{test}))})$, where $\text{ie}(\sigma_*)$ and $\text{ge}(\sigma_*)$ are the information exponent and the general exponent of the polynomial $\sigma_*$, respectively.*

This theorem ensures the effectiveness of TTT in learning nonlinear single-index models, extending its known applicability to linear models. Our analysis reveals several strengths of our approach:

- Efficient sample complexity: Theorem 1 implies that $N_{test} = \tilde{\Omega}(r^{2 \vee \Theta(\text{ge}(\sigma_*^{test}))})$ to ensure low predictive loss $o(1)$, which does not depend on the entire dimension $d$. This shows that transformers can adapt to the low-dimensionality of $\beta$. In addition, this statistical complexity does not depend on either the degree of the polynomial $\deg(\sigma_*^{test})$ or $\text{ie}(\sigma_*^{test})$, which outperforms CSQ learners and shows a comparable performance with that of SQ learners.

- Flexibility for varying nonlinearity: Our framework allows the link function $\sigma_*$ to vary across tasks. This adaptability is enabled by TTT, which fine-tunes the MLP layer each time using task-specific test data.

- Statistical guarantee for practical settings: We provide an explicit convergence rate with respect to the context length $N_{test}$. This result offers a statistical guarantee in practical scenarios where the dimensions $d, r$ are large but fixed.

## 1.2 RELATED WORKS

**In-context learning and its theoretical analysis**   In-context learning (Brown et al., 2020) is transformer's ability to adapt to the specific task using few labeled examples, without updating any parameters. Agarwal et al. (2024) demonstrated that many in-context examples lead to considerably improved performance, and Jiang et al. (2024) confirmed that many-shot ICL is also beneficial for multimodal tasks. The theoretical background of ICL is extensively studied. For example, a wide array of works (Garg et al., 2023; von Oswald et al., 2023; Zhang et al., 2024; Gatmiry et al., 2024) have shown that linear transformers can be trained to perform linear regression in-context. As for nonlinear transformers, Cheng et al. (2024) demonstrated that nonlinear transformers learn to perform gradient descent and thus learn nonlinear functions. Also, Nichani et al. (2024) analyzed learning of causal structure by softmax transformer. Recently, Dherin et al. (2025) showed that ICL in a single-transformer block(a self-attention layer and subsequent MLP layer) corresponds to low-rank update in MLP layer. Regarding limitations of ICL, Bigoulaeva et al. (2025) argued that pretraining datasets impose a fundamental limit on the model's capability with ICL. Furthermore, (Naim & Asher, 2025) demonstrated that the transformer's ICL ability to generalize functions is limited to certain input values, and found that this limitation comes from layer normalization and softmax attention.

**Test-time training**  Test-time training (Sun et al. (2020), Liu et al. (2021)) updates the model using test data before making predictions, thereby addressing distribution shifts. TTT achieved success in many fields. For example, Bertrand et al. (2023) applied TTT for the video object segmentation task and achieved a significant improvement in the performance. Moreover, Hu et al. (2025) analyzed test-time learning of large language models and achieved at least 20% higher performance on domain knowledge adaptation. Furthermore, Zhang et al. (2025) proposed adopting a large chunk update, and validated the effectiveness of their approach to long-context data through tasks like image sets and language model. As for TTT combined with few-shot prediction, Akyürek et al. (2025) reported that introducing TTT with in-context examples resulted in 6 times higher accuracy in the Abstraction and Reasoning Corpus and a 7.3 percent higher score on BIG-Bench Hard. Finally, regarding the theory behind TTT for in-context learning, Gozeten et al. (2025) analyzed the linear transformer with a single gradient step and characterized the prediction risk of the model with TTT, showing that TTT can mitigate distribution shift.

## 2  Preliminaries and problem settings

**Notations**  Let $\mathrm{He}_i(z) = (-1)^i \mathrm{e}^{\frac{z^2}{2}} \frac{\mathrm{d}^i}{\mathrm{d}z^i} \mathrm{e}^{\frac{-z^2}{2}}$ be the degree-$i$ (probabilist's) Hermite polynomial. $\mathbb{S}^{d-1}$ denotes the unit sphere in $\mathbb{R}^d$. For matrix $\boldsymbol{A}$, we denote its $\ell_2$ operator norm and Frobenius norm as $\|\boldsymbol{A}\|_2$ and $\|\boldsymbol{A}\|_F$, respectively. For a set S, $\mathrm{Unif}(\mathrm{S})$ denotes the uniform distribution over S. $\tilde{O}, \tilde{\Omega}, \tilde{\Theta}$ means $O, \Omega, \Theta$ where polylogarithmic terms of $d$ and $1/\varepsilon$ are hidden. $O_d, \Omega_d, \Theta_d$ means the order with respect to the dimension $d, r$, while $\Theta_\varepsilon$ denotes the order in terms of $\varepsilon$.

### 2.1  In-context learning and test-time training

We consider the basic setting in ICL, which is introduced by Garg et al. (2023) (see Oko et al. (2024) and Lee et al. (2024)). In this setting, the model is given a sequence $(\boldsymbol{x}_1, y_1, \cdots, \boldsymbol{x}_N, y_N, \boldsymbol{x})$ called prompt. The labeled pairs $(\boldsymbol{x}_i, y_i) \in \mathbb{R}^d \times \mathbb{R}$ are called contexts, and $\boldsymbol{x} \in \mathbb{R}^d$ is referred to as query. The model is asked to predict the output that corresponds to $\boldsymbol{x}$ based on the context. The context is sometimes abbreviated as $(\boldsymbol{X}_n, \boldsymbol{y}_n)$ where $\boldsymbol{X}_n = (\boldsymbol{x}_1, \cdots, \boldsymbol{x}_n), \boldsymbol{y}_n = (y_1, \cdots, y_n)$. In this work, we assume that the $\boldsymbol{x}$ and $y$ are generated as follows:

$$\boldsymbol{x}_1^t, \ldots, \boldsymbol{x}_N^t, \boldsymbol{x}^t \overset{\text{i.i.d.}}{\sim} \mathcal{N}(0, \boldsymbol{I}_d), \ y_i^t = f_*^t(\boldsymbol{x}_i^t) + \zeta_i, \ \zeta_i \sim \mathrm{Unif}(-\tau, \tau).$$

ICL aims to predict $y = f_*^t(\boldsymbol{x}) + \zeta$ by mere observation of the context, without updating parameters for each prompt. However, we introduce test-time training to further enhance the model's accuracy. In pretraining, we train the model with $T_{pt}$ distinct datasets $(\boldsymbol{x}_1^t, y_1^t, \cdots, \boldsymbol{x}_{N_{pt}}^t, y_{N_{pt}}^t, \boldsymbol{x}^t, y^t)_{t=1}^{T_{pt}}$, with each prompts consisting of $N_{pt}$ queries. In test-time, we divide the context into four groups with $i$-th group's length $N_i$. This means the test-time prompt is $(\boldsymbol{X}_{N_1}, \boldsymbol{y}_{N_1}, \boldsymbol{X}_{N_2}, \boldsymbol{y}_{N_2}, \boldsymbol{X}_{N_3}, \boldsymbol{y}_{N_3}, \boldsymbol{X}_{N_4}, \boldsymbol{y}_{N_4}, \boldsymbol{x})$. Each group of data plays a different role in test-time training: See Subsection 2.4 for the detail. Let $N_{test} = \sum_{i=1}^4 N_i$ be the total number of contexts test-time.

For the evaluation of the model $f(\boldsymbol{x}, \theta)$ with parameter $\theta$, we define prediction risk as

$$\mathcal{R}_f(\theta) = \mathbb{E}[|f(\boldsymbol{x}, \theta) - y|],$$

where $y = f_*(\boldsymbol{x}) + \zeta$ and the expectation is taken over the prompt $\boldsymbol{x}_i, \boldsymbol{x} \sim \mathcal{D}_{\boldsymbol{x}}, f_* \sim \mathcal{D}_{f_*}, \zeta_i, \zeta \sim \mathcal{D}_\zeta$. Note that $\mathcal{R}_f(\theta)$ not only depends on the dimensions $d, r$ but also on the context length $N$.

### 2.2  Single index model

We consider single-index models for the input-output relationship, where the output depends solely on the direction of the feature vector $\beta$. To predict this relationship accurately, the models are expected to learn the target direction $\beta$ from the high-dimensional data in $\mathbb{R}^d$. Consequently, single-index models have been extensively studied in machine learning theory (Bai & Lee, 2020; Ba et al., 2022; Bietti et al., 2022; Mousavi-Hosseini et al., 2023; Berthier et al., 2024), particularly to examine adaptability to the low-dimensional subspace. The target function $f_*^t(\boldsymbol{x})$ is determined as:

$$f_*^t(\boldsymbol{x}) = \sigma_*^t(\langle \beta^t, \boldsymbol{x} \rangle).$$

1. **Feature vector** The feature vector $\beta$ is chosen uniformly from the unit sphere in an $r$-dimensional subspace. Let $S_r$ be an $r$-dimensional linear subspace in $\mathbb{R}^d$. For each prompt, $\beta^t$ is uniformly drawn from its support $\mathrm{Supp}(\beta) = \{\beta \mid \beta \in S_r, \|\beta\| = 1\}$.

2. **Link function** We take $\sigma_*^t(z) = \sum_{i=Q}^P \frac{c_i^t}{i!} \mathrm{He}_i(z)$ where $1 \leq Q \leq P$. We assume $P$ and $Q$ are constants of $O(1)$. The coefficients $c_i^t$ are drawn from any distribution $\mathcal{D}_{\sigma^*}$ that satisfies

$$\mathbb{E}[c_Q^t] = \Theta(1) \neq 0, \ \sum_{i=Q}^P (c_i^t)^2 \leq R_c = \Theta(1) \text{ (a.s.) and } (c_Q^t, \cdots, c_P^t) \neq (0, \cdots, 0) \text{ (a.s.)}.$$

**Remark 2.** *We draw a new feature vector $\beta^t$ and a new link function $\sigma_*^t(z)$ for each prompt. When $r \ll d$, $\beta^t$ has low-dimensional support. We will later demonstrate that our model can leverage this low-dimensionality. Note that our analysis is also valid when $r = d$. As for the link function, we do not assume a specific distribution of the coefficients $c_i$: our framework allows different distributions as long as they satisfy the assumptions above.*

For simplicity, we assume that $N_{pt}, T_{pt}, N_1, N_2, N_3, N_4 = \mathrm{poly}(d)$, and there exists $\alpha_r > 0$ that $r = \Omega(d^{\alpha_r})$, which means that $r$ grows faster than $\mathrm{polylog}(d)$.

The complexity of learning single-index model is governed by three key quantities: the degree of the polynomial $\deg(\sigma_*)$, the information exponent $\mathrm{ie}(\sigma_*)$, and the general exponent $\mathrm{ge}(\sigma_*)$.

- The information exponent (Dudeja & Hsu, 2018; Arous et al., 2021) $\mathrm{ie}(\sigma_*)$ is the smallest non-zero degree of the hermite expansion of $\sigma_*$.

- The general exponent (Damian et al., 2024) $\mathrm{ge}(\sigma_*)$ is the minimum of $\mathrm{ie}(f \circ \sigma_*)$ with respect to all the $L_2$- measurable transformation $f$.

By definition, $\deg(\sigma_*) \geq \mathrm{ie}(\sigma_*) \geq \mathrm{ge}(\sigma_*)$ holds. This relationship characterizes the statistical complexity required by different algorithms.

- The kernel method, which relies on pre-determined feature maps, requires the sample complexity $n \gtrsim d^{\Theta(\deg(\sigma_*))}$ to ensure low predictive error (Ghorbani et al., 2020; Donhauser et al., 2021).

- The models with access to correlational statistical query (CSQ), of the form $\mathbb{E}[\phi(\boldsymbol{x})y]$, require a sample size of $n \gtrsim d^{\Theta(\mathrm{ie}(\sigma_*))}$, known as CSQ lower bound (Damian et al., 2022; Abbe et al., 2023). This improved sample complexity, independent of $\deg(\sigma_*)$, has been achieved by models like two-layer neural network with online SGD (Arous et al., 2021; Damian et al., 2023) or one-step gradient descent(Damian et al., 2022; Dandi et al., 2025).

- For the broader class of statistical queries (SQ), which takes the form $\mathbb{E}[\phi(\boldsymbol{x}, y)]$, the sample complexity is further improved to $n \gtrsim d^{\Theta(\mathrm{ge}(\sigma_*))}$. This advantage comes from applying nonlinear transformations to the label $y$, thereby reducing $\mathrm{ie}(\sigma_*)$. Recent works have achieved this complexity by reusing the data (Lee et al., 2024; Arnaboldi et al., 2025) or adjusting the loss function (Joshi et al., 2024). Furthermore, when $\sigma_*$ is a polynomial, the following result holds:

**Lemma 3** (Lee et al. (2024), Proposition 6). *It holds that* $\mathrm{ge}(\sigma_*) = \begin{cases} 1 \ (\textit{if } \sigma_* \textit{ is not even}) \\ 2 \ (\textit{if } \sigma_* \textit{ is even}) \end{cases}$.

*Moreover,* $\mathrm{ge}(\sigma_*) = \min_{j \geq 1} \mathrm{ie}(\sigma_*^j)$ *holds.*

This implies that, for any polynomial $\sigma_*$, $\mathrm{ge}(\sigma_*) \leq 2$ is a small constant, regardless of $\deg(\sigma_*)$.

Our goal is to achieve the test-time sample complexity of $N_{test} = r^{\Theta(\mathrm{ge}(\sigma_*))}$, which is independent of the entire dimension $d$ and surpasses the CSQ limit.

## 2.3 STUDENT MODEL

To facilitate our theoretical analysis, we need to establish a concrete architecture for the student model. We employ a single-layer transformer model, formulated as follows. In pretraining, we use the same model as Nishikawa et al. (2025). We first construct the embedding $\boldsymbol{E}$ as

$$\boldsymbol{E} = \begin{bmatrix} \boldsymbol{x}_1 & \cdots & \boldsymbol{x}_N & \boldsymbol{x} \\ y_1 & \cdots & y_N & 1 \end{bmatrix} \in \mathbb{R}^{(d+1)\times(N+1)}.$$ Then, we apply the softmax attention layer as

$$\mathrm{Attn}(\boldsymbol{E}) = \boldsymbol{W}^V \boldsymbol{E} \cdot \mathrm{softmax}(\mathrm{Mask}(\rho^{-1} \cdot (\boldsymbol{W}^K \boldsymbol{E})^\top \boldsymbol{W}^Q \boldsymbol{E})),$$

where $\rho$ is temperature and $\boldsymbol{W}^V, \boldsymbol{W}^K, \boldsymbol{W}^Q \in \mathbb{R}^{(d+1)\times(d+1)}$ are the parameters for attention layer. The softmax is applied to each column, while the Mask function converts all the elements in the final row into $-\infty$ to prevent the model from focusing on the uninformative final row. We further apply a multi-layer perceptron (MLP) layer. For the activation function, we use ReLU function $\sigma(z) = \max\{0, z\}$ throughout the paper. With the parameters $\boldsymbol{W}^F \in \mathbb{R}^{m\times(d+1)}, \boldsymbol{b} \in \mathbb{R}^m$, and $\boldsymbol{a} \in \mathbb{R}^m$ where $m$ is the network width, the model's output is

$$f_{\mathrm{IC}}(\Gamma, \boldsymbol{X}_N, \boldsymbol{y}_N, \boldsymbol{x}) = \mathrm{MLP} \circ \mathrm{Attn}(\boldsymbol{E})_{:,N+1} = \boldsymbol{a}^\top \sigma(\boldsymbol{W}^F \mathrm{Attn}(\boldsymbol{E})_{:,N+1} + \boldsymbol{b}),$$

where $\sigma$ is applied entry-wise. Finally, we adopt some simplification. Let $\boldsymbol{W}^{KQ} = (\boldsymbol{W}^K)^\top \boldsymbol{W}^Q \in \mathbb{R}^{(d+1)\times(d+1)}$ and $\boldsymbol{W}^{FV} = \boldsymbol{W}^F \boldsymbol{W}^V \in \mathbb{R}^{(m+1)\times(d+1)}$. We use the following parametrization:

$$\boldsymbol{W}^{KQ} = \begin{bmatrix} \Gamma & \boldsymbol{0}_{d\times 1} \\ \boldsymbol{0}_{1\times d} & 1 \end{bmatrix}, \ \boldsymbol{W}^{FV} = \begin{bmatrix} \boldsymbol{O}_{(m+1)\times d} & \boldsymbol{v} \end{bmatrix},$$

for $\Gamma \in \mathbb{R}^{d\times d}$ and $\boldsymbol{v} \in \mathbb{R}^{(d+1)\times 1}$. This kind of simplification, specifying some of the parameters as zero, is often adopted in many theoretical works on transformers (Zhang et al., 2023; Huang et al., 2023; Kim & Suzuki, 2025). Overall, the model's output is written as

$$f_{\mathrm{IC}}(\Gamma, \boldsymbol{X}_N, \boldsymbol{y}_N, \boldsymbol{x}) = \sum_{j=1}^m a_j \sigma \left( v_j \frac{\sum_{i=1}^N y_i \mathrm{e}^{y_i/\rho} \mathrm{e}^{\boldsymbol{x}_i^\top \Gamma \boldsymbol{x}/\rho}}{\sum_{i=1}^N \mathrm{e}^{y_i/\rho} \mathrm{e}^{\boldsymbol{x}_i^\top \Gamma \boldsymbol{x}/\rho}} + b_j \right).$$

See Appendix A in Nishikawa et al. (2025) for how to derive this equation. At test-time, we adopt low-rank adaptation(LoRA). Specifically, we change the attention matrix $\Gamma^*$ as $\Gamma_u = \Gamma^* + \boldsymbol{u}^\top \boldsymbol{u}$, where $\Gamma^*$ is fixed during test-time and $\boldsymbol{u} \in \mathbb{R}^d$ is a trainable parameter vector. Our goal is to find $\hat{\boldsymbol{u}} \approx \beta$ using test-time context data. See section 2.4 for how test data is used to find $\hat{\boldsymbol{u}}$. The final prediction of the model is as follows:

$$f_{\mathrm{TF}}(\boldsymbol{x}, \hat{\boldsymbol{u}}, \boldsymbol{v}, \boldsymbol{a}, \boldsymbol{b}) = \sum_{j=1}^m a_j \sigma(v_j \langle \hat{\boldsymbol{u}}, \boldsymbol{x} \rangle + b_j).$$

**Remark 4.** *The use of in-context data in attention output leads to data-efficient prediction, but it prevents the model from achieving near-zero error. This is because of the high order term of $\langle \beta, \boldsymbol{x} \rangle$. When $N$ is sufficiently large, the denominator of the attention output $N^{-1} \sum_{i=1}^N \mathrm{e}^{y_i/\rho} \mathrm{e}^{\boldsymbol{x}_i^\top \Gamma \boldsymbol{x}/\rho}$ approximates $\mathbb{E}[\exp \sigma_*(\langle \beta, \boldsymbol{x}_1 \rangle /\rho) \exp(\langle \Gamma_* \boldsymbol{x}, \boldsymbol{x}_1 \rangle /\rho)]$, as explained in Nishikawa et al. (2025). The key of their work is, since $\exp(\langle \Gamma_* \boldsymbol{x}, \boldsymbol{x}_1 \rangle /\rho)$ contains all the Hermite coefficients and $\exp \sigma_*(\langle \beta, \boldsymbol{x}_1 \rangle /\rho)$ has nonzero coefficient for $\mathrm{He}_{\mathrm{ge}(\sigma_*)}$, this attention layer can compute $\langle \beta, \boldsymbol{x} \rangle^{\mathrm{ge}(\sigma_*)}$. However, this means that the output of the attention module contains $\langle \beta, \boldsymbol{x} \rangle^i$ for $i \geq \mathrm{ge}(\sigma_*) + 1$ as well. This causes inevitable prediction error, even when $N \to \infty$. Meanwhile, when we find $\hat{\boldsymbol{u}}$ that is sufficiently near the feature vector $\beta$, we can achieve near-zero error. Therefore we do not use in-context data in the final prediction $f_{\mathrm{TF}}$.*

## 2.4 TRAINING ALGORITHM

We employ a gradient-based training algorithm, as specified in Algorithm 1. The training procedure consists of pretraining and test-time training, with the latter divided into three stages.

• Pretraining: We optimize $\Gamma$ via one-step gradient descent over $T_{pt}$ prompts. This scheme is originally taken from Nishikawa et al. (2025). The effectiveness of one-step gradient descent is confirmed by works such as Ba et al. (2022) and Damian et al. (2022), which demonstrate that one-step gradient already captures the key feature.

• TTT stage I: We apply a single gradient descent step to $\boldsymbol{u}^{(0)}$ with $L_2$-regularization. This stage prevents catastrophic forgetting, a phenomenon where the parameter change caused by TTT deprives the model of the fundamental ability to adapt to the original task. The goal of this stage is

**Algorithm 1:** Pretraining and test-time training of transformer

**Input** : Learning rate $\eta_{pt}, \eta_1, \eta_2$, regularization rate $\lambda_{pt}, \lambda_1$, initialization scale $\alpha_{pt}, \alpha_1, \alpha_2$, dimensions $d, r$, temperature $\rho$.

1 **Initialize** $\Gamma(0) \sim \boldsymbol{I}_d/\sqrt{d},\ \boldsymbol{v}(0) \sim \mathrm{Unif}(\{\pm 1\}^m),\ \boldsymbol{b}(0) = \boldsymbol{0}_m,\ \boldsymbol{a}(0) = \alpha_{pt}\boldsymbol{1}_m.$

2 **Pretraining: One-step Gradient descent on Attention Matrix**

3    $\Gamma^* \leftarrow \Gamma(0) - \eta_{pt}\frac{1}{2T_{pt}}\sum_{t=1}^{T_{pt}}\nabla_\Gamma((f_{\mathrm{IC}}(\Gamma, \boldsymbol{X}_{N_{pt}}, \boldsymbol{y}_{N_{pt}}, \boldsymbol{x}^t) - y^t)^2 + \lambda_{pt}\|\Gamma\|_F^2).$

4 **Test-Time Training**

5    $\boldsymbol{a}(0) = \alpha_1\boldsymbol{1}_m.$

6    **for** $i = 1$ **to** $N_1$ **do**

7      $\boldsymbol{x}_i \leftarrow \sqrt{r}\Gamma^*\boldsymbol{x}_i.$

8    $\boldsymbol{u}^{(0)} \sim \mathcal{N}(0, \boldsymbol{I}_d),\ \boldsymbol{u}^{(0)} \leftarrow \sqrt{r}\Gamma_*\boldsymbol{u}^{(0)},\ \boldsymbol{u}^{(0)} \leftarrow \frac{\boldsymbol{u}^{(0)}}{\sqrt{r}\|\boldsymbol{u}^{(0)}\|}.$

9    **Stage I: Initialization of u with signals from Original Model**

10      Draw $\boldsymbol{w}_1, \cdots, \boldsymbol{w}_{N_{new}} \overset{\mathrm{i.i.d.}}{\sim} \mathcal{N}(0, \boldsymbol{I}_d).$

11      $b = \frac{1}{N_{new}}\sum_{i=1}^{N_{new}} g(\Gamma^*, \boldsymbol{X}_{N_1}, \boldsymbol{y}_{N_1}, \boldsymbol{w}_i),$

12      $\boldsymbol{u}^{(1)} = \boldsymbol{u}^{(0)} - \eta_1\{\sqrt{r}\Gamma^*\nabla_{\boldsymbol{u}}\frac{1}{2N_{new}}\sum_{i=1}^{N_{new}}(f_{\mathrm{IC}}(\Gamma_{\boldsymbol{u}}, \boldsymbol{X}_{N_1}, \boldsymbol{y}_{N_1}, \boldsymbol{w}_i) -$

       $(g(\Gamma_*, \boldsymbol{X}_{N_1}, \boldsymbol{y}_{N_1}, \boldsymbol{w}_i) - b))^2 + \frac{\lambda_1}{2}\nabla_{\boldsymbol{u}}\|\boldsymbol{u}\|^2\},$

13      $\boldsymbol{u}^{(1)} \leftarrow \frac{\boldsymbol{u}^{(1)}}{\|\boldsymbol{u}^{(1)}\|}.$

14    $\boldsymbol{a}(0) = \alpha_2\boldsymbol{1}_m.$

15    **for** $i = N_1 + 1$ **to** $N_1 + N_2$ **do**

16      $\boldsymbol{x}_i \leftarrow \sqrt{r}\Gamma^*\boldsymbol{x}_i.$

17    **Stage II: Strong Recovery**

18      **for** $t = 1$ **to** $t = N_3$ **do**

19        $\boldsymbol{u}^{(t+1)} = \boldsymbol{u}^{(t)} - \eta_2\sqrt{r}\Gamma^*\nabla_u(\frac{1}{2}(f_{\mathrm{IC}}(\Gamma_u, \boldsymbol{X}_{N_2}, \boldsymbol{y}_{N_2}, \boldsymbol{x}_{N_1+N_2+t}) - y_{N_1+N_2+t})^2).$

       $\boldsymbol{u}^{(t+1)} \leftarrow \frac{\boldsymbol{u}^{(t+1)}}{\|\boldsymbol{u}^{(t+1)}\|}.$

20    **Initialize** $b_j^* \sim \mathrm{Unif}([-\log^2 d, \log^2 d]),\ \boldsymbol{v}^* = \boldsymbol{v}(0).$

21    **Stage III: Training of MLP Layer**

22      $\boldsymbol{a}^* \leftarrow$

       $\arg\min_{\boldsymbol{a}} \frac{1}{2N_4}\sum_{t=N_1+N_2+N_3+1}^{N_1+N_2+N_3+N_4}(f_{\mathrm{TF}}(\boldsymbol{x}_t, \boldsymbol{u}^{(N_3+1)}, \boldsymbol{v}^*, \boldsymbol{a}, \boldsymbol{b}^*) - y_t)^2 + \frac{\lambda_2}{2}\|\boldsymbol{a}\|^2.$

**Output:** Prediction $f_{\mathrm{TF}}(\boldsymbol{x}, \boldsymbol{u}^{(N_3+1)}, \boldsymbol{v}^*, \boldsymbol{a}^*, \boldsymbol{b}^*).$

to find a good initial value of $\boldsymbol{u}^{(1)}$ that satisfies $\langle \beta, \boldsymbol{u}^{(1)}\rangle \geq 1/\mathrm{polylog}(d)$. For this stage, we use the output from the attention layer of the original model as a teacher signal, which is defined as $g(\Gamma_*, \boldsymbol{X}_{N1}, \boldsymbol{y}_{N_1}, \boldsymbol{x}) = \frac{N_1^{-1}\sum_{i=1}^{N_1} y_i e^{y_i/\rho} e^{\boldsymbol{x}_i^\top \Gamma_* \boldsymbol{x}}}{N_1^{-1}\sum_{i=1}^{N_1} e^{y_i/\rho} e^{\boldsymbol{x}_i^\top \Gamma_* \boldsymbol{x}}}$. In this stage, the query $\boldsymbol{x}$ do not require ground-truth label $y$. Therefore, we use $N_{new}$ newly generated vectors $w_1, \ldots, w_{N_{new}} \overset{\mathrm{i.i.d.}}{\sim} \mathcal{N}(0, \boldsymbol{I}_d)$ for the query. The necessity of this stage is further discussed in Subsection 3.1.2.

- TTT stage II: We continue to optimize $\boldsymbol{u}$ by applying multi-step online SGD scheme that originates from Lee et al. (2024), using $(\boldsymbol{X}_{N_2}, \boldsymbol{y}_{N_2})$ as contexts. This stage aims to align $\boldsymbol{u}$ more closely to the target $\beta$. As we will see, increasing the number of the SGD steps leads to the convergence of $\boldsymbol{u}$ to $\beta$.

- TTT stage III: Finally, we train the MLP layer to fit to the nonlinear link function $\sigma_*^{\mathrm{test}}$. Specifically, we randomize $\boldsymbol{v}$ and $\boldsymbol{b}$ and optimize $\boldsymbol{a}$ with ridge regression, following the recipe in Nishikawa et al. (2025). This problem is convex with respect to $\boldsymbol{a}$, ensuring that the global optimum can be readily found.

## 3   MAIN RESULT

Based on the problem settings above, we are now ready to present our main result.

**Theorem 1** (Formal). *We denote the link function drawn in inference-time as $\sigma_*^{\text{test}}$. Suppose that $T_{pt}, N_{pt} = \tilde{\Omega}(r^2 d^{Q+2})$, $N_1 = \tilde{\Omega}(r^{\text{ge}(\sigma_*^{\text{test}})+2})$, $N_{new} = \tilde{\Omega}(r^{\text{ge}(\sigma_*^{\text{test}})+2})$, $N_2 = \tilde{\Theta}(r^2)$. Moreover, we assume that $m, N_{pt}, T_{pt}, N_1, N_2, N_3, N_4 = O(\text{poly}(d))$ and there exists $\alpha_r > 0$ that $r = \Omega(d^{\alpha_r})$. When we fix $d$ large enough, then there exists $\lambda_{pt}, \lambda_1, \eta_{pt}, \eta_1, \eta_2$ such that the prediction risk is low with probability at least 0.99 over the training data and random initialization. Concretely, for the model trained via Algorithm 1, we have that*

$$|\mathcal{R}_{f_{\text{TF}}}(\boldsymbol{u}, \boldsymbol{v}^*, \boldsymbol{a}^*, \boldsymbol{b}^*) - \tau| = \tilde{O}(N_4^{-1/2}) + \tilde{O}(m^{-1/2}) + \tilde{O}\left(\sqrt{\frac{r\sqrt{r}}{N_3}}\right),$$

*with probability at least 0.99.*

The proof is given in Appendix G. Our result has the following advantages compared to previous works.

**(i) Nonlinearity**   While Gozeten et al. (2025) also investigates test-time training combined with ICL, their analysis is restricted to linear datasets and a transformer with linear attention. By contrast, we consider a more complex and general problem of prediction for a single-index model with a nonlinear polynomial. In addition, the student model in our work utilizes nonlinear softmax attention, thereby extending the analysis beyond linear transformers.

**(ii) The adaptability to link function**   Our framework has the flexibility of using a task-specific link function $\sigma_*^t$. In the previous work by Nishikawa et al. (2025), the link function is fixed throughout all the tasks (only $\beta$ varies across different tasks) because the training of the relevant layer is done only in pretraining. In contrast, the use of test-time training allows the model to adapt to the characteristics of each task. Therefore, our algorithm is effective even when the underlying link function varies from one task to another.

**(iii) The convergence rate of predictive loss with respect to n**   The analysis by Nishikawa et al. (2025) guarantees that the ICL risk is $o_d(1)$, but it provides no guarantee that the risk diminishes as $n \to \infty$. In fact, this is an inherent consequence of adopting softmax attention, as we discussed in Remark 4. In contrast, our result overcomes that limitation. When $d$ is a sufficiently large constant, our theory ensures that the prediction risk can be made arbitrarily close to the inevitable noise $\tau$ by increasing the context size $N_{test}$ and the network width $m$. This is because the increase in the test-time context length $N_3$ allows the vector $\boldsymbol{u}$ to converge to the ground truth $\beta$.

In addition, it is worth noting that we achieve efficient sample complexity. Theorem 1 implies that $N_{test} = \tilde{\Omega}(r^{2+\text{ge}(\sigma_*^{\text{test}})})$ to ensure low predictive loss $o(1)$, yielding two key benefits. First, our sample complexity does not depend on the entire dimension $d$, which means this model also adapts to the low-dimensionality of $\beta$. Second, our sample complexity is independent of either $\deg(\sigma_*^{\text{test}})$ or $\text{ie}(\sigma_*^{\text{test}})$, which breaks the CSQ upper bound. Moreover, for polynomials where $\text{ge}(\sigma_*) \leq 2$ holds, the required number of samples is small. In conclusion, our sample complexity is nearly optimal. It is instructive to compare this with Nishikawa et al. (2025). Although they also achieve sample complexity with these two features ($N_{test} = \tilde{\Omega}(r^{3\text{ge}(\sigma_*)/2})$), their result guarantees convergence with respect to $d$, not $N_{test}$. This implies that their result is valid only in the asymptotic limit where $d \to \infty$. Moreover, they do not provide the convergence rate with respect to $d$. In contrast, our result holds for any sufficiently large fixed $d$.

## 3.1 Proof sketch

The outline of the proof of our main theorem is as follows: using the output of the original model, we can achieve weak recovery i.e. nontrivial overlap between $\beta$ and $\boldsymbol{u}$. Once weak recovery is completed, more training using the signal $y$ leads to strong recovery i.e. $\langle \beta, \boldsymbol{u} \rangle \geq 1 - \varepsilon$. Finally, the training of MLP layer allows the model to fit to the nonlinearity of the link function.

### 3.1.1 Exploiting the pretrained attention matrix

As shown in Nishikawa et al. (2025), the attention matrix $\Gamma^*$ after pretraining captures the $r$-dimensional Span of $\beta$:

**Lemma 5** (Informal, Correspond to Proposition 22 in Nishikawa et al. (2025))**.** *After running the pretraining in Algorithm 1 with $T_{pt}, N_{pt} = \tilde{\Omega}(r^2 d^{Q+2})$, it holds that*

$$\Gamma^* \approx c_r \mathbb{E}_\beta[\beta\beta^\top]$$

*with high probability, where $c_r = \tilde{\Theta}(\sqrt{r})$.*

This means that $\Gamma^*$ can project vectors into the $r$-dimensional subspace $\mathrm{Supp}(\beta)$. Therefore, by multiplying $\sqrt{r}\Gamma^*$ to $\boldsymbol{x}_i$ and the gradient (the coefficient $\sqrt{r}$ is just for adjusting the scale), we can make the problem virtually $r$-dimensional, even though the entire dimension is $d$. This leads to the sample complexity only scaling up with $r$, not the whole dimension $d$.

### 3.1.2 WEAK RECOVERY

First, we initialize $\boldsymbol{u}$ by one step gradient descent using the output from the original model $g(\Gamma_*, \boldsymbol{X}_{N_1}, \boldsymbol{y}_{N_1}, w_i)$. We do not use the signal $y$ in this process. Intuitively, this self-distillation can be seen as a prevention of catastrophic forgetting. Taking the information from the original model makes the LoRA model similar to the original model, thereby preserving the desired features of the original model.
To clarify why we need to use the original model as teacher, consider training the LoRA model with the true signal $y$. Then, by calculating the gradient, we can get the following:

**Lemma 6** (Informal)**.** *The following holds with high probability:*

$$\frac{1}{2}\sqrt{r}\Gamma^*\nabla_{\boldsymbol{u}}(f_{\mathrm{IC}}(\Gamma, \boldsymbol{X}_N, \boldsymbol{y}_N, \boldsymbol{x}) - y)^2 \approx \tilde{\Theta}(\alpha m)\langle\beta, \boldsymbol{u}\rangle\{(\sqrt{r})^{-(\mathrm{ie}(\sigma_*)-1)} + \langle\beta, \boldsymbol{u}\rangle^{2\mathrm{ie}(\sigma_*)-2}\}\beta.$$

See Appendix C for details. At initialization $\langle\beta, \boldsymbol{u}\rangle = \tilde{O}(1/r)$ holds, so the signal strength is $r^{\Theta(\mathrm{ie}(\sigma_*^{\mathrm{test}}))}$. Therefore, when we train $\boldsymbol{u}$ from the signal $y$, we need at least $r^{\Theta(\mathrm{ie}(\sigma_*^{\mathrm{test}}))}$ data.
However, using the original model as teacher signal reduces this data length to $r^{\Theta(\mathrm{ge}(\sigma_*^{\mathrm{test}}))}$. As Nishikawa et al. (2025) showed, the attention layer after pretraining can compute $\langle\beta, \boldsymbol{x}\rangle^{\mathrm{ge}(\sigma_*^{\mathrm{test}})}$ in-context. Then, noting that $\mathrm{ie}(\mathrm{He}_{\mathrm{ge}(\sigma_*)}) = \mathrm{ge}(\sigma_*)$, when we learn from the original model, the signal strength becomes $r^{\Theta(\mathrm{ge}(\sigma_*^{\mathrm{test}}))}$. This improved signal strength results in the required data length $N_{test} = r^{\Theta(\mathrm{ge}(\sigma_*^{\mathrm{test}}))}$, which surpasses CSQ limit.

### 3.1.3 STRONG RECOVERY

Weak recovery is insufficient for reliably predicting $\langle\beta, \boldsymbol{x}\rangle$ using only $\boldsymbol{u}$. Therefore, we further optimize $\boldsymbol{u}$ until we achieve strong recovery, defined as $\langle\beta, \boldsymbol{u}\rangle \geq 1-\varepsilon$ for a small error tolerance $\varepsilon > 0$. After achieving $\langle\beta, \boldsymbol{u}^{(1)}\rangle > 1/\mathrm{polylog}(d)$, the signal strength is $O(1/\mathrm{polylog}(d))$, independent of $\mathrm{ge}(\sigma_*^{\mathrm{test}})$. Consequently, the sample size required to achieve strong recovery does not depend on $\mathrm{ge}(\sigma_*^{\mathrm{test}})$. Moreover, we achieve a smaller sample complexity for achieving strong recovery compared to the prior work by Lee et al. (2024). While their work suggested a linear increase in $\langle\beta, \boldsymbol{u}^{(n)}\rangle$, which resulted in the required data length $N = \Theta_\varepsilon(\varepsilon^{-2})$, we find that beyond a certain point, the error $1 - \langle\beta, \boldsymbol{u}^{(n)}\rangle$ converges to 0 geometrically. This accelerated geometric convergence yields an improved sample complexity of $N = \Theta_\varepsilon(\frac{1}{\varepsilon}\log\frac{1}{\varepsilon})$. See Appendix E for further details.

### 3.1.4 ESTIMATION OF THE LINK FUNCTION

Finally, we train MLP layer to predict the link function $\sigma_*^{\mathrm{test}}$. Following prior work Nishikawa et al. (2025), we show that training MLP layer leads to small empirical loss. Moreover, we derive the upper bound of the predictive risk using Rademacher complexity. See Appendix F for details.

## 4 SYNTHETIC EXPERIMENT

We conducted a numerical experiment to examine the effectiveness of TTT compared to ICL. To evaluate whether our theory holds with real-world settings, we trained a GPT-2 model (Radford

et al., 2019) to learn the Gaussian single-index functions. See Appendix I for further experimental details.

The dimensions were set to $d = r = 16$. For each task $t$, the data was generated as follows: $\boldsymbol{x}_1^t, \cdots, \boldsymbol{x}_N^t, \boldsymbol{x}^t \overset{\text{i.i.d.}}{\sim} \mathcal{N}(0, \boldsymbol{I}_d)$, $\beta^t \sim \text{Unif}(\mathbb{S}^{r-1})$ and $y_i^t = \sigma_*^t(\langle \beta, \boldsymbol{x}_i \rangle) = \frac{1}{\sqrt{3!}} \text{He}_3(\langle \beta, \boldsymbol{x}_i \rangle) + \frac{c^t}{\sqrt{4!}} \text{He}_4(\langle \beta, \boldsymbol{x}_i \rangle)$, where $c^t \sim \text{Unif}(-0.5, 0.5)$. We compared the following two settings:

1. In-context learning, configured as Garg et al. (2023), Oko et al. (2024) and Nishikawa et al. (2025). We first pretrained the model with the data $(\boldsymbol{X}^t, \boldsymbol{y}^t, \boldsymbol{x}^t, y^t)_{t=1}^{T_{pt}}$, then we calculated the predictive loss.

2. Test-time training: We adapted the same pre-trained model as we used in the evaluation of ICL by fine-tuning the LoRA parameters applied to the attention and MLP layers.

Figure 1a highlights the result. Contrary to our initial expectation, the pretrained model's ICL ability is powerful enough to adapt to the varying link function, while TTT does not lead to significant improvement of the prediction. This discrepancy is likely due to the difference between our theoretical setting and the practical experimental setup. While our theoretical analysis assumed a single-layer transformer consisting of the attention layer followed by the MLP layer, the experiment utilized a 6-layer GPT-2 model. As noted by Oko et al. (2024), MLP layers followed by linear attention (a structure present in GPT-2) are capable of adapting to nonlinear link functions. It is plausible that the GPT-2 model learned to fit the task-specific link function during pretraining, leaving little room for improvement via TTT. Nevertheless, we observe that TTT at least does not degrade performance even when ICL functions effectively.

This result spurred us to analyze the limitation of ICL and the true potential of TTT in more challenging tasks: we changed the distribution of link function as $y_i^t = \frac{c_3^t}{\sqrt{3!}} \text{He}_3(\langle \beta, \boldsymbol{x}_i \rangle) + \frac{c_4^t}{\sqrt{4!}} \text{He}_4(\langle \beta, \boldsymbol{x}_i \rangle)$, where $c_3^t \sim \text{Unif}(0.5, 1.5), c_4^t \sim \text{Unif}(-0.5, 0.5)$, only in test-time. The result, summarized in Figure 1b, clearly shows the advantage of TTT over ICL. As the data length grows, the accuracy of TTT steadily improves, substantially surpassing the performance of ICL. Taken together, these results highlight TTT as a robust strategy that matches ICL in standard settings while significantly outperforming it under distribution shifts.

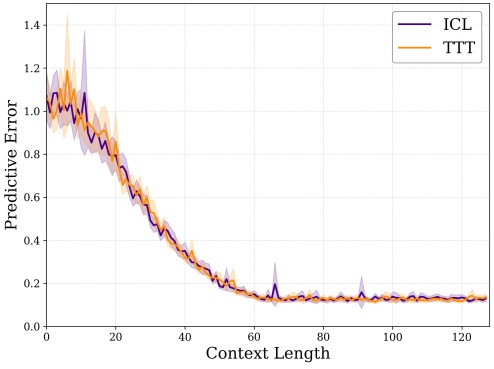 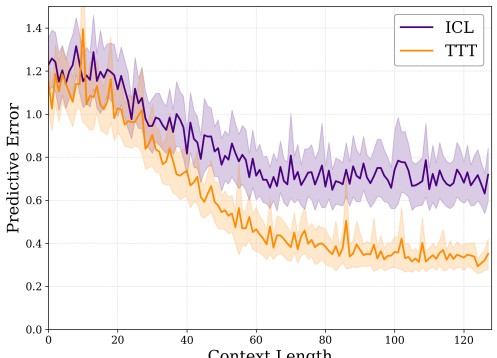

(a) In-distribution setting: The distribution of the link function is consistent between pretraining and test-time.

(b) Out-of-distribution setting: The distribution of the link function at test-time differs from that of pre-training

Figure 1: The predictive error of In-context learning (ICL) and Test-time training (TTT) for a pre-trained GPT-2 model on single-index polynomials (see Section 4 for details). While (a) shows that ICL effectively adapts to varying link functions for in-distribution tasks, (b) demonstrates that TTT outperforms ICL under distribution shift, showing its superior adaptability.

To further assess the capabilities of TTT, we examined its ability to leverage low-dimensional task structures. We fixed the ambient dimension at $d = 16$ and compared the predictive error in the in-distribution setting for a low intrinsic dimension ($r = 4$) versus a full-rank intrinsic dimension ($r = 16$). As illustrated in Figure 2, the predictive error is substantially lower for $r = 4$ than for

$r = 16$, even though the ambient dimension $d$ remains the same. This finding highlights that the TTT-equipped model effectively adapts to the task's intrinsic low-dimensionality.

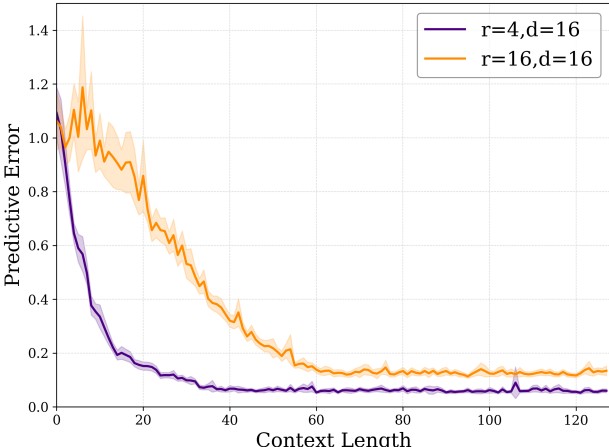

Figure 2: The predictive error of Test-time training for a pretrained GPT-2 model on single-index polynomials, comparing different intrinsic dimensions $r$ (with fixed ambient dimension $d = 16$). Notably, when the intrinsic dimension is low ($r = 4$), the error remains minimal even with very short context lengths.

## 5 CONCLUSION

We have investigated test-time training combined with in-context learning. We provided an upper bound of the predictive loss in terms of $m$ and $N$. Our result shows that for a large test-time sample size $N_{test}$, the predictive loss approaches the inevitable noise $\tau$, even when $d$ remains finite.

**Future work and limitation** We outline some limitations and future research directions.

- The result of the numerical experiment demonstrates that the ICL model's loss decreases as we increase $n$ in in-distribution cases. This finding is contrary to our theoretical expectation that ICL cannot adapt to the change in link function. Future work could clarify whether ICL without TTT can manage varying link functions.

- We only considered the single-index model where the link function $\sigma_*$ is a polynomial. Investigating a more general class of input-output relationships is a possible extension of this work.

- In this work, we assumed that the test-time distribution of the feature vector $\beta$ is the same as in the pretraining. Considering a distribution shift, such as investigating the situation where $\mathrm{Supp}(\beta)_{test}$ is slightly different from $\mathrm{Supp}(\beta)_{pt}$ is another interesting direction.

- Algorithm 1 divides the test-time training into 3 stages, training different layers sequentially. This differs from the typical situation where the entire model is trained at once, as was done in the experiment. Whether a similar upper bound of the predictive risk can be established in such settings remains to be examined.

## LLM USAGE STATEMENT

Large language models are used for three purposes: to proofread and polish English writing, to help us find related works, and to write the code for the synthetic experiment. We did not use any LLM assistant for designing the problem settings and constructing the proofs.

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

CONTENTS

## A  DEFINITION OF HIGH PROBABILITY EVENTS

**Definition 7.** *We say that an event $A$ occurs with high probability when the following holds:*

$$1 - P(A) \le O(d^{-C_*}),$$

*where $C_*$ is a sufficiently large constant that is independent of $d$ and $r$.*

Note that we can redefine $C_*$ to be sufficiently large if needed. A basic example is the Gaussian tail bound. When $x \sim \mathcal{N}(0,1)$ and $t > 0$, we have

$$P(|x| > t) \le 2\exp(-t^2/2).$$

Thus, by letting $t = \sqrt{2C_* \log d}$, we see that

$$P(|x| > \sqrt{2C_* \log d}) \le 2d\exp(-C_*) = O(d^{-C_*}).$$

In such a situation, we say that $|x| = O(\sqrt{\log d})$ with high probability.

When $A_1, \ldots, A_M$ occurs with high probability where $M = \text{poly}(d)$, then the intersection $A_1 \cap A_2 \cap \cdots \cap A_M$ occurs with high probability. In this paper, we assume that $N_{pt}, T_{pt}, N_1, N_2, N_3, N_4, m = O(\text{poly}(d))$. Because of this assumption, when we take the intersection of high probability events $A_1, \ldots, A_M$, $M = \text{poly}(d)$ is satisfied.

## B  PRETRAINING

We follow the pretraining algorithm that was considered in Nishikawa et al. (2025). In the original paper, the link function $\sigma_*$ is fixed across all tasks, whereas in this paper, the link function varies depending on the tasks. We should take this difference into account and ensure that the pretraining algorithm works properly even in our setting.

**Lemma 5** (Formal). *After running the pretraining in Algorithm 1 with $T_{pt}, N_{pt} = \tilde{\Omega}(r^2 d^{Q+2})$, it holds that*

$$\Gamma^* = \frac{1}{r^{1/2}\log^{C_\kappa} d}\left(r\mathbb{E}_\beta[\beta\beta^\top] + \boldsymbol{N}\right)$$

*with high probability over the data distribution, where $\|\boldsymbol{N}\|_F = O_d(1/\sqrt{d})$ holds, where $C_\kappa$ can be taken to be sufficiently large.*

*Proof.* We only consider the difference between our settings and Nishikawa et al. (2025). See Section C of Nishikawa et al. (2025) for the original argument.

If we fix $t$, we may apply Lemma 19 and Lemma 20 in Nishikawa et al. (2025). In addition to that, since the norm upper bound remains unchanged, we can bound the difference between the expectation and the actual value just like Eq.(C.3) and Eq.(C.4) in Lemma 21 of Nishikawa et al. (2025). What remains is to calculate $\mathbb{E}_{\boldsymbol{x},\beta,t}[yz^k\beta\boldsymbol{x}^\top] = (\rho\sqrt{d})^{-k}\mathbb{E}_\beta[\mathbb{E}_{\boldsymbol{x},t}[\sigma_*'^t(\langle\beta,\boldsymbol{x}\rangle)(\langle\beta,\boldsymbol{x}\rangle)^k]\beta\beta^\top] + (\rho\sqrt{d})^{-k}k\mathbb{E}_\beta[\mathbb{E}_{\boldsymbol{x},t}[\sigma_*^t(\langle\beta,\boldsymbol{x}\rangle)(\langle\beta,\boldsymbol{x}\rangle)^{k-1}]\beta\beta^\top]$. Because of the definition of $\text{ie}(\sigma_*)$, $\mathbb{E}_{\boldsymbol{x},\beta,t}[yz^k\beta\boldsymbol{x}^\top] = 0$ when $k < Q - 1$. As we assume that $\mathbb{E}_t[c_Q] = \Theta(1)$, when $k = Q - 1$, we see that

$$\mathbb{E}_\beta[\mathbb{E}_{\boldsymbol{x},t}[\sigma_*'^t(\langle\beta,\boldsymbol{x}\rangle)(\langle\beta,\boldsymbol{x}\rangle)^k]\beta\beta^\top] \asymp \mathbb{E}_\beta[\mathbb{E}_{\boldsymbol{x}}[\mathbb{E}_t[c_Q]\text{He}_{Q-1}(\langle\beta,\boldsymbol{x}\rangle)\langle\beta,\boldsymbol{x}\rangle^{Q-1}]\beta\beta^\top]$$
$$\asymp (\rho\sqrt{d})^{-(Q-1)}\mathbb{E}_\beta[\beta\beta^\top],$$

and this is the main term that is proportional to $\mathbb{E}_\beta[\beta\beta^\top]$. This means that we can use the same argument as Nishikawa et al. (2025) with $\text{ie}(\sigma_*) = Q$. $\qquad\square$

Let $\kappa = \log^{-C_\kappa} d$. In other words, $\kappa$ satisfies $\kappa = \Theta(\log^{-C_\kappa} d)$ where $C_\kappa$ can be taken sufficiently large.

**Lemma 8.** *Suppose that the context length satisfies $N_1 = \tilde{\Omega}(r^{\text{ge}(\sigma_*)+1})$. Then, it holds with high probability that*

$$g(\Gamma^*, \boldsymbol{X}_{N_1}, \boldsymbol{y}_{N_1}, \boldsymbol{x}) = P_1' + P_2'\left(\frac{\langle\boldsymbol{x},\beta\rangle}{\sqrt{r}}\right)^{\text{ge}(\sigma_*)} + n_3,$$

*where $P_1' = o_d(1)$, $P_2' = \Theta_d((\log d)^{-C_{P_2}})$ and $n_3 = o_d(P_2' r^{-\text{ge}(\sigma_*)/2-1/2}\log^{-2\deg(\sigma_*)+2} d)$.*

*Proof.* When $N_1 = \tilde{\Omega}(r^{\mathrm{ge}(\sigma_*)+1})$, the (h.o.t.) in the proof of Proposition 11 in Nishikawa et al. (2025) can be evaluated as $o(\rho^{-1} P_2 r^{-\mathrm{ge}(\sigma_*)/2-1/2} \log^{-2\deg(\sigma_*)+2} d)$ by carefully examining the term. Using this fact, the same argument as the proof of Proposition 11 in Nishikawa et al. (2025) yields the result. $\square$

## C  GRADIENT CALCULATION

For the sake of simplicity, we write $\sigma_*^{\mathrm{test}}$ as $\sigma_*$. Let $\Gamma_u = \Gamma^* + uu^{\mathrm{T}}$, $\boldsymbol{x}_i^* = \sqrt{r}\Gamma^*\boldsymbol{x}_i$, $\beta^* = \sqrt{r}\Gamma^*\beta$ and $f_{\mathrm{IC}}(\boldsymbol{x}) = \sum_{j=1}^m a_j \sigma\left(v_j \frac{\sum_{i=1}^{N_1} y_i \mathrm{e}^{y_i/\rho} \mathrm{e}^{\boldsymbol{x}_i^{*\top}\Gamma_u\boldsymbol{x}/\rho}}{\sum_{i=1}^{N_1} \mathrm{e}^{y_i/\rho}\mathrm{e}^{\boldsymbol{x}_i^{*\top}\Gamma_u\boldsymbol{x}/\rho}} + b_j\right)$. Moreover, let us define $A_i, B_i$ as $\bar{\sigma}_*(z)\exp((\bar{\sigma}_*(z)) = \sum_{i\geq 0}\frac{A_i}{i!}z^i$ and $\exp((\bar{\sigma}_*(z)) = \sum_{i\geq 0}\frac{B_i}{i!}z^i$, where

$$\bar{\sigma}_*(z) := \begin{cases} \frac{\sigma_*(z)}{\rho} & \left(\text{if } \left|\frac{\sigma_*(z)}{\rho}\right| \leq \frac{1}{\log d}\right) \\ 0 \ (\text{otherwise}) \end{cases}.$$

**Lemma 9.** *Take $\rho = \Theta((\log d)^{C_\rho})$ where $C_\rho$ is a constant sufficiently large. Suppose that $\boldsymbol{u}$ satisfies $\|u\| = C_u \leq 1$ and $\langle \boldsymbol{u}, \boldsymbol{x}_i^* \rangle = C_u \tilde{O}_d(1)$. Then*

$$\sqrt{r}\Gamma^* \frac{\partial f_{\mathrm{IC}}}{\partial \boldsymbol{u}} = \alpha L_m \left\{ \sqrt{r}\Gamma^*\boldsymbol{x} \cdot (\gamma(\boldsymbol{x}, y)\langle\beta^*, \boldsymbol{u}\rangle + \sqrt{r}\Gamma^*\gamma(\boldsymbol{x}, y)\langle\boldsymbol{x}, \boldsymbol{u}\rangle\beta^* + \sqrt{r}\Gamma^*\boldsymbol{x}\cdot n_1 + \langle\boldsymbol{x}, \boldsymbol{u}\rangle\boldsymbol{n}_2 \right\},$$

*where*

$$\gamma(\boldsymbol{x}, y) = P_0 + P_1 z + \cdots + P_{\mathrm{ie}(\sigma)-1} z^{\mathrm{ie}(\sigma)-1} + \ldots$$

*is satisfied with $z = \langle\beta, \sqrt{r}\Gamma^{*\top}\Gamma_u\boldsymbol{x}\rangle/\rho$, and $n_1 = \tilde{O}\left(C_u\sqrt{\frac{1}{N_1}}\right)$ and $\boldsymbol{n}_2 = \tilde{O}\left(\sqrt{\frac{r}{N_1}}\right)$ hold.*

Note that from Lemma 29, the condition $\langle\boldsymbol{u}, \boldsymbol{x}_i^*\rangle = C_u\tilde{O}_d(1)$ is satisfied at the initialization.

*Proof.* Let

$$\pi_1(\boldsymbol{x}, y) = \frac{1}{N_1}\sum_{i=1}^{N_1} \frac{y_i}{\rho}\mathrm{e}^{y_i/\rho}\mathrm{e}^{\boldsymbol{x}_i^{*\top}\Gamma_u\boldsymbol{x}/\rho},$$

$$\pi_2(\boldsymbol{x}, y) = \frac{1}{N_1}\sum_{i=1}^{N_1} \mathrm{e}^{y_i/\rho}\mathrm{e}^{\boldsymbol{x}_i^{*\top}\Gamma_u\boldsymbol{x}/\rho},$$

$$\xi_1(\boldsymbol{x}, y) = \frac{1}{N_1}\sum_{i=1}^{N_1} \frac{y_i}{\rho}\mathrm{e}^{y_i/\rho}\mathrm{e}^{\boldsymbol{x}_i^{*\top}\Gamma_u\boldsymbol{x}/\rho}\{\langle\boldsymbol{x}_i^*, \boldsymbol{u}\rangle\boldsymbol{x} + \langle\boldsymbol{u}, \boldsymbol{x}\rangle\boldsymbol{x}_i^*\},$$

$$\xi_2(\boldsymbol{x}, y) = \frac{1}{N_1}\sum_{i=1}^{N_1} \mathrm{e}^{y_i/\rho}\mathrm{e}^{\boldsymbol{x}_i^{*\top}\Gamma_u\boldsymbol{x}/\rho}\{\langle\boldsymbol{x}_i^*, \boldsymbol{u}\rangle\boldsymbol{x} + \langle\boldsymbol{u}, \boldsymbol{x}\rangle\boldsymbol{x}_i^*\}.$$

Then, we can write the gradient of $f_{\mathrm{IC}}$ with respect to $\boldsymbol{u}$ as

$$\frac{\partial f_{\mathrm{IC}}}{\partial \boldsymbol{u}} = \sum_{j=1}^m a_j\sigma'\left(v_j\frac{\sum_{i=1}^{N_1} y_i\mathrm{e}^{y_i/\rho}\mathrm{e}^{\boldsymbol{x}_i^*\Gamma_u\boldsymbol{x}/\rho}}{\sum_{i=1}^{N_1}\mathrm{e}^{y_i/\rho}\mathrm{e}^{\boldsymbol{x}_i^*\Gamma_u\boldsymbol{x}/\rho}} + b_j\right)\cdot\frac{\xi_1\pi_2 - \xi_2\pi_1}{\pi_2^2}.$$

First, from the assumption $\langle\boldsymbol{u}, \boldsymbol{x}_i^*\rangle = C_u\tilde{O}_d(1)$ with $C_u \leq 1$ and Lemma 25, we can see that

$$\boldsymbol{x}_i^* uu^\top\boldsymbol{x} = \langle\boldsymbol{x}_i^*, \boldsymbol{u}\rangle\langle\boldsymbol{u}, \boldsymbol{x}\rangle = \tilde{O}_d(1).$$

Therefore, by taking $C_\rho$ sufficiently large, we can say $\boldsymbol{x}_i^{*\top}\Gamma_u\boldsymbol{x}/\rho = o_d(1)$ with high probability. Hence, following the same argument as Proposition 11 in Nishikawa et al. (2025), we can say that the content of $\rho'(\cdot)$ is $P_1(1 + o(1))$ using the positive constant $P_1$. Therefore, $\rho'(\cdot) = 1$ if and only if $v_j = 1$, which means that

$$\frac{\partial f_{\mathrm{IC}}}{\partial \boldsymbol{u}} = \alpha L_m\frac{\xi_1\pi_2 - \xi_2\pi_1}{\pi_2^2}.$$

Next, we evaluate the term $\frac{\xi_1\pi_2-\xi_2\pi_1}{\pi_2^2}$. For that purpose, we first derive the expectation of each component of this term, and then bound the deviation of its actual value from the expectation. Let $k_\rho = \frac{1}{2}(\exp(\tau/\rho) + \exp(-\tau/\rho))$, then the following holds:

$$\mathbb{E}_{\boldsymbol{x}_1,\zeta_1}[\mathrm{e}^{\bar{\sigma}_*(\langle\beta,\boldsymbol{x}_1\rangle)/\rho+\zeta_1/\rho}\mathrm{e}^{\boldsymbol{x}_1^{*\top}\Gamma_u\boldsymbol{x}/\rho}\boldsymbol{x}_1^*] = k_\rho\mathbb{E}_{\boldsymbol{x}_1}[\mathrm{e}^{\bar{\sigma}_*(\langle\beta,\boldsymbol{x}_1\rangle)}\mathrm{e}^{\boldsymbol{x}_1^{*\top}\Gamma_u\boldsymbol{x}/\rho}\boldsymbol{x}_1^*]$$

$$= k_\rho\mathbb{E}_{\boldsymbol{x}_1}\left[\sum_{i\geq 0}\frac{B_i}{i!}\mathrm{He}_i(\langle\beta,\boldsymbol{x}_1\rangle)\mathrm{e}^{\boldsymbol{x}_1^{*\top}\Gamma_u\boldsymbol{x}/\rho}\boldsymbol{x}_1^*\right]$$

$$= \sqrt{r}\Gamma^* k_\rho\beta\mathbb{E}_{\boldsymbol{x}_1}\left[\sum_{i\geq 0}\frac{B_{i+1}}{i!}\mathrm{He}_i(\langle\beta,\boldsymbol{x}_1\rangle)\mathrm{e}^{\boldsymbol{x}_1^{*\top}\Gamma_u\boldsymbol{x}/\rho}\right]$$

$$+ \sqrt{r}\Gamma^* k_\rho\frac{\Gamma_u\boldsymbol{x}}{\rho}\mathbb{E}_{\boldsymbol{x}_1}\left[\sum_{i\geq 0}\frac{B_i}{i!}\mathrm{He}_i(\langle\beta,\boldsymbol{x}_1\rangle)\mathrm{e}^{\boldsymbol{x}_1^{*\top}\Gamma_u\boldsymbol{x}/\rho}\right], \tag{1}$$

(recall that $z = \langle\beta, \sqrt{r}\Gamma^{*\top}\Gamma_u\boldsymbol{x}\rangle/\rho$). Here, we used Stein's lemma to derive the final equality. The expectation in the second term of the RHS can be calculated as

$$\mathbb{E}_{\boldsymbol{x}_1}\left[\sum_{i\geq 0}\frac{B_i}{i!}\mathrm{He}_i(\langle\beta,\boldsymbol{x}_1\rangle)\mathrm{e}^{\boldsymbol{x}_1^{*\top}\Gamma_u\boldsymbol{x}/\rho}\right]$$

$$= \mathbb{E}_{\boldsymbol{x}_1}\left[\sum_{i\geq 0}\frac{B_i}{i!}\mathrm{He}_i(\langle\beta,\boldsymbol{x}_1\rangle)\sum_{j\geq 0}\frac{1}{j!}\left(\frac{\|\sqrt{r}\Gamma^{*\top}\Gamma_u\boldsymbol{x}\|}{\rho}\right)^j\exp\left(\frac{\|\sqrt{r}\Gamma^{*\top}\Gamma_u\boldsymbol{x}\|^2}{2\rho^2}\right)\mathrm{He}_j\left(\left\langle\boldsymbol{x}_1,\frac{\sqrt{r}\Gamma^{*\top}\Gamma_u\boldsymbol{x}}{\|\sqrt{r}\Gamma^{*\top}\Gamma_u\boldsymbol{x}\|}\right\rangle\right)\right]$$

$$= \sum_{i\geq 0}\frac{B_i}{i!}\left(\frac{\|\sqrt{r}\Gamma^{*\top}\Gamma_u\boldsymbol{x}\|}{\rho}\right)^i\exp\left(\frac{\|\sqrt{r}\Gamma^{*\top}\Gamma_u\boldsymbol{x}\|^2}{2\rho^2}\right)\left\langle\beta,\frac{\sqrt{r}\Gamma^{*\top}\Gamma_u\boldsymbol{x}}{\|\sqrt{r}\Gamma^{*\top}\Gamma_u\boldsymbol{x}\|}\right\rangle^i$$

$$= \exp\left(\frac{\|\sqrt{r}\Gamma^{*\top}\Gamma_u\boldsymbol{x}\|^2}{2\rho^2}\right)\left(\sum_{i\geq 0}\frac{B_i}{i!}\langle\beta,\sqrt{r}\Gamma^{*\top}\Gamma_u\boldsymbol{x}/\rho\rangle^i\right).$$

The expectation in the first term of Eq. (1) can be evaluated in a similar way, and we obtain:

$$\mathbb{E}_{\boldsymbol{x}_1,\zeta_1}[\mathrm{e}^{\bar{\sigma}_*(\langle\boldsymbol{x}_1,\beta\rangle)/\rho+\zeta_1/\rho}\mathrm{e}^{\boldsymbol{x}_1^{*\top}\Gamma_u\boldsymbol{x}/\rho}\boldsymbol{x}_1^*]$$

$$= \sqrt{r}\Gamma^* k_\rho\beta\exp\left(\frac{\|\sqrt{r}\Gamma^{*\top}\Gamma_u\boldsymbol{x}\|^2}{2\rho^2}\right)\left(\sum_{i\geq 0}\frac{B_{i+1}}{i!}z^i\right)$$

$$+ \sqrt{r}\Gamma^*\frac{k_\rho}{\rho}\Gamma_u\boldsymbol{x}\exp\left(\frac{\|\sqrt{r}\Gamma^{*\top}\Gamma_u\boldsymbol{x}\|^2}{2\rho^2}\right)\left(\sum_{i\geq 0}\frac{B_i}{i!}z^i\right). \tag{2}$$

Similarly, using $k'_\rho := \frac{\tau}{2\rho}(\exp(\tau/\rho) - \exp(-\tau/\rho))$, we have that

$$\mathbb{E}_{\boldsymbol{x}_1,\zeta_1}\left[\frac{\bar{\sigma}_*(\langle\boldsymbol{x}_1,\beta\rangle)+\zeta_1}{\rho}\mathrm{e}^{\bar{\sigma}_*(\langle\boldsymbol{x}_1,\beta\rangle)/\rho+\zeta_1/\rho}\mathrm{e}^{\boldsymbol{x}_1^{*\top}\Gamma_u\boldsymbol{x}/\rho}\boldsymbol{x}_1^*\right]$$

$$= \sqrt{r}\Gamma^* k_\rho\exp\left(\frac{\|\sqrt{r}\Gamma^{*\top}\Gamma_u\boldsymbol{x}\|^2}{2\rho^2}\right)\left(\sum_{i\geq 0}\frac{A_{i+1}}{i!}z^i\right)\beta + \sqrt{r}\Gamma^*\frac{k_\rho}{\rho}\exp\left(\frac{\|\sqrt{r}\Gamma^{*\top}\Gamma_u\boldsymbol{x}\|^2}{2\rho^2}\right)\left(\sum_{i\geq 0}\frac{A_i}{i!}z^i\right)\Gamma_u\boldsymbol{x}$$

$$+ \sqrt{r}\Gamma^* k'_\rho\exp\left(\frac{\|\sqrt{r}\Gamma^{*\top}\Gamma_u\boldsymbol{x}\|^2}{2\rho^2}\right)\left(\sum_{i\geq 0}\frac{B_{i+1}}{i!}z^i\right)\beta + \sqrt{r}\Gamma^*\frac{k'_\rho}{\rho}\exp\left(\frac{\|\sqrt{r}\Gamma^{*\top}\Gamma_u\boldsymbol{x}\|^2}{2\rho^2}\right)\left(\sum_{i\geq 0}\frac{B_i}{i!}z^i\right)\Gamma_u\boldsymbol{x}.$$

Let $F_1 = \frac{1}{N_1}\sum_{i=1}^{N_1}\left[\frac{y_i}{\rho}\mathrm{e}^{y_i/\rho}\mathrm{e}^{\boldsymbol{x}_i^{*\top}\Gamma_u\boldsymbol{x}/\rho}\boldsymbol{x}_i^*\right]$ and $F_2 = \frac{1}{N_1}\sum_{i=1}^{N_1}[\mathrm{e}^{y_i/\rho}\mathrm{e}^{\boldsymbol{x}_i^{*\top}\Gamma_u\boldsymbol{x}/\rho}\boldsymbol{x}_i^*]$. Then, the expectation of $\xi_1$ and $\xi_2$ can be written as $\mathbb{E}_{\boldsymbol{x}_1,\zeta_1}[\xi_1] = \langle[\mathbb{E}(F_1)],\boldsymbol{u}\rangle\boldsymbol{x} + \langle\boldsymbol{u},\boldsymbol{x}\rangle\mathbb{E}[F_1]$ and $\mathbb{E}_{\boldsymbol{x}_1,\zeta_1}[\xi_2] = \langle\mathbb{E}[F_2],\boldsymbol{u}\rangle\boldsymbol{x} + \langle\boldsymbol{u},\boldsymbol{x}\rangle\mathbb{E}[F_2]$ respectively.

Based on the argument above, the term $\frac{\xi_1\pi_2-\xi_2\pi_1}{\pi_2^2}$ is concentrated on

$$(\mathbb{E}[\pi_2])^{-2}\left\{\langle\mathbb{E}[F_1]\mathbb{E}[\pi_2],\boldsymbol{u}\rangle\,\boldsymbol{x}+\langle\boldsymbol{u},\boldsymbol{x}\rangle\,\mathbb{E}[F_1]\mathbb{E}[\pi_2]-\langle\mathbb{E}[F_2]\mathbb{E}[\pi_1],\boldsymbol{u}\rangle\,\boldsymbol{x}-\langle\boldsymbol{u},\boldsymbol{x}\rangle\,\mathbb{E}[F_2]\mathbb{E}[\pi_1]\right\}.$$

Also, following the same argument to obtain Eq. (2), we have

$$\mathbb{E}_{\boldsymbol{x}_1,\zeta_1}\left[\mathrm{e}^{\bar\sigma_*(\langle\boldsymbol{x}_1,\beta\rangle)/\rho+\zeta_1/\rho}\mathrm{e}^{\boldsymbol{x}_1^{*\top}\Gamma_u\boldsymbol{x}/\rho}\right]=k_\rho\exp\left(\frac{\|\sqrt{r}\Gamma^{*\top}\Gamma_u\boldsymbol{x}\|^2}{2\rho^2}\right)\left(\sum_{i\geq0}\frac{B_i}{i!}z^i\right),$$

and

$$\mathbb{E}_{\boldsymbol{x}_1,\zeta_1}\left[\frac{\bar\sigma_*(\langle\boldsymbol{x}_1,\beta\rangle)+\zeta_1}{\rho}\mathrm{e}^{\bar\sigma_*(\langle\boldsymbol{x}_1,\beta\rangle)/\rho+\zeta_1/\rho}\mathrm{e}^{\boldsymbol{x}_1^{*\top}\Gamma_u\boldsymbol{x}/\rho}\right]$$

$$=k_\rho\exp\left(\frac{\|\sqrt{r}\Gamma^{*\top}\Gamma_u\boldsymbol{x}\|^2}{2\rho^2}\right)\left(\sum_{i\geq0}\frac{A_i}{i!}z^i\right)+k_\rho'\exp\left(\frac{\|\sqrt{r}\Gamma^{*\top}\Gamma_u\boldsymbol{x}\|^2}{2\rho^2}\right)\left(\sum_{i\geq0}\frac{B_i}{i!}z^i\right).$$

Using these result, we obtain that

$$(\mathbb{E}_{\boldsymbol{x}_1,\zeta_1}[\pi_2])^{-2}\left(\mathbb{E}_{x_1,\zeta_1}[F_1]\mathbb{E}_{x_1,\zeta_1}[\pi_2]-\mathbb{E}_{x_1,\zeta_1}[F_2]\mathbb{E}_{x_1,\zeta_1}[\pi_1]\right)$$

$$=\sqrt{r}\Gamma^*\left(k_\rho\sum_{i\geq0}\frac{B_i}{i!}z^i\right)^{-2}\times$$

$$\left\{\left[k_\rho\left(\sum_{i\geq0}\frac{A_{i+1}}{i!}z^i\right)\beta+\frac{k_\rho}{\rho}\left(\sum_{i\geq0}\frac{A_i}{i!}z^i\right)\Gamma_u\boldsymbol{x}+k_\rho'\left(\sum_{i\geq0}\frac{B_{i+1}}{i!}z^i\right)\beta+\frac{k_\rho'}{\rho}\left(\sum_{i\geq0}\frac{B_i}{i!}z^i\right)\Gamma_u\boldsymbol{x}\right]\cdot k_\rho\left(\sum_{i\geq0}\frac{B_i}{i!}z^i\right)\right.$$

$$\left.-\left[k_\rho\beta\left(\sum_{i\geq0}\frac{B_{i+1}}{i!}z^i\right)+\frac{k_\rho}{\rho}\Gamma_u\boldsymbol{x}\left(\sum_{i\geq0}\frac{B_i}{i!}z^i\right)\right]\cdot\left[k_\rho\left(\sum_{i\geq0}\frac{A_i}{i!}z^i\right)+k_\rho'\left(\sum_{i\geq0}\frac{B_i}{i!}z^i\right)\right]\right\}$$

$$=\sqrt{r}\Gamma^*\left(k_\rho\sum_{i\geq0}\frac{B_i}{i!}z^i\right)^{-2}\left\{\left[k_\rho\left(\sum_{i\geq0}\frac{A_{i+1}}{i!}z^i\right)\beta+k_\rho'\left(\sum_{i\geq0}\frac{B_{i+1}}{i!}z^i\right)\beta\right]\cdot k_\rho\left(\sum_{i\geq0}\frac{B_i}{i!}z^i\right)\right.$$

$$\left.-k_\rho\beta\left(\sum_{i\geq0}\frac{B_{i+1}}{i!}z^i\right)\cdot\left[k_\rho\left(\sum_{i\geq0}\frac{A_i}{i!}z^i\right)+k_\rho'\left(\sum_{i\geq0}\frac{B_i}{i!}z^i\right)\right]\right\}.$$

Therefore, we can expand this value using $z$. Specifically,

$$(\mathbb{E}_{x_1,\zeta_1}[\pi_2])^{-2}\left(\mathbb{E}_{x_1,\zeta_1}[F_1]\mathbb{E}_{x_1,\zeta_1}[\pi_2]-\mathbb{E}_{x_1,\zeta_1}[F_2]\mathbb{E}_{x_1,\zeta_1}[\pi_1]\right)=\sqrt{r}\Gamma^*\left(P_0+P_1z+\dots\right)\beta$$

holds. By letting $\gamma(\boldsymbol{x},y)=P_0+P_1z+\dots$, we obtain

$$\frac{\mathbb{E}[\xi_1]\mathbb{E}[\pi_2]-\mathbb{E}[\xi_2]\mathbb{E}[\pi_1]}{\mathbb{E}[\pi_2]^2}=\boldsymbol{x}\cdot\left(\gamma(\boldsymbol{x},y)\left\langle\sqrt{r}\Gamma^*\beta,\boldsymbol{u}\right\rangle\right)+\gamma(\boldsymbol{x},y)\left\langle\boldsymbol{x},\boldsymbol{u}\right\rangle\sqrt{r}\Gamma^*\beta.$$

Following the same argument as Lemma 20 in Nishikawa et al. (2025) yields the order of $P_i$.

Next we deal with the deviation from the expectation. Using the same technique as Lemma 18 in Nishikawa et al. (2025), we have

$$\frac{1}{N_1}\sum_{i=1}^{N_1}\frac{y_i}{\rho}\mathrm{e}^{y_i/\rho}\mathrm{e}^{\boldsymbol{x}_i^{*\top}\Gamma_u\boldsymbol{x}/\rho}=\mathbb{E}_{\boldsymbol{x}_1,\zeta_1}\left[\frac{\bar\sigma_*(\langle\boldsymbol{x}_1,\beta\rangle)+\zeta_1}{\rho}\mathrm{e}^{\bar\sigma_*(\langle\boldsymbol{x}_1,\beta\rangle)/\rho+\zeta_1/\rho}\mathrm{e}^{\boldsymbol{x}_1^{*\top}\Gamma_u\boldsymbol{x}/\rho}\right]+\tilde{O}(N_1^{-1/2}),$$

and

$$\frac{1}{N_1}\sum_{i=1}^{N_1}\mathrm{e}^{y_i/\rho}\mathrm{e}^{\boldsymbol{x}_i^{*\top}\Gamma_u\boldsymbol{x}/\rho}=\mathbb{E}_{\boldsymbol{x}_1,\zeta_1}\left[\mathrm{e}^{\bar\sigma_*(\langle\boldsymbol{x}_1,\beta\rangle)/\rho+\zeta_1/\rho}\mathrm{e}^{\boldsymbol{x}_1^{*\top}\Gamma_u\boldsymbol{x}/\rho}\right]+\tilde{O}(N_1^{-1/2}).$$

Moreover, noting that $\sqrt{r}\Gamma^*\boldsymbol{x}_i^* = \tilde{O}(\sqrt{r})$ (see Lemma 28), from Lemma 30, we have

$$\frac{1}{N_1}\sum_{i=1}^{N_1}\sqrt{r}\Gamma^*\frac{y_i}{\rho}\mathrm{e}^{y_i/\rho}\mathrm{e}^{\boldsymbol{x}_i^{*\top}\Gamma_u\boldsymbol{x}/\rho}\boldsymbol{x}_i^*$$

$$=\sqrt{r}\Gamma^*\mathbb{E}_{\boldsymbol{x}_1,\zeta_1}\left[\frac{\bar{\sigma}_*(\langle\boldsymbol{x}_1,\beta\rangle)+\zeta_1}{\rho}\mathrm{e}^{\bar{\sigma}_*(\langle\boldsymbol{x}_1,\beta\rangle)/\rho+\zeta_1/\rho}\mathrm{e}^{\boldsymbol{x}_1^{*\top}\Gamma_u\boldsymbol{x}/\rho}\boldsymbol{x}_1^*\right]+\tilde{O}(r^{1/2}N_1^{-1/2}),$$

and

$$\frac{1}{N_1}\sum_{i=1}^{N_1}\sqrt{r}\Gamma^*\mathrm{e}^{y_i/\rho}\mathrm{e}^{\boldsymbol{x}_i^{*\top}\Gamma_u\boldsymbol{x}/\rho}\boldsymbol{x}_i^* = \sqrt{r}\Gamma^*\mathbb{E}_{\boldsymbol{x}_1,\zeta_1}\left[\mathrm{e}^{\bar{\sigma}_*(\langle\boldsymbol{x}_1,\beta\rangle)/\rho+\zeta_1/\rho}\mathrm{e}^{\boldsymbol{x}_1^{*\top}\Gamma_u\boldsymbol{x}/\rho}\boldsymbol{x}_1^*\right]+\tilde{O}(r^{1/2}N_1^{-1/2}).$$

Hence, we obtain that

$$\sqrt{r}\Gamma^*\pi_2^{-2}\left(F_1\pi_2-F_2\pi_1\right)=\frac{\mathrm{poly}(z)\beta+\delta_2}{\mathrm{poly}(z)+\delta_1},$$

for some $\delta_1$ and $\delta_2$ which satisfy $\delta_1=\tilde{O}(1/\sqrt{N_1})$ and $\delta_2=\tilde{O}\left(\sqrt{\frac{r}{N_1}}\right)$ with high probability. Let $\boldsymbol{n}_2=\sqrt{r}\Gamma^*\{\pi_2^{-2}\left(F_1\pi_2-F_2\pi_1\right)-\gamma(\boldsymbol{x},y)\beta\}$, then $\boldsymbol{n}_2=\tilde{O}(\sqrt{\frac{r}{N_1}})$ holds.

Also, noting that $\langle\boldsymbol{x}_i^*,\boldsymbol{u}\rangle=C_u\tilde{O}_d(1)$ with high probability, it holds that

$$\frac{1}{N_1}\sum_{i=1}^{N_1}\frac{y_i}{\rho}\mathrm{e}^{y_i/\rho}\mathrm{e}^{\boldsymbol{x}_i^{*\top}\Gamma_u\boldsymbol{x}/\rho}\langle\boldsymbol{x}_i^*,\boldsymbol{u}\rangle$$

$$=\mathbb{E}_{\boldsymbol{x}_1,\zeta_1}\left[\frac{\bar{\sigma}_*(\langle\boldsymbol{x}_1,\beta\rangle)+\zeta_1}{\rho}\mathrm{e}^{\bar{\sigma}_*(\langle\boldsymbol{x}_1,\beta\rangle)/\rho+\zeta_1/\rho}\mathrm{e}^{\boldsymbol{x}_1^{*\top}\Gamma_u\boldsymbol{x}/\rho}\langle\boldsymbol{x}_1^*,\boldsymbol{u}\rangle\right]+\tilde{O}(C_uN_1^{-1/2}),$$

and

$$\frac{1}{N_1}\sum_{i=1}^{N_1}\mathrm{e}^{y_i/\rho}\mathrm{e}^{\boldsymbol{x}_i^{*\top}\Gamma_u\boldsymbol{x}/\rho}\langle\boldsymbol{x}_i^*,\boldsymbol{u}\rangle = \mathbb{E}_{\boldsymbol{x}_1,\zeta_1}\left[\mathrm{e}^{\bar{\sigma}_*(\langle\boldsymbol{x}_1,\beta\rangle)/\rho+\zeta_1/\rho}\mathrm{e}^{\boldsymbol{x}_1^{*\top}\Gamma_u\boldsymbol{x}/\rho}\langle\boldsymbol{x}_1^*,\boldsymbol{u}\rangle\right]+\tilde{O}(C_uN_1^{-1/2}).$$

Therefore, letting $n_1=\pi_2^{-2}\{\langle F_1,\boldsymbol{u}\rangle\pi_2-\langle F_2,\boldsymbol{u}\rangle\pi_1\}-\gamma(\boldsymbol{x},y)\langle\beta^*,\boldsymbol{u}\rangle$, we obtain that $n_1=\tilde{O}(C_u\sqrt{\frac{1}{N_1}})$.

In summary, we obtain that

$$\sqrt{r}\Gamma_*\frac{\xi_1\pi_2-\xi_2\pi_1}{\pi_2^2}=\sqrt{r}\Gamma_*[\boldsymbol{x}\cdot(\gamma(\boldsymbol{x},y)\langle\beta^*,\boldsymbol{u}\rangle)+\gamma(\boldsymbol{x},y)\langle\boldsymbol{x},\boldsymbol{u}\rangle\beta^*]+\sqrt{r}\Gamma^*\boldsymbol{x}\cdot n_1+\langle\boldsymbol{x},\boldsymbol{u}\rangle\boldsymbol{n}_2,$$

where $n_1=\tilde{O}\left(C_u\sqrt{\frac{1}{N_1}}\right)$ and $\boldsymbol{n}_2=\tilde{O}\left(\sqrt{\frac{r}{N_1}}\right)$. $\qquad\square$

**Lemma 10.** *The norm of the vector $\|\sqrt{r}\Gamma^*\frac{\partial f_{\mathrm{IC}}}{\partial\boldsymbol{u}}\|$ has sub-Weibull tail with tail index $1/(P+2)$.*

*Proof.* It suffices to show that $\left\|\sqrt{r}\Gamma_*\frac{\xi_1\pi_2-\xi_2\pi_1}{\pi_2^2}\right\|$ has sub-Weibull tail. Note that $\frac{\xi_1\pi_2-\xi_2\pi_1}{\pi_2^2}=\frac{\xi_1}{\pi_2}-\frac{\xi_2}{\pi_2}\frac{\pi_1}{\pi_2}$. Therefore we examine $\frac{\xi_1}{\pi_2},\frac{\xi_2}{\pi_2}$ and $\frac{\pi_1}{\pi_2}$. Let us define

$$p_i=\frac{\mathrm{e}^{\boldsymbol{x}_i^{*\top}\Gamma_u\boldsymbol{x}/\rho}}{\sum_{i=1}^{N_1}\mathrm{e}^{y_i/\rho}\mathrm{e}^{\boldsymbol{x}_i^{*\top}\Gamma_u\boldsymbol{x}/\rho}},$$

then the following holds:

$$\frac{\xi_1}{\pi_2}=\sum_{i=1}^{N_1}p_i\frac{y_i}{\rho}\{\langle\boldsymbol{x}_i^*,\boldsymbol{u}\rangle\boldsymbol{x}+\langle\boldsymbol{u},\boldsymbol{x}\rangle\boldsymbol{x}_i^*\}$$

$$\frac{\xi_2}{\pi_2}=\sum_{i=1}^{N_1}p_i\{\langle\boldsymbol{x}_i^*,\boldsymbol{u}\rangle\boldsymbol{x}+\langle\boldsymbol{u},\boldsymbol{x}\rangle\boldsymbol{x}_i^*\}$$

$$\frac{\pi_1}{\pi_2} = \sum_{i=1}^{N_1} p_i \frac{y_i}{\rho}$$

By the triangle inequality, we have

$$\left\| \frac{\xi_1}{\pi_2} \right\| \le \sum_{i=1}^{N_1} p_i \frac{y_i}{\rho} \| \langle \boldsymbol{x}_i^*, \boldsymbol{u} \rangle \, \boldsymbol{x} + \langle \boldsymbol{u}, \boldsymbol{x} \rangle \, \boldsymbol{x}_i^* \|.$$

By further applying the Cauchy-Schwarz inequality, we have

$$\left\| \frac{\xi_1}{\pi_2} \right\| \le \left( \sum_{i=1}^{N_1} p_i^2 \right) \left( \sum_{i=1}^{N_1} \frac{y_i^2}{\rho^2} \| \langle \boldsymbol{x}_i^*, \boldsymbol{u} \rangle \, \boldsymbol{x} + \langle \boldsymbol{u}, \boldsymbol{x} \rangle \, \boldsymbol{x}_i^* \|^2 \right).$$

Note that $\sum_{i=1}^{N_1} p_i^2 \le 1$. Next, we can see that

$$y_i^2 \| \langle \boldsymbol{x}_i^*, \boldsymbol{u} \rangle \, \boldsymbol{x} + \langle \boldsymbol{u}, \boldsymbol{x} \rangle \, \boldsymbol{x}_i^* \|^2 \le 2 y_i^2 \| \langle \boldsymbol{x}_i^*, \boldsymbol{u} \rangle \, \boldsymbol{x} \|^2 + 2 y_i^2 \| \langle \boldsymbol{u}, \boldsymbol{x} \rangle \, \boldsymbol{x}_i^* \|^2$$

Since $y_i = \sigma_*(\langle \beta, \boldsymbol{x} \rangle) + \zeta_i$, the tail probability of each term is almost equal to that of $\mathrm{poly}(t)$ where $t \sim \mathcal{N}(0, 1)$, which means $y_i^2 \| \langle \boldsymbol{x}_i^*, \boldsymbol{u} \rangle \, \boldsymbol{x} \|^2$ and $y_i^2 \| \langle \boldsymbol{u}, \boldsymbol{x} \rangle \, \boldsymbol{x}_i^* \|^2$ have sub-Weibull tail with tail index $2/(2P+4)$. This means that $\left\| \frac{\xi_1}{\pi_2} \right\|$ has sub-Weibull tail with tail index $1/(P+2)$. By applying the same argument, you can see that $\frac{\xi_2}{\pi_2} \frac{\pi_1}{\pi_2}$ has sub-Weibull tail. Therefore, considering that $\| \sqrt{r} \Gamma^* \|_2$ is a finite constant, $\| \sqrt{r} \Gamma^* \frac{\partial f_{\mathrm{IC}}}{\partial \boldsymbol{u}} \|$ has sub-Weibull tail with tail index $1/(P+2)$. $\quad\square$

**Lemma 6** (Formal). *It holds that*

$$\frac{1}{2} \sqrt{r} \Gamma^* \nabla_{\boldsymbol{u}} (f_{\mathrm{IC}}(\boldsymbol{x}) - y)^2$$

$$= \Theta(\alpha m) \langle \beta, \boldsymbol{u} \rangle \frac{\beta}{\kappa^{\mathrm{ie}(\sigma_*)+1} \rho^{\mathrm{ie}(\sigma_*)}} \{ (\kappa \sqrt{r})^{-(\mathrm{ie}(\sigma_*)-1)} (1 + O(1/\sqrt{d})) + \langle \beta, \boldsymbol{u} \rangle^{2\mathrm{ie}(\sigma_*)-2} (1 + O(1/\sqrt{d})) \}$$

$$+ \tilde{O}(\alpha^2 m^2 C_u \sqrt{r}) + \tilde{O}\left( \alpha m C_u \sqrt{\frac{r}{N_1}} \right) + \nu,$$

*with high probability, where $\nu$ satisfies $\nu = \tilde{O}(\alpha m C_u \sqrt{r})$ with high probability and $\mathbb{E}_{\boldsymbol{x}}[\nu] = 0$. Moreover, $\|\nu\|$ has sub-Weibull tail.*

*Proof.* Note that

$$\frac{1}{2} \sqrt{r} \Gamma^* \nabla_{\boldsymbol{u}} (f_{\mathrm{IC}}(\boldsymbol{x}) - y)^2 = f_{\mathrm{IC}}(\boldsymbol{x}) \cdot \sqrt{r} \Gamma^* \nabla_{\boldsymbol{u}} f_{\mathrm{IC}}(\boldsymbol{x}) - y \cdot \sqrt{r} \Gamma^* \nabla_{\boldsymbol{u}} f_{\mathrm{IC}}(\boldsymbol{x}). \qquad (3)$$

First, we analyze the first term of the RHS. Since $y_i = \tilde{O}_p(1)$, we have that

$$|f_{\mathrm{IC}}(\boldsymbol{x})| \le \alpha m \left| \frac{\sum_{i=1}^{N_1} y_i \mathrm{e}^{y_i/\rho} \mathrm{e}^{\boldsymbol{x}_i^* \Gamma_u \boldsymbol{x}/\rho}}{\sum_{i=1}^{N_1} \mathrm{e}^{y_i/\rho} \mathrm{e}^{\boldsymbol{x}_i^* \Gamma_u \boldsymbol{x}/\rho}} \right| = \tilde{O}_p(\alpha m).$$

Moreover, from Lemma 9, it holds that

$$\sqrt{r} \Gamma^* \frac{\partial f_{\mathrm{IC}}}{\partial \boldsymbol{u}} = \alpha L_m \left\{ \sqrt{r} \Gamma^* \boldsymbol{x} \cdot (\gamma(\boldsymbol{x}, y) \langle \beta^*, \boldsymbol{u} \rangle + \sqrt{r} \Gamma^* \gamma(\boldsymbol{x}, y) \langle \boldsymbol{x}, \boldsymbol{u} \rangle \beta^* + \sqrt{r} \Gamma^* \boldsymbol{x} \cdot n_1 + \langle \boldsymbol{x}, \boldsymbol{u} \rangle n_2 \right\},$$

where $\sqrt{r} \Gamma^* \boldsymbol{x} = \tilde{O}(\sqrt{r})$ (See Lemma 28) and $\gamma(\boldsymbol{x}, y) = \tilde{O}(1)$ with high probability, which means that $\sqrt{r} \Gamma^* \frac{\partial f_{\mathrm{IC}}}{\partial \boldsymbol{u}} = \tilde{O}_p(\alpha m C_u \sqrt{r})$. Therefore, we obtain

$$f_{\mathrm{IC}}(\boldsymbol{x}) \cdot \sqrt{r} \Gamma^* \nabla_{\boldsymbol{u}} f_{\mathrm{IC}}(\boldsymbol{x}) = \tilde{O}_p(\alpha^2 m^2 C_u \sqrt{r}).$$

Next, we evaluate the second term of Eq. (3). For that purpose, let

$$\nu = y \cdot \sqrt{r} \Gamma^* \nabla_{\boldsymbol{u}} f_{\mathrm{IC}}(\boldsymbol{x}) - \mathbb{E}_{\boldsymbol{x}}[y \cdot \sqrt{r} \Gamma^* \nabla_{\boldsymbol{u}} f_{\mathrm{IC}}(\boldsymbol{x})].$$

By noticing that $y = \tilde{O}_p(1)$ and $\sqrt{r} \Gamma^* \nabla_{\boldsymbol{u}} f_{\mathrm{IC}}(\boldsymbol{x}) = \tilde{O}(\alpha m C_u \sqrt{r})$ with high probaibility, we have

$$\nu = \tilde{O}_p(\alpha m C_u \sqrt{r}).$$

Moreover, $\mathbb{E}_{\boldsymbol{x}}[\nu] = 0$ by definition, and $\|\nu\|$ has sub-Weibull tail from Lemma 10. Thus, what remains is to calculate $\mathbb{E}_{\boldsymbol{x}}[y \cdot \sqrt{r}\Gamma^* \nabla_{\boldsymbol{u}} f_{\mathrm{IC}}(\boldsymbol{x})]$:

$$\mathbb{E}_{\boldsymbol{x}}[y\alpha L_m \left\{ \sqrt{r}\Gamma^*\boldsymbol{x} \cdot \gamma(\boldsymbol{x}, y) \langle \beta^*, \boldsymbol{u}\rangle + \sqrt{r}\Gamma^*\gamma(\boldsymbol{x}, y) \langle \boldsymbol{x}, \boldsymbol{u}\rangle \beta^* \right\}]$$

$$= \alpha L_m \langle \beta^*, \boldsymbol{u}\rangle \mathbb{E}_{\boldsymbol{x}}[\sqrt{r}\Gamma^*\boldsymbol{x}\sigma_*(\langle\beta, \boldsymbol{x}\rangle)(P_0 + P_1 z + \dots)]$$

$$+ \alpha L_m \sqrt{r}\Gamma^*\beta^* \mathbb{E}_{\boldsymbol{x}}[\langle \boldsymbol{u}, \boldsymbol{x} \cdot \sigma_*(\langle\beta, \boldsymbol{x}\rangle)(P_0 + P_1 z + \dots)\rangle]. \tag{4}$$

Now, we calculate $\mathbb{E}_{\boldsymbol{x}}[\sqrt{r}\Gamma^*\boldsymbol{x}\sigma_*(\langle\beta, \boldsymbol{x}\rangle)z^k]$ in order to investigate $\mathbb{E}_{\boldsymbol{x}}[\sqrt{r}\Gamma^*\boldsymbol{x}\sigma_*(\langle\beta, \boldsymbol{x}\rangle)(P_0 + P_1 z + \dots)]$:

$$\mathbb{E}_{\boldsymbol{x}}[\sqrt{r}\Gamma^*\boldsymbol{x}\sigma_*(\langle\beta, \boldsymbol{x}\rangle)z^k]$$

$$= \sqrt{r}\Gamma^* \left\{ \mathbb{E}_{\boldsymbol{x}}[\sigma'_*(\langle\beta, \boldsymbol{x}\rangle)z^k]\beta + \mathbb{E}_{\boldsymbol{x}}[\sigma_*(\langle\beta, \boldsymbol{x}\rangle kz^{k-1})\Gamma_u^\top\beta] \right\} \tag{5}$$

holds from Stein's lemma, and when $k < \mathrm{ie}(\sigma_*) - 1$, $\mathbb{E}_{\boldsymbol{x}}[\sigma'_*(\langle\beta, \boldsymbol{x}\rangle)z^k] = 0$ from the definition of the information exponent. When $k = \mathrm{ie}(\sigma_*) - 1$, we have that

$$\sqrt{r}\Gamma^* \mathbb{E}_{\boldsymbol{x}}[\sigma'_*(\langle\beta, \boldsymbol{x}\rangle)z^k] \asymp \sqrt{r}\Gamma^*\beta \left( \frac{\langle\beta, \sqrt{r}\Gamma_u^\top\Gamma^*\beta\rangle}{\rho} \right)^{\mathrm{ie}(\sigma_*)-1},$$

and from Lemma 31, it holds that

$$\beta^* = \sqrt{r}\Gamma^*\beta = \kappa^{-1}(\beta + O(1/\sqrt{d})),$$

and

$$\langle\beta, \sqrt{r}\Gamma_u^\top\Gamma^*\beta\rangle = \langle\Gamma_u\beta, \sqrt{r}\Gamma_*\beta\rangle$$

$$= \langle\Gamma^*\beta, \sqrt{r}\Gamma_*\beta\rangle + \langle\beta, \boldsymbol{u}\rangle \langle\boldsymbol{u}, \sqrt{r}\Gamma^*\beta\rangle$$

$$= (\sqrt{r}\kappa^2)^{-1}(1 + O(1/\sqrt{d})) + \kappa^{-1}\langle\beta, \boldsymbol{u}\rangle \left\{ \langle\beta, \boldsymbol{u}\rangle + O(1/\sqrt{d}) \right\}$$

$$= (\sqrt{r}\kappa^2)^{-1}(1 + O(1/\sqrt{d})) + \kappa^{-1}\langle\beta, \boldsymbol{u}\rangle^2 \left\{ 1 + O(1/\sqrt{d}) \right\}$$

with high probability. By ignoring small terms, this yields that

$$\sqrt{r}\Gamma^*\beta \left( \frac{\langle\beta, \sqrt{r}\Gamma_u^\top\Gamma_*\beta\rangle}{\rho} \right)^{\mathrm{ie}(\sigma_*)-1}$$

$$= \frac{P_{\mathrm{ie}(\sigma_*)-1}\beta}{\kappa^{\mathrm{ie}(\sigma_*)}\rho^{\mathrm{ie}(\sigma_*)-1}} \left\{ (\kappa\sqrt{r})^{-(\mathrm{ie}(\sigma_*)-1)}(1 + O(1/\sqrt{d})) + \langle\beta, \boldsymbol{u}\rangle^{2\mathrm{ie}(\sigma_*)-2}(1 + O(1/\sqrt{d})) \right\}.$$

We can see that the other term in Eq. (5) is negligible. Using $P_{\mathrm{ie}(\sigma_*)-1} = \Theta(1/\rho)$, plugging this result into Eq. (4) yields

$$\mathbb{E}_{\boldsymbol{x}}[y\alpha L_m \left\{ \sqrt{r}\Gamma^*\boldsymbol{x} \cdot \gamma(\boldsymbol{x}, y) \langle\beta^*, \boldsymbol{u}\rangle + \sqrt{r}\Gamma^*\gamma(\boldsymbol{x}, y) \langle\boldsymbol{x}, \boldsymbol{u}\rangle \beta^* \right\}]$$

$$= \alpha L_m \langle\beta, \boldsymbol{u}\rangle \frac{\beta}{\kappa^{\mathrm{ie}(\sigma_*)+1}\rho^{\mathrm{ie}(\sigma_*)}} \left\{ (\kappa\sqrt{r})^{-(\mathrm{ie}(\sigma_*)-1)}(1 + O(1/\sqrt{d})) + \langle\beta, \boldsymbol{u}\rangle^{2\mathrm{ie}(\sigma_*)-2}(1 + O(1/\sqrt{d})) \right\}$$

with high probability. Finally, we evaluate the term $\mathbb{E}_{\boldsymbol{x}}[\alpha L_m \sigma_*(\langle\beta, \boldsymbol{x}\rangle)(\sqrt{r}\Gamma^*\boldsymbol{x} \cdot n_1 + \langle\boldsymbol{x}, \boldsymbol{u}\rangle n_2)]$. From Corollary 17 of Oko et al. (2024), we have $\sigma_*(\langle\beta, \boldsymbol{x}\rangle) = O_p(\log^{\deg(\sigma_*)/2} d)$. Moreover, from Lemma 9, it holds that $n_1 = \tilde{O}(C_u\sqrt{\frac{1}{N_1}})$ and $\boldsymbol{n}_2 = \tilde{O}(\sqrt{\frac{r}{N_1}})$ with high probability. Therefore, we arrive at

$$\mathbb{E}_{\boldsymbol{x}}[\alpha L_m \sigma_*(\langle\beta, \boldsymbol{x}\rangle)(\sqrt{r}\Gamma^*\boldsymbol{x} \cdot n_1 + \langle\boldsymbol{x}, \boldsymbol{u}\rangle \boldsymbol{n}_2)] = \tilde{O}_p \left( \alpha m C_u \sqrt{\frac{r}{N_1}} \right).$$

Combining all results above completes the proof. $\qquad\square$

The following lemma will be useful in the analysis of weak recovery (the next section).

**Lemma 11.** *When $\mathrm{ge}(\sigma_*) = 2$, $\gamma(\boldsymbol{x}, y) = \tilde{O}(1/\sqrt{r})$ holds with high probability.*

*Proof.* Recall that

$$\gamma(\boldsymbol{x}, y) = P_0 + P_1 z + \dots$$

$$= \left(k_\rho \sum_{i\geq0} \frac{B_i}{i!} z^i\right)^{-2} \left\{\left[k_\rho\left(\sum_{i\geq0}\frac{A_{i+1}}{i!}z^i\right) + k'_\rho\left(\sum_{i\geq0}\frac{B_{i+1}}{i!}z^i\right)\right] \cdot k_\rho\left(\sum_{i\geq0}\frac{B_i}{i!}z^i\right)\right.$$

$$\left. -k_\rho\left(\sum_{i\geq0}\frac{B_{i+1}}{i!}z^i\right) \cdot \left[k_\rho\left(\sum_{i\geq0}\frac{A_i}{i!}z^i\right) + k'_\rho\left(\sum_{i\geq0}\frac{B_i}{i!}z^i\right)\right]\right\}.$$

By expanding the RHS in powers of $z$ and comparing their coefficients, we obtain

$$P_0 = \frac{(k_\rho A_0 + k'_\rho B_0) \cdot k_\rho B_1 - k_\rho B_0(k_\rho A_1 + k'_\rho B_1)}{(k_\rho B_0)^2}$$

$$= \frac{A_0 B_1 - B_0 A_1}{B_0^2}.$$

When $\mathrm{ge}(\sigma_*)$ is 2, $\sigma_*$ is even, and thus $\exp(\bar{\sigma}_*)$ and $\bar{\sigma}_* \exp(\bar{\sigma}_*)$ are also even. This means that $\exp(\bar{\sigma}_*)$ and $\bar{\sigma}_* \exp(\bar{\sigma}_*)$ can be expanded in polynomial of $z^2$. This means that the coefficients of $z$ in the hermite expansion of $\exp(\bar{\sigma}_*)$ and $\bar{\sigma}_* \exp(\bar{\sigma}_*)$ are 0, namely $A_1 = B_1 = 0$, which yields $P_0 = 0$. Hence, we have

$$\gamma(\boldsymbol{x}, y) = P_1 z + P_2 z^2 + \dots,$$

where $P_1 = \tilde{O}(1)$ and $z = \tilde{O}_p(1/\sqrt{r})$, which yields the assertion. $\qquad\square$

## D  ONE STEP GRADIENT DESCENT FOR WEAK RECOVERY

**Lemma 12.** *Set $C_u = 1/\sqrt{r}$, $N_1 = \tilde{\Omega}(r^{\mathrm{ge}(\sigma_*)+1})$ and $N_{new} = \tilde{\Omega}(r^{\mathrm{ge}(\sigma_*)+2})$. Let $w_i \overset{i.i.d.}{\sim} \mathcal{N}(0, \boldsymbol{I}_d)$ for $i = 1, 2, \dots, N_{new}$ and $\boldsymbol{h} = \frac{1}{2}\sqrt{r}\Gamma^* \frac{1}{N_{new}}\sum_{i=1}^{N_{new}} \nabla_{\boldsymbol{u}}(f_{\mathrm{IC}}(\boldsymbol{w}_i) - (g(\Gamma^*, \boldsymbol{w}_i) - b))^2$, then*

$$\boldsymbol{h} = P_2'\Theta(\alpha_1 m)\langle\beta, \boldsymbol{u}\rangle\frac{\beta}{\kappa^{2\mathrm{ge}(\sigma_*)}\rho^{\mathrm{ge}(\sigma_*)}}(\sqrt{r})^{-(2\mathrm{ge}(\sigma_*)-1)}(1 + O(1/\sqrt{d}))$$

$$+ \tilde{O}(\alpha_1^2 m^2 r^{-1/2} r^{-\frac{\mathrm{ge}(\sigma_*)-1}{2}}) + \tilde{O}\left(\alpha_1 m\sqrt{\frac{r^{-\mathrm{ge}(\sigma_*)}}{N_1}}\right) + \tilde{O}\left(\alpha_1 m\sqrt{\frac{r^{-\mathrm{ge}(\sigma_*)}}{N_{new}}}\right) + \tilde{O}(\alpha_1 m r^{-\mathrm{ge}(\sigma_*)-1/2})$$

*holds with high probability.*

*Proof.* From Lemma 8, we have that

$$g(\Gamma^*, \boldsymbol{x}) = P_1' + P_2'\left(\frac{\langle\boldsymbol{x}, \beta\rangle}{\sqrt{r}}\right)^{\mathrm{ge}(\sigma_*)} + n_3,$$

where $P_1' = o_d(1)$, $P_2' = \Theta_d((\log d)^{-C_{P_2}})$ and $n_3 = o_d(P_2' r^{-\mathrm{ge}(\sigma_*)/2-1/2}\log^{-2\deg(\sigma_*)+2} d)$. Noticing that $g(\Gamma^*, \boldsymbol{x}) = O(1)$ with high probability, then, letting $b = \frac{1}{N_{new}}\sum_{i=1}^{N_{new}} g(\Gamma^*, \boldsymbol{w}_i)$ and applying Lemma 26, it holds that

$$g(\Gamma^*, \boldsymbol{w}_i) - b = P_2' r^{-\frac{\mathrm{ge}(\sigma_*)}{2}}\mathrm{He}_{\mathrm{ge}(\sigma_*)}(\langle\beta, \boldsymbol{w}_i\rangle) + n_3 + \tilde{O}(1/\sqrt{N_{new}}).$$

When $N_{new} = O(r^{\mathrm{ge}(\sigma_*)+2})$, the term $\tilde{O}(1/\sqrt{N_{new}})$ can be dominated by $n_3$, which means that

$$g(\Gamma^*, \boldsymbol{w}_i) - b = P_2' r^{-\frac{\mathrm{ge}(\sigma_*)}{2}}\mathrm{He}_{\mathrm{ge}(\sigma_*)}(\langle\beta, \boldsymbol{w}_i\rangle) + n_3,$$

and $n_3 = o_d(P_2' r^{-\mathrm{ge}(\sigma_*)/2-1/2}\log^{-2\deg(\sigma_*)+2} d)$ with high probability.

In this stage, we use $g(\Gamma^*, \boldsymbol{w}_i) - b$ as a teacher signal. Thus, as $C_u = 1/\sqrt{r}$ and $g(\Gamma^*, \boldsymbol{w}_i) - b = \tilde{O}(r^{-\frac{\mathrm{ge}(\sigma_*)}{2}})$ , with high probability, we have

$$\frac{1}{2}\sqrt{r}\Gamma^*\nabla_{\boldsymbol{u}}(f_{\mathrm{IC}}(\boldsymbol{w}_i) - (g(\Gamma^*, \boldsymbol{w}_i) - b))^2$$

$$= \tilde{O}(\alpha_1^2 m^2 r^{-1/2} r^{-\frac{\mathrm{ge}(\sigma_*)-1}{2}}) - \left\{P_2' r^{-\frac{\mathrm{ge}(\sigma_*)}{2}}\mathrm{He}_{\mathrm{ge}(\sigma_*)}(\langle\beta, \boldsymbol{w}_i\rangle) + n_3\right\} \cdot \sqrt{r}\Gamma^*\nabla_{\boldsymbol{u}}f_{\mathrm{IC}}(\boldsymbol{w}_i).$$

Following the same argument as Lemma 6 yields

$$P_2' r^{-\frac{\mathrm{ge}(\sigma_*)}{2}} \mathrm{He}_{\mathrm{ge}(\sigma_*)}(\langle \beta, \boldsymbol{w}_i \rangle) \cdot \sqrt{r} \Gamma^* \nabla_{\boldsymbol{u}} f_{\mathrm{IC}}(\boldsymbol{w}_i)$$

$$= P_2' \Theta(\alpha_1 m) \langle \beta, \boldsymbol{u} \rangle \frac{r^{-\frac{\mathrm{ge}(\sigma_*)}{2}} \beta}{\kappa^{\mathrm{ge}(\sigma_*)+1} \rho^{\mathrm{ge}(\sigma_*)}} \{ (\kappa \sqrt{r})^{-(\mathrm{ge}(\sigma_*)-1)} (1 + O(1/\sqrt{d})) + \langle \beta, \boldsymbol{u} \rangle^{2\mathrm{ge}(\sigma_*)-2} (1 + O(1/\sqrt{d})) \}$$

$$+ \nu + \tilde{O}\left( \alpha_1 m \sqrt{\frac{r^{-\mathrm{ge}(\sigma_*)}}{N_1}} \right)$$

with high probability. Since $\langle \beta, \boldsymbol{u} \rangle = \tilde{O}_p(1/r)$ at the initialization, the first term (regarding $(\kappa\sqrt{r})^{-(\mathrm{ge}(\sigma_*)-1)}$) dominates the second term (regarding $\langle \beta, \boldsymbol{u} \rangle^{2\mathrm{ge}(\sigma_*)-2}$). Taking the average over $i = 1, \ldots, N_{new}$ yields that

$$P_2' r^{-\frac{\mathrm{ge}(\sigma_*)}{2}} \frac{1}{N_{new}} \sum_{i=1}^{N_{new}} \mathrm{He}_{\mathrm{ge}(\sigma_*)}(\langle \beta, \boldsymbol{w}_i \rangle) \cdot \sqrt{r} \Gamma^* \nabla_{\boldsymbol{u}} f_{\mathrm{IC}}(\boldsymbol{w}_i)$$

$$= P_2' \Theta(\alpha_1 m) \langle \beta, \boldsymbol{u} \rangle \frac{\beta}{\kappa^{2\mathrm{ge}(\sigma_*)} \rho^{\mathrm{ge}(\sigma_*)}} (\sqrt{r})^{-(2\mathrm{ge}(\sigma_*)-1)} (1 + O(1/\sqrt{d}))$$

$$+ \frac{1}{N_{new}} \sum_{i=1}^{N_{new}} \nu_i + \tilde{O}\left( \alpha_1 m \sqrt{\frac{r^{-\mathrm{ge}(\sigma_*)}}{N_1}} \right),$$

where $\nu_i$ is a series of i.i.d. mean-zero random variable vectors which satisfy $\nu_i = \tilde{O}_p(\alpha_1 m \cdot r^{-\mathrm{ge}(\sigma_*)/2})$. Then, from Hoeffding's inequality, we have

$$\frac{1}{N_{new}} \sum_{i=1}^{N_{new}} \nu_i = \tilde{O}\left( \alpha_1 m \sqrt{\frac{r^{-\mathrm{ge}(\sigma_*)}}{N_{new}}} \right),$$

with high probability. Next we investigate the effect of $n_3$. From Lemma 9, $\sqrt{r} \Gamma^* \nabla_{\boldsymbol{u}} f_{\mathrm{IC}}(\boldsymbol{w}_i) = \tilde{O}(\alpha_1 m C_u) = \tilde{O}(\alpha_1 m r^{-1/2})$ holds. This leads to $n_3 \sqrt{r} \Gamma^* \nabla_{\boldsymbol{u}} f_{\mathrm{IC}}(\boldsymbol{w}_i) = \tilde{O}(\alpha_1 m r^{-\mathrm{ge}(\sigma_*)/2-1})$. Moreover, from Lemma 11, when $\mathrm{ge}(\sigma_*) = 2$ we have $\gamma(\boldsymbol{x}, y) = O(1/\sqrt{r})$. Combining this fact with $N_1 = \tilde{\Omega}(r^{\mathrm{ge}(\sigma_*)+1})$ yields $\sqrt{r} \Gamma^* \nabla_{\boldsymbol{u}} f_{\mathrm{IC}}(\boldsymbol{w}_i) = \tilde{O}(\alpha_1 m/r)$. Therefore, whether $\mathrm{ge}(\sigma_*) = 1$ or $\mathrm{ge}(\sigma_*) = 2$, we obtain

$$n_3 \sqrt{r} \Gamma^* \nabla_{\boldsymbol{u}} f_{\mathrm{IC}}(\boldsymbol{w}_i) = \tilde{O}(\alpha_1 m r^{-\mathrm{ge}(\sigma_*)-1/2}).$$

Taking the average over $i = 1, \ldots, N_{new}$ completes the proof. $\qquad\square$

**Lemma 13.** *Under the condition of Lemma 12, it holds that*

$$\langle \beta, \boldsymbol{h} \rangle = P_2' \Theta(\alpha_1 m) \langle \beta, \boldsymbol{u} \rangle \frac{1}{\kappa^{2\mathrm{ge}(\sigma_*)} \rho^{\mathrm{ge}(\sigma_*)}} (\sqrt{r})^{-(2\mathrm{ge}(\sigma_*)-1)} (1 + O(1/\sqrt{d}))$$

$$+ \tilde{O}\left( \alpha_1^2 m^2 r^{-\frac{\mathrm{ge}(\sigma_*)}{2}} \right) + \tilde{O}\left( \alpha_1 m \sqrt{\frac{r^{-\mathrm{ge}(\sigma_*)}}{N_1}} \right) + \tilde{O}\left( \alpha_1 m \sqrt{\frac{r^{-\mathrm{ge}(\sigma_*)}}{N_{new}}} \right) + o(\alpha_1 m r^{-\mathrm{ge}(\sigma_*)-1/2}),$$

*with high probability.*

*Proof.* As $\|\beta\| = 1$, using the result of Lemma 12 and taking the inner product of $\boldsymbol{h}$ and $\beta$ yields

$$\langle \beta, \boldsymbol{h} \rangle = P_2' \Theta(\alpha_1 m) \langle \beta, \boldsymbol{u} \rangle \frac{1}{\kappa^{2\mathrm{ge}(\sigma_*)} \rho^{\mathrm{ge}(\sigma_*)}} (\sqrt{r})^{-(2\mathrm{ge}(\sigma_*)-1)} (1 + O(1/\sqrt{d}))$$

$$+ \tilde{O}(\alpha_1^2 m^2 r^{-\frac{\mathrm{ge}(\sigma_*)}{2}}) + \tilde{O}\left( \alpha_1 m \sqrt{\frac{r^{-\mathrm{ge}(\sigma_*)}}{N_1}} \right) + \tilde{O}\left( \alpha_1 m \sqrt{\frac{r^{-\mathrm{ge}(\sigma_*)}}{N_{new}}} \right) + \tilde{O}\left( \alpha_1 m r^{-\mathrm{ge}(\sigma_*)-1/2} \right).$$

However, by carefully investigating the last term ($\tilde{O}(\alpha_1 m r^{-\mathrm{ge}(\sigma_*)-1/2})$) in the RHS, we can obtain a tighter bound of that term. Indeed, this term arises from

$$n_3 \left\langle \beta, \frac{1}{N_{new}} \sum_{i=1}^{N_{new}} \sqrt{r}\Gamma^* \nabla_{\boldsymbol{u}} f_{\mathrm{IC}}(\boldsymbol{w}_i) \right\rangle$$

$$= n_3 \alpha_1 L_m \frac{1}{N_{new}} \sum_{i=1}^{N_{new}} \left\{ \left\langle \beta, \sqrt{r}\Gamma^*\boldsymbol{x} \right\rangle \cdot \gamma(\boldsymbol{x}, y) \left\langle \beta^*, \boldsymbol{u} \right\rangle + \gamma(\boldsymbol{x}, y) \left\langle \boldsymbol{x}, \boldsymbol{u} \right\rangle \left\langle \beta^*, \sqrt{r}\Gamma^*\beta \right\rangle \right.$$

$$\left. + \left\langle \beta, \sqrt{r}\Gamma^*\boldsymbol{x} \right\rangle \cdot n_1 + \left\langle \boldsymbol{x}, \boldsymbol{u} \right\rangle \left\langle \beta, \boldsymbol{n}_2 \right\rangle \right\}.$$

We notice that the leading term of the right hand side is $n_3 \alpha_1 L_m P_0 \left\langle \boldsymbol{u}, \boldsymbol{x} \right\rangle$ when $\mathrm{ge}(\sigma_*) = 1$, and $n_3 \alpha_1 m P_1 \left\langle \beta, \sqrt{r}\Gamma_*^\top \Gamma_u \boldsymbol{x} \right\rangle / \rho \cdot \left\langle \boldsymbol{x}, \boldsymbol{u} \right\rangle$ when $ge(\sigma_*) = 2$. Note that $\mathbb{E}_{\boldsymbol{x}}[P_0 \left\langle \boldsymbol{u}, \boldsymbol{x} \right\rangle] = 0$ and $\mathbb{E}_{\boldsymbol{x}}[P_1 \left\langle \sqrt{r}\Gamma_u^\top \Gamma_* \beta, \boldsymbol{x} \right\rangle / \rho \cdot \left\langle \boldsymbol{x}, \boldsymbol{u} \right\rangle] = P_1 \left\langle \sqrt{r}\Gamma_u^\top \Gamma_* \beta, \boldsymbol{u} \right\rangle / \rho = \tilde{O}(r^{-3/2})$ because of $\sqrt{r}\Gamma_u^\top \Gamma^*\beta = \tilde{O}(1/\sqrt{r})\beta + \tilde{O}(1/r) + \left\langle \beta, \boldsymbol{u} \right\rangle \boldsymbol{u}$ (see Lemma 32). Moreover, we have $\mathbb{E}_{\boldsymbol{x}}[P_0^2 \left\langle \boldsymbol{u}, \boldsymbol{x} \right\rangle^2] = \tilde{O}(1)$ and $\mathbb{E}_{\boldsymbol{x}}[P_1^2 \left\langle \sqrt{r}\Gamma_u^\top \Gamma_* \beta, \boldsymbol{x} \right\rangle^2 / \rho^2 \cdot \left\langle \boldsymbol{x}, \boldsymbol{u} \right\rangle^2] \leq P_1^2/\rho^2 \cdot \mathbb{E}[\left\langle \sqrt{r}\Gamma_u^\top \Gamma_* \beta, \boldsymbol{x} \right\rangle^4]^{1/2} \mathbb{E}[\left\langle \boldsymbol{x}, \boldsymbol{u} \right\rangle^4]^{1/2} = \tilde{O}(1)$. Therefore, Lemma 26 yields that

$$\frac{1}{N_{new}} \sum_{i=1}^{N_{new}} P_0 \left\langle \boldsymbol{u}, \boldsymbol{w}_i \right\rangle = \tilde{O}\left( \sqrt{\frac{1}{r N_{new}}} \right),$$

$$\frac{1}{N_{new}} \sum_{i=1}^{N_{new}} P_1 \left\langle \beta, \sqrt{r}\Gamma_u \Gamma^* \boldsymbol{w}_i \right\rangle / \rho \left\langle \boldsymbol{w}_i, \boldsymbol{u} \right\rangle = \tilde{O}(r^{-3/2}) + \tilde{O}(r^{-1} N_{new}^{-1/2}).$$

Since $n_3 = o_d(P_2' r^{-\mathrm{ge}(\sigma_*)/2-1/2} \log^{-2 \deg(\sigma_*)+2} d)$ and $N_{new} = \tilde{\Omega}(r^{\mathrm{ge}(\sigma_*)+2})$, this implies that

$$n_3 \left\langle \beta, \frac{1}{N_{new}} \sum_{i=1}^{N_{new}} \sqrt{r}\Gamma^* \nabla_{\boldsymbol{u}} f_{\mathrm{IC}}(\boldsymbol{w}_i) \right\rangle = o(\alpha_1 m r^{-\mathrm{ge}(\sigma_*)-1/2}).$$

This yields the assertion. $\qquad\square$

Setting the parameters directly yields the following lemma:

**Lemma 14.** *Set $N_1 = \tilde{\Omega}(r^{\mathrm{ge}(\sigma_*)+2})$, $N_{new} = \tilde{\Omega}(r^{\mathrm{ge}(\sigma_*)+2})$, $\alpha_1 m = \tilde{\Theta}(r^{-\mathrm{ge}(\sigma_*)/2-1})$, $\eta_1 = \tilde{\Theta}(r^{3\mathrm{ge}(\sigma_*)/2+3/2})$ and $\lambda_1 = \eta_1^{-1}$, then we have*

$$\boldsymbol{u}^1 = \tilde{\Theta}(1)\beta + \tilde{O}(1),$$

*and*

$$\left\langle \beta, \boldsymbol{u}^1 \right\rangle = \tilde{\Theta}(1) + o(1),$$

*with high probability. In particular, we have that*

$$\left\langle \beta, \frac{\boldsymbol{u}^1}{\|\boldsymbol{u}^1\|} \right\rangle \geq 1/\mathrm{polylog}(d).$$

# E  STRONG RECOVERY

After achieving weak recovery, we train the vector $\boldsymbol{u}$, regarding the in-context examples $(\boldsymbol{x}_i, y_i)$ as training data. The idea of proof in this section is taken from Lee et al. (2024), but we achieve better sample complexity.

**Lemma 15.** *Let $\boldsymbol{h} = \frac{1}{2}\sqrt{r}\Gamma^* \nabla_{\boldsymbol{u}}(f_{\mathrm{IC}}(\boldsymbol{x}) - y)^2$. Then, we have that*

$$\boldsymbol{h} = \Theta(\alpha_2 m) \langle \beta, \boldsymbol{u} \rangle^{2\mathrm{ie}(\sigma_*)-1} \frac{\beta}{\kappa^{\mathrm{ie}(\sigma_*)+1}\rho^{\mathrm{ie}(\sigma_*)}} (1 + O(1/\sqrt{d}))$$

$$+ \tilde{O}\left(\alpha_2^2 m^2 \sqrt{r}\right) + \tilde{O}\left(\alpha_2 m \sqrt{\frac{r}{N_2}}\right) + \nu,$$

$$\langle \beta, \boldsymbol{h} \rangle = \Theta(\alpha_2 m) \langle \beta, \boldsymbol{u} \rangle^{2\mathrm{ie}(\sigma_*)-1} \frac{1}{\kappa^{\mathrm{ie}(\sigma_*)+1}\rho^{\mathrm{ie}(\sigma_*)}} (1 + O(1/\sqrt{d}))$$

$$+ \tilde{O}(\alpha_2^2 m^2 \sqrt{r}) + \tilde{O}\left(\alpha_2 m \sqrt{\frac{r}{N_2}}\right) + \nu_6,$$

$$\langle \boldsymbol{u}, \boldsymbol{h} \rangle = \Theta(\alpha_2 m) \langle \beta, \boldsymbol{u} \rangle^{2\mathrm{ie}(\sigma_*)} \frac{1}{\kappa^{\mathrm{ie}(\sigma_*)+1}\rho^{\mathrm{ie}(\sigma_*)}} (1 + O(1/\sqrt{d}))$$

$$+ \tilde{O}(\alpha_2^2 m^2 \sqrt{r}) + \tilde{O}\left(\alpha_2 m \sqrt{\frac{r}{N_2}}\right) + \nu_7,$$

*with high probability, where $\nu_6$ and $\nu_7$ are mean-zero sub-Weibull random variables. Moreover, $\nu_6 = \tilde{O}(\alpha_2 m)$ and $\nu_7 = \tilde{O}(\alpha_2 m)$ hold with high probability.*

*Proof.* Remember that, in the stage of strong recovery, the initialization scale is $\alpha_2$, the context length is $N_2$ and $C_u = 1$. By using the same argument to derive Lemma 6, we have

$$\boldsymbol{h} = \Theta(\alpha_2 m) \langle \beta, \boldsymbol{u} \rangle \frac{\beta}{\kappa^{\mathrm{ie}(\sigma_*)+1}\rho^{\mathrm{ie}(\sigma_*)}} \{(\kappa\sqrt{r})^{-(\mathrm{ie}(\sigma_*)-1)}(1 + O(1/\sqrt{d})) + \langle \beta, \boldsymbol{u} \rangle^{2\mathrm{ie}(\sigma_*)-2}(1 + O(1/\sqrt{d}))\}$$

$$+ \tilde{O}(\alpha_2^2 m^2 \sqrt{r}) + \tilde{O}\left(\alpha_2 m \sqrt{\frac{r}{N_2}}\right) + \nu.$$

Now, since $\langle \beta, \boldsymbol{u} \rangle \geq 1/\mathrm{polylog}(d)$, the term $\langle \beta, \boldsymbol{u} \rangle^{2\mathrm{ie}(\sigma_*)-2}$ dominates over the term $(\kappa\sqrt{r})^{-(\mathrm{ie}(\sigma_*)-1)}$. Therefore, with high probability, we have

$$\boldsymbol{h} = \Theta(\alpha_2 m) \langle \beta, \boldsymbol{u} \rangle^{2\mathrm{ie}(\sigma_*)-1} \frac{\beta}{\kappa^{\mathrm{ie}(\sigma_*)+1}\rho^{\mathrm{ie}(\sigma_*)}} (1 + O(1/\sqrt{d}))$$

$$+ \tilde{O}(\alpha_2^2 m^2 \sqrt{r}) + \tilde{O}\left(\alpha_2 m \sqrt{\frac{r}{N_2}}\right) + \nu.$$

Next, since $\|\beta\| = 1$, we have

$$\langle \beta, \boldsymbol{h} \rangle = \Theta(\alpha_2 m) \langle \beta, \boldsymbol{u} \rangle^{2\mathrm{ie}(\sigma_*)-1} \frac{1}{\kappa^{\mathrm{ie}(\sigma_*)+1}\rho^{\mathrm{ie}(\sigma_*)}} (1 + O(1/\sqrt{d}))$$

$$+ \tilde{O}(\alpha_2^2 m^2 \sqrt{r}) + \tilde{O}\left(\alpha_2 m \sqrt{\frac{r}{N_2}}\right) + \langle \nu, \beta \rangle,$$

with high probability, where $\nu = \tilde{O}(\alpha_2 m \sqrt{r})$. Then, since we see that

$$\nu = y \cdot \alpha_2 L_m \left\{ \sqrt{r}\Gamma^* \boldsymbol{x} \cdot \gamma(\boldsymbol{x}, y) \langle \beta^*, \boldsymbol{u} \rangle + \sqrt{r}\Gamma^* \gamma(\boldsymbol{x}, y) \langle \boldsymbol{x}, \boldsymbol{u} \rangle \beta^* + \sqrt{r}\Gamma^* \boldsymbol{x} \cdot n_1 + \langle \boldsymbol{x}, \boldsymbol{u} \rangle n_2 \right\}$$

$$- \mathbb{E}_{\boldsymbol{x}}[y \cdot \alpha_2 L_m \left\{ \sqrt{r}\Gamma^* \boldsymbol{x} \cdot (\gamma(\boldsymbol{x}, y) \langle \beta^*, \boldsymbol{u} \rangle + \sqrt{r}\Gamma^* \gamma(\boldsymbol{x}, y) \langle \boldsymbol{x}, \boldsymbol{u} \rangle \beta^* + \sqrt{r}\Gamma^* \boldsymbol{x} \cdot n_1 + \langle \boldsymbol{x}, \boldsymbol{u} \rangle n_2 \right\}],$$

it holds that

$$\langle \beta, \nu \rangle$$

$$= y \cdot \alpha_2 L_m \left\{ \langle \beta, \sqrt{r}\Gamma^* \boldsymbol{x} \rangle \cdot \gamma(\boldsymbol{x}, y) \langle \beta^*, \boldsymbol{u} \rangle + \gamma(\boldsymbol{x}, y) \langle \boldsymbol{x}, \boldsymbol{u} \rangle \langle \beta, \sqrt{r}\Gamma^* \beta^* \rangle + \langle \beta, \sqrt{r}\Gamma^* \boldsymbol{x} \rangle \cdot n_1 + \langle \boldsymbol{x}, \boldsymbol{u} \rangle \langle \beta, n_2 \rangle \right\}$$

$$- \mathbb{E}_{\boldsymbol{x}}[y \cdot \alpha_2 L_m \left\{ \langle \beta, \sqrt{r}\Gamma^* \boldsymbol{x} \rangle \cdot \gamma(\boldsymbol{x}, y) \langle \beta^*, \boldsymbol{u} \rangle + \gamma(\boldsymbol{x}, y) \langle \boldsymbol{x}, \boldsymbol{u} \rangle \langle \beta, \sqrt{r}\Gamma^* \beta^* \rangle + \langle \beta, \sqrt{r}\Gamma^* \boldsymbol{x} \rangle \cdot n_1 + \langle \boldsymbol{x}, \boldsymbol{u} \rangle \langle \beta, n_2 \rangle \right\}].$$

Here, the first term of the right hand side satisfies

$$y \cdot \alpha_2 L_m \left\{ \langle \beta, \sqrt{r}\Gamma^* \boldsymbol{x} \rangle \cdot \gamma(\boldsymbol{x}, y) \langle \beta^*, \boldsymbol{u} \rangle + \gamma(\boldsymbol{x}, y) \langle \boldsymbol{x}, \boldsymbol{u} \rangle \langle \beta, \sqrt{r}\Gamma^* \beta^* \rangle \right.$$

$$\left. + \langle \beta, \sqrt{r}\Gamma^* \boldsymbol{x} \rangle \cdot n_1 + \langle \boldsymbol{x}, \boldsymbol{u} \rangle \langle \beta, n_2 \rangle \right\}$$

$$= \tilde{O}(\alpha_2 m),$$

with high probability, and $\langle\beta,\nu\rangle$ is the difference between this random variable and its expectation. Thus, by defining $\nu_6 = \langle\beta,\nu\rangle$, Hoeffding's inequality yields that $\nu_6 = \tilde{O}(\alpha_2 m)$ with high probability. Moreover, since $|\langle\beta,\nu\rangle| \leq \|\nu\|$, $\nu_6 = \langle\beta,\nu\rangle$ has sub-Weibull tail.

Likewise, we have that

$$\langle\boldsymbol{u},\boldsymbol{h}\rangle = \Theta(\alpha_2 m)\langle\beta,\boldsymbol{u}\rangle^{2\mathrm{ie}(\sigma_*)}\frac{1}{\kappa^{\mathrm{ie}(\sigma_*)+1}\rho^{\mathrm{ie}(\sigma_*)}}(1+O(1/\sqrt{d}))$$
$$+\tilde{O}(\alpha_2^2 m^2\sqrt{r}) + \tilde{O}\left(\alpha_2 m\sqrt{\frac{r}{N_2}}\right) + \nu_7,$$

where $\nu_7$ is a mean-zero sub-Weibull random variable satisfying $\nu_7 = \tilde{O}(\alpha_2 m)$. $\qquad\square$

**Lemma 16.** *Let* $a^{(n)} = \langle\beta,\boldsymbol{u}^{(n)}\rangle$. *Suppose that* $c_1 \leq a^{(n)} \leq 1-\varepsilon$ *where* $c_1 = \tilde{\Theta}(1)$. *Set* $\alpha_2 m = \tilde{\Theta}(\varepsilon/r)$, $N_2 = \tilde{\Theta}(r^2)$ *and* $\eta_2 = \tilde{\Theta}(1/\sqrt{r})$. *Then, there exists* $c_3 = \tilde{\Theta}(1)$ *which satisfies*

$$a^{(n+1)} \geq a^{(n)} + \frac{c_3\varepsilon}{r\sqrt{r}}a^{(n)\,2\mathrm{ie}(\sigma_*)-1}(1-a^{(n)^2})(1-O(1/\sqrt{d})) + \nu_8 - \tilde{O}(\varepsilon/r^2),$$

*with high probability, where* $\nu_8$ *is a mean-zero sub-Weibull random variable which satisfies* $\nu_8 = \tilde{O}(\varepsilon/r\sqrt{r})$ *with high probability.*

*Proof.* Using the projection matrix $\boldsymbol{P}_u = \boldsymbol{I} - uu^\top$, online SGD update can be written as

$$\boldsymbol{u}^{(n+1)} = \boldsymbol{u}^{(n)} + \frac{\boldsymbol{u}^{(n)}+\eta_2\boldsymbol{P}_{\boldsymbol{u}^{(n)}}\boldsymbol{h}}{\|\boldsymbol{u}^{(n)}+\eta_2\boldsymbol{P}_{\boldsymbol{u}^{(n)}}\boldsymbol{h}\|},$$

which gives

$$\left\langle\beta,\boldsymbol{u}^{(n+1)}\right\rangle$$
$$= \left\langle\beta,\boldsymbol{u}^{(n)}\right\rangle + \left\langle\beta,\frac{\boldsymbol{u}^{(n)}+\eta_2\boldsymbol{P}_{\boldsymbol{u}^{(n)}}\boldsymbol{h}}{\|\boldsymbol{u}^{(n)}+\eta_2\boldsymbol{P}_{\boldsymbol{u}^{(n)}}\boldsymbol{h}\|}\right\rangle$$
$$= \left\langle\beta,\boldsymbol{u}^{(n)}\right\rangle + \eta_2\left\langle\beta,\boldsymbol{P}_{\boldsymbol{u}^{(n)}}\boldsymbol{h}\right\rangle - \frac{1}{2}\eta_2^2\|\boldsymbol{P}_{\boldsymbol{u}^{(n)}}\boldsymbol{h}\|^2 + O(\eta_2^3)$$
$$= \left\langle\beta,\boldsymbol{u}^{(n)}\right\rangle + \eta_2\left\langle\beta,\boldsymbol{h}\right\rangle - \eta_2\left\langle\beta,\boldsymbol{u}^{(n)}\right\rangle\left\langle\boldsymbol{u}^{(n)},\boldsymbol{h}\right\rangle - \frac{1}{2}\eta_2^2\|\boldsymbol{P}_{\boldsymbol{u}^{(n)}}\boldsymbol{h}\|^2 + O((\eta_2\|\boldsymbol{h}\|)^3).$$

By the definition of $a^{(n)}$, i.e., $a^{(n)} = \langle\beta,\boldsymbol{u}^{(n)}\rangle$, we have

$$a^{(n+1)} = a^{(n)} + \eta_2\langle\beta,\boldsymbol{h}\rangle - \eta_2 a^{(n)}\left\langle\boldsymbol{u}^{(n)},\boldsymbol{h}\right\rangle - \frac{1}{2}\eta_2^2\|\boldsymbol{P}_{\boldsymbol{u}^{(n)}}\boldsymbol{h}\|^2 + O((\eta_2\|\boldsymbol{h}\|)^3).$$

Now, from Lemma 15, we also have

$$\langle\beta,\boldsymbol{h}\rangle = \frac{\varepsilon}{r}a^{(n)\,2\mathrm{ie}(\sigma_*)-1}(\tilde{\Theta}(1)+O(1/\sqrt{d})) + \tilde{O}(\varepsilon/r\sqrt{r}) + \tilde{O}(\varepsilon/r\sqrt{r}) + \nu_6$$

and

$$\langle\boldsymbol{u},\boldsymbol{h}\rangle = \frac{\varepsilon}{r}a^{(n)\,2\mathrm{ie}(\sigma_*)}(\tilde{\Theta}(1)+O(1/\sqrt{d})) + \tilde{O}(\varepsilon/r\sqrt{r}) + \tilde{O}(\varepsilon/r\sqrt{r}) + \nu_7.$$

Moreover, since $\|\boldsymbol{P}_{\boldsymbol{u}^{(n)}}\boldsymbol{h}\| = \tilde{O}(\alpha_2 m\sqrt{r}) = \tilde{O}(\varepsilon/\sqrt{r})$, it holds that

$$\frac{1}{2}\eta_2^2\|\boldsymbol{P}_{\boldsymbol{u}^{(n)}}\boldsymbol{h}\|^2 = \tilde{O}(\varepsilon^2/r^2).$$

Therefore, ignoring the term $O((\eta_2\|\boldsymbol{h}\|)^3)$, we arrive at

$$a^{(n+1)} = a^{(n)} + \eta_2\langle\beta,\boldsymbol{h}\rangle - \eta_2 a^{(n)}\langle\boldsymbol{u},\boldsymbol{h}\rangle - \frac{1}{2}\eta_2^2\|\boldsymbol{P}_{\boldsymbol{u}^{(n)}}\boldsymbol{h}\|^2$$
$$= a^{(n)} + \frac{\eta_2\varepsilon}{r}a^{(n)\,2\mathrm{ie}(\sigma_*)-1}(1-a^{(n)^2})(\tilde{\Theta}(1)+O(1/\sqrt{d})) + \eta_2(\nu_6-a^{(n)}\nu_7) + \eta_2\tilde{O}(\varepsilon/r\sqrt{r}) + \tilde{O}(\varepsilon/r^2)$$
$$\geq a^{(n)} + \frac{c_3\varepsilon}{r\sqrt{r}}a^{(n)\,2\mathrm{ie}(\sigma_*)-1}(1-a^{(n)^2})(1-O(1/\sqrt{d})) + \nu_8 - \tilde{O}(\varepsilon/r^2),$$

where $\nu_8$ is a mean-zero sub-Weibull random variable satisfying $\nu_8 = \tilde{O}(\varepsilon/r\sqrt{r})$. $\qquad\square$

**Lemma 17.** *Suppose that $\boldsymbol{u}^{(1)}$ satisfies $\langle \beta, \boldsymbol{u}^{(1)} \rangle \geq c_1$, where $c_1 \geq 1/\mathrm{polylog}(d)$. Then, there exists $N_3 = \tilde{\Theta}(\frac{r\sqrt{r}}{\varepsilon} \log \frac{1}{\varepsilon})$ such that*

$$\left\langle \beta, \boldsymbol{u}^{(N_3+1)} \right\rangle \geq 1 - \varepsilon$$

*with high probability.*

*Proof.* Before going into the proof, we first explain the main idea of the proof. Following the same argument as Lemma 19 in Lee et al. (2024), we can see that after $\tilde{\Theta}(r\sqrt{r}/\varepsilon)$ steps, $a^{(n)}$ becomes larger than a constant $\sqrt{\frac{k+1}{k+2}}$, where $k = 2\mathrm{ie}(\sigma_*) - 1$. Then, by applying the Mean Value theorem, we can observe that $1 - a^{(i+1)} \lesssim (1 - \frac{c_3\varepsilon}{r\sqrt{r}}C_k)(1 - a^{(i)})$ where $C_k = \Theta(1)$. This means $1 - a^{(i)}$ converges to 0 geometrically, which yields the required data length $N_3 = \tilde{\Theta}(\frac{r\sqrt{r}}{\varepsilon} \log \frac{1}{\varepsilon})$.

Let $\nu_9^{(i)} = r\sqrt{r}\nu_8^{(i)}/\varepsilon$ and $k = 2\mathrm{ie}(\sigma_*) - 1$. Because $v_9^{(i)}$ is a sequence of mean-zero sub-Weibull random variables with $v_9^{(i)} = \tilde{O}_p(1)$, we have

$$\left| \sum_{i=j}^{j+l} \nu_9^{(i)} \right| = C\sqrt{l}, \tag{6}$$

for any $1 \leq j, l \leq N_3$ with high probability, where $C = \tilde{O}(1)$. Let $c_k = 1 - \sqrt{\frac{k+1}{k+2}}$. Note that $c_k$ is a constant that only depends on $k$, that is $c_k = O(1)$. Suppose that $c_1 \leq a^{(i)} \leq 1 - \frac{1}{3}c_k$ for all $i = 1, 2, \ldots, N_3$. Then, from Lemma 16, we have

$$a^{(n+1)} \geq a^{(n)} + \frac{c_3\varepsilon}{r\sqrt{r}} a^{(n)\,2\mathrm{ie}(\sigma_*)-1}(1 - a^{(n)^2})(1 - O(1/\sqrt{d})) + \nu_8 - \tilde{O}(\varepsilon/r^2),$$

for $i \leq N_3$. The term $O(1/\sqrt{d})$ is smaller than 1, thus we may ignore this term. Moreover, the term $\tilde{O}(\varepsilon/r^2)$ is dominated by $\frac{c_3\varepsilon}{r\sqrt{r}} a^{(n)\,2\mathrm{ie}(\sigma_*)-1}(1 - a^{(n)^2}) = \tilde{O}(\frac{\varepsilon}{r\sqrt{r}})$. Let $c_2 = c_1^{(2\mathrm{ie}(\sigma_*)-1)}$. By ignoring these terms and using $1 - a^{(i)^2} \geq \frac{1}{3}c_k$, we have

$$a^{(i+1)} \geq a^{(i)} + \frac{c_2 c_3}{r\sqrt{r}}\varepsilon \cdot c_k/3 + \frac{\varepsilon}{r\sqrt{r}} \cdot \nu_9^{(i)}$$

$$\geq a^{(1)} + \frac{c_2 c_3 c_k}{3r\sqrt{r}}\varepsilon i - \frac{\varepsilon}{r\sqrt{r}}\left| \sum_{j=1}^{i} \nu_9^{(j)} \right|$$

$$\geq a^{(1)} + (\frac{c_2 c_3 c_k}{3r\sqrt{r}}\varepsilon)i - \frac{\varepsilon}{r\sqrt{r}}C\sqrt{i}.$$

If $i \leq \frac{r^3 c_1^2}{4\varepsilon^2 C^2}$, then $\frac{\varepsilon}{r\sqrt{r}}C\sqrt{i} \leq c_1/2$, and if $i \geq (\frac{6C}{c_2 c_3 c_k})^2$, then $\frac{\varepsilon}{r\sqrt{r}}C\sqrt{i} \leq \frac{c_2 c_3 c_k}{6r\sqrt{r}}\varepsilon i$. By observing the order in terms of $r$, we have $\frac{r^3 c_1^2}{4\varepsilon^2 C^2} \geq (\frac{6C}{c_2 c_3 c_k})^2$ when $r$ is sufficiently large. Therefore, it holds that

$$a^{(i+1)} \geq \frac{c_1}{2} + \frac{c_2 c_3 c_k}{6r\sqrt{r}}\varepsilon i. \tag{7}$$

When $i = \frac{6r\sqrt{r}}{c_2 c_3 c_k \varepsilon} = \tilde{\Theta}(r\sqrt{r}/\varepsilon)$, then the RHS of Eq. (7) exceeds 1. Therefore, there exists $i^* \leq N_3 = \tilde{\Theta}(\frac{r\sqrt{r}}{\varepsilon} \log \frac{1}{\varepsilon})$ such that $a^{(i^*)} \geq 1 - c_k/3$. Next we prove that $a^{(i)} \geq 1 - c_k = \sqrt{\frac{k+1}{k+2}}$ for all $i = i^*, i^*+1, \ldots, N_3$. In this setting, $a^{(i+1)} - a^{(i)} = \tilde{O}(\frac{\varepsilon}{r\sqrt{r}})$ holds. Therefore, if there exists $i \geq i^*$ such that $a^{(i-1)} \geq 1 - c_k/3$ and $a^{(i)} \leq 1 - c_k/3$, we have $a^{(i)} \geq 1 - 2c_k/3$ with high probability. Also, if $a^{(i+l)} \leq 1 - c_k/3$ holds for all $l = 0, 1, \ldots, j-1$, we have

$$a^{(i+j)} \geq 1 - \frac{2c_k}{3} + \frac{c_2 c_3 c_k}{3r\sqrt{r}}\varepsilon j - \frac{\varepsilon}{r\sqrt{r}}C\sqrt{j}.$$

If $j \leq \frac{r^3 c_k^2}{9\varepsilon^2 C^2} = \tilde{O}(r^3/\varepsilon^2)$, then $\frac{\varepsilon}{r\sqrt{r}}C\sqrt{j} \leq c_k/3$, and if $j \geq (\frac{3C}{c_2 c_3 c_k})^2 = \tilde{O}(1)$, then $\frac{\varepsilon}{r\sqrt{r}}C\sqrt{j} \leq \frac{c_2 c_3 c_k}{3r\sqrt{r}}\varepsilon j$. Since $\frac{r^3 c_k^2}{9\varepsilon^2 C^2} \geq (\frac{3C}{c_2 c_3 c_k})^2$ when $r$ is sufficiently large, we have

$$a^{(i+j)} \geq 1 - c_k$$

until $i + j = N_3 + 1$ or $a^{(i+j)} \geq 1 - c_k/3$ holds again. By repeating this argument if necessary, we get $a^{(i+1)} \geq 1 - c_k = \sqrt{\frac{k+1}{k+2}}$ for all $i^* \leq i \leq N_3$.

We continue by showing that, after we achieve $a^{(i^*)} \leq 1 - c_k$, the number of remaining steps needed to ensure $a^{(i)} \geq 1 - \varepsilon$ is $\tilde{O}(\frac{r\sqrt{r}}{\varepsilon}\log\frac{1}{\varepsilon})$. Let $F(x) = x + \frac{c_3 \varepsilon}{r\sqrt{r}}x^k(1 - x^2)$. Then,

$$a^{(i^*+i+1)} = F(a^{(i^*+i)}) + \frac{\varepsilon}{r\sqrt{r}}\nu_9^{(i^*+i)} - \tilde{O}(\varepsilon/r^2).$$

By the Mean Value theorem, there exists $\gamma$ such that

$$\frac{1 - F(a^{(i^*+i)})}{1 - a^{(i^*+i)}} = F'(\gamma),$$

and $a^{(i^*+i)} < \gamma < 1$. Now $F'(x) = 1 + \frac{c_2 c_3 \varepsilon}{r\sqrt{r}}x^{k-1}(k - (k+2)x^2)$, and since $\gamma > a^{(i^*+i)} \geq \sqrt{\frac{k+1}{k+2}}$, we have $k - (k+2)\gamma^2 < -1$, which leads to

$$F'(\gamma) \leq 1 - \frac{c_3 \varepsilon}{r\sqrt{r}}\gamma^{k-1} < 1 - \frac{c_3 \varepsilon}{r\sqrt{r}}\left(\sqrt{\frac{k+1}{k+2}}\right)^{k-1}.$$

Let $C_k = \left(\sqrt{\frac{k+1}{k+2}}\right)^{k-1}$, then $1 - F(a^{(i^*+i)}) < (1 - \frac{c_3 \varepsilon}{r\sqrt{r}}C_k)(1 - a^{(i^*+i)})$, which yields that

$$1 - a^{i^*+i+1} < \left(1 - \frac{c_3 \varepsilon}{r\sqrt{r}}C_k\right)(1 - a^{(i^*+i)}) + \frac{\varepsilon}{r\sqrt{r}}\nu_9^{(i^*+i)} + \tilde{O}(\varepsilon/r^2).$$

Noting that $(1 - \frac{c_3 \varepsilon}{r\sqrt{r}}C_k) < 1$, taking the sum leads to

$$1 - a^{(N_3+1)} < \left(1 - \frac{c_3 \varepsilon}{r\sqrt{r}}C_k\right)^{N_3+1-i^*}(1 - a^{(i^*)}) + \sum_{i=i^*}^{N_3}\frac{\varepsilon}{r\sqrt{r}}\nu_9^{(i)} + \tilde{O}(\varepsilon N_3/r^2).$$

Since $\lim_{t\to\infty}(1 - \frac{1}{t})^t = 1/e$, if we set $N_3 = i^* - 1 + \frac{r\sqrt{r}}{c_3 C_k \varepsilon}\log(1/\varepsilon)$, then we have

$$\left(1 - \frac{c_3 \varepsilon}{r\sqrt{r}}C_k\right)^{N_3+1-i^*}(1 - a^{(i^*)}) < \left(1 - \frac{c_3 \varepsilon}{r\sqrt{r}}C_k\right)^{\frac{r\sqrt{r}}{c_3 C_k \varepsilon}\log(1/\varepsilon)}$$
$$\approx (1/e)^{\log(1/\varepsilon)} = \varepsilon.$$

Finally, by noticing that

$$\left|\sum_{i=i^*}^{N_3}\frac{\varepsilon}{r\sqrt{r}}\nu_9^{(i)}\right| \leq \frac{\varepsilon}{r\sqrt{r}}C\sqrt{N_3 - i^* + 1} = \tilde{O}(r^{-3/4}),$$

we can see that this term is negligible. Moreover, the third term $\tilde{O}(\varepsilon N_3/r^2) = \tilde{O}(\frac{\log 1/\varepsilon}{\sqrt{r}})$ is also negligible. By summarizing the argument above, we conclude that

$$1 - a^{(N_3+1)} < \varepsilon.$$

$\square$

In Lemma 9, we assumed that $\langle \boldsymbol{u}, \boldsymbol{x}_i \rangle = \tilde{O}_d(1)$. Now $\boldsymbol{u}^{(n)}$ and $\boldsymbol{x}_i$ are not independent of each other, so we need to ensure that $\langle \boldsymbol{u}^{(j)}, \boldsymbol{x}_i \rangle = \tilde{O}_d(1)$ for all $j = 1, 2, \ldots, N_3 + 1$.

**Lemma 18.** *It holds that*

$$\langle \boldsymbol{x}_i^*, \boldsymbol{h}\rangle = \Theta(\alpha_2 m)\, \langle \beta, \boldsymbol{u}\rangle^{2\mathrm{ie}(\sigma_*)-1}\, \frac{\langle \boldsymbol{x}_i^*, \beta\rangle}{(\kappa\rho)^{\mathrm{ie}(\sigma_*)}}(1 + O(1/\sqrt{d}))$$

$$+ \tilde{O}(\alpha_2^2 m^2 r) + \tilde{O}(\alpha_2 m r/\sqrt{N_2}) + \nu_*,$$

*where $\nu_*$ is a mean-zero random variable satisfying $\nu_* = \tilde{O}(\alpha_2 m \sqrt{r})$.*

*Proof.* From Lemma 15, we have

$$\boldsymbol{h} = \Theta(\alpha_2 m)\, \langle \beta, \boldsymbol{u}\rangle^{2\mathrm{ie}(\sigma_*)-1}\, \frac{\beta}{(\kappa\rho)^{\mathrm{ie}(\sigma_*)}}(1 + O(1/\sqrt{d}))$$

$$+ \tilde{O}(\alpha_2^2 m^2 \sqrt{r}) + \tilde{O}(\alpha_2 m \sqrt{\frac{r}{N_2}}) + \nu.$$

Considering $\boldsymbol{x}_i^* = \tilde{O}(\sqrt{r})$, taking the inner product of $\boldsymbol{x}_i^*$ and $\boldsymbol{h}$ leads to

$$\langle \boldsymbol{x}_i^*, \boldsymbol{h}\rangle = \Theta(\alpha_2 m)\, \langle \beta, \boldsymbol{u}\rangle^{2\mathrm{ie}(\sigma_*)-1}\, \frac{\langle \boldsymbol{x}_i^*, \beta\rangle}{(\kappa\rho)^{\mathrm{ie}(\sigma_*)}}(1 + O(1/\sqrt{d}))$$

$$+ \tilde{O}(\alpha_2^2 m^2 r) + \tilde{O}(\alpha_2 m r/\sqrt{N_2}) + \langle \boldsymbol{x}_i^*, \nu\rangle.$$

Following the same argument as Lemma 15 yields $\langle \boldsymbol{x}_i^*, \nu\rangle = \tilde{O}(\alpha_2 m \sqrt{r})$. Thus, by letting $\nu_* = \langle \boldsymbol{x}_i^*, \nu\rangle$, we obtain the assertion. □

**Lemma 19.** *Set $\alpha_2 m = \tilde{\Theta}(\varepsilon/r)$, $N_2 = \tilde{\Theta}(r^2)$, $N_3 = \tilde{\Theta}(\frac{r\sqrt{r}}{\varepsilon}\log\frac{1}{\varepsilon})$ and $\eta_2 = \tilde{\Theta}(1/\sqrt{r})$(This is exactly the same situation as Lemma 16 and Lemma 17). Then we have*

$$\left\langle \boldsymbol{x}_i^*, \boldsymbol{u}^{(j)}\right\rangle = \tilde{O}_d(1),$$

*for all $j = 1, 2, \ldots, N_3 + 1$.*

*Proof.* When $j = 1$, $\boldsymbol{x}_i$ and $\boldsymbol{u}^{(1)}$ are independent of each other, thus Lemma 29 yields the desired result. Substituting the parameters into the result of Lemma 18 yields

$$\eta_2 \langle \boldsymbol{x}_i^*, \boldsymbol{h}\rangle = \tilde{O}\left(\frac{\varepsilon}{r\sqrt{r}}(1 + O(1/\sqrt{d}))\right) + \tilde{O}\left(\frac{\varepsilon}{r\sqrt{r}}\right) + \eta_2 \nu_*,$$

where $\eta_2 \nu_* = \tilde{O}(\varepsilon/r)$. First, following the same argument to derive Eq. (6) yields

$$\left|\sum_{k=1}^{j} \eta_2 \nu_*\right| = \tilde{O}(\varepsilon \sqrt{j}/r),$$

for $j = 1, 2, \ldots, N_3$. Since $\varepsilon\sqrt{N_3}/r = \tilde{O}_d(r^{-1/4}) = o_d(1)$, we can ignore the effect of $\eta_2 \nu_*$. Then, we can see that

$$\left\langle \boldsymbol{x}_i^*, \boldsymbol{u}^{(j+1)}\right\rangle = \left\langle \boldsymbol{x}_i^*, \boldsymbol{u}^{(j)}\right\rangle + \eta_2 \langle \boldsymbol{x}_i^*, \boldsymbol{h}\rangle - \eta_2 \left\langle \boldsymbol{h}, \boldsymbol{u}^{(j)}\right\rangle \left\langle \boldsymbol{u}^{(j)}, \boldsymbol{x}_i^*\right\rangle - \frac{1}{2}\eta_2^2 \|\boldsymbol{P}_{\boldsymbol{u}^{(n)}}\boldsymbol{h}\|^2 + O((\eta_2\|\boldsymbol{h}\|)^3)$$

$$= \left\langle \boldsymbol{x}_i^*, \boldsymbol{u}^{(j)}\right\rangle + \left\langle \beta, \boldsymbol{u}^{(j)}\right\rangle^{2\mathrm{ie}(\sigma_*)-1}\tilde{O}\left(\frac{\varepsilon}{r\sqrt{r}}\right)\left(\langle \beta, \boldsymbol{x}_i^*\rangle - \left\langle \boldsymbol{u}^{(j)}, \boldsymbol{x}_i^*\right\rangle \left\langle \beta, \boldsymbol{u}^{(j)}\right\rangle\right)$$

$$- \tilde{O}\left(\frac{\varepsilon}{r\sqrt{r}}\right) - \tilde{O}\left(\frac{\varepsilon^2}{r^2}\right)$$

holds. Now $\langle \beta, \boldsymbol{x}_i^*\rangle = \tilde{O}_p(1)$ and $\langle \beta, \boldsymbol{u}^{(j)}\rangle \leq 1$. Therefore, if $\left\langle \boldsymbol{u}^{(j)}, \boldsymbol{x}_i^*\right\rangle = \tilde{O}_p(1)$ holds, then $\left\langle \boldsymbol{x}_i^*, \boldsymbol{u}^{(j+1)}\right\rangle = \left\langle \boldsymbol{x}_i^*, \boldsymbol{u}^{(j)}\right\rangle + \tilde{O}\left(\frac{\varepsilon}{r\sqrt{r}}\right)$. Therefore, by induction, we have

$$\left\langle \boldsymbol{x}_i^*, \boldsymbol{u}^{(N_3+1)}\right\rangle = \tilde{O}\left(\log\frac{1}{\varepsilon}\right),$$

which does not depend on $d$. □

This lemma allows us to use Lemma 9. Note that $N_3 = \tilde{\Theta}(\frac{r\sqrt{r}}{\varepsilon}\log\frac{1}{\varepsilon})$ means, by ignoring the term $\log\frac{1}{\varepsilon}$, $\varepsilon = \tilde{\Theta}(\frac{r\sqrt{r}}{N_3})$). Now we can estimate $\langle\beta, \boldsymbol{x}\rangle$ using $\langle\boldsymbol{u}^{(N_3+1)}, \boldsymbol{x}\rangle$. For notational simplicity, we write $\boldsymbol{u}^{(N_3+1)}$ as $\boldsymbol{u}$ from now on.

**Lemma 20.** *Let $\boldsymbol{x} \sim \mathcal{N}(0, \boldsymbol{I}_d)$, and $\boldsymbol{u}$ be a vector independent of $\boldsymbol{x}$ satisfying $\|\boldsymbol{u}\| = 1$ and $\langle\beta, \boldsymbol{u}\rangle \geq 1 - \varepsilon$. Then, we have*

$$\langle\beta, \boldsymbol{x}\rangle = \langle\boldsymbol{u}, \boldsymbol{x}\rangle + O_p(\sqrt{2\varepsilon\log d}).$$

*Proof.* First, note that

$$\langle\beta, \boldsymbol{x}\rangle = \langle\boldsymbol{u}, \boldsymbol{x}\rangle + \langle\beta - \boldsymbol{u}, \boldsymbol{x}\rangle.$$

Since we have $\langle\beta, \boldsymbol{u}\rangle \geq 1 - \varepsilon$, we have that

$$\|\beta - \boldsymbol{u}\|^2 = \|\beta\|^2 + \|\boldsymbol{u}\|^2 - 2\langle\beta, \boldsymbol{u}\rangle \leq 2\varepsilon,$$

which yields

$$\|\beta - \boldsymbol{u}\| \leq \sqrt{2\varepsilon}.$$

Then, combining with Lemma 25 yields $\langle\beta - \boldsymbol{u}, \boldsymbol{x}\rangle = O_p(\sqrt{2\varepsilon\log d})$. $\qquad\square$

# F  TRAINING MLP LAYER

In this section, we show that the MLP layer can fit the polynomial $\sigma_*^{\text{test}}$. Most of the argument in this section is taken from Nishikawa et al. (2025), but we do not omit the proof for readability.

**Lemma 21.** *Suppose that there exists $g(\boldsymbol{x})$ such that*

$$|g(\boldsymbol{x}) - \langle\beta, \boldsymbol{x}\rangle| \leq \delta.$$

*Then, there exists $\pi(v, b)$ such that*

$$\left|\mathbb{E}_{v\sim\mathrm{Unif}\{\pm 1\}, b\sim[-\log^2 d, \log^2 d]}[\pi(v, b)\sigma(v \cdot g(\boldsymbol{x}) + b)] - \sigma_*(\langle\beta, \boldsymbol{x}\rangle)\right| = O(\delta(\log d)^{2\deg(\sigma_*)-2})$$

*with high probability over $\boldsymbol{x} \sim \mathcal{N}(0, \boldsymbol{I}_d)$. Moreover, $\sup_{v,b}|\pi(v, b)| = \tilde{O}(1)$ holds.*

*Proof.* Let $\sigma_*(z) = \sum_{k=0}^{\deg(\sigma_*)} s_k z^k$. Now from Lemma 9 in Damian et al. (2022), there exists $\pi'_k(v, b)$ such that $\sup_{v,b}|\pi'_k(v, b)| = O(1)$ and

$$\mathbb{E}_{v\sim\mathrm{Unif}\{\pm 1\}, b\sim[-1,1]}[\pi'_k(v, b)\sigma(vz + b)] = z^k$$

for any $|z| \leq 1$. If we define

$$\pi(v, b) = \sum_{k=0}^{\deg(\sigma_*)} s_k \frac{\pi'_k(v, b\log^{-2} d)}{\log^2 d}\log^{2k} d,$$

then, we have $\sup_{v,b}|\pi(v, b)| = O(\log^{2\deg(\sigma_*)-2} d)$. Moreover, when $|z\log^{-2} d| \leq 1$, we have

$$\mathbb{E}_{v\sim\mathrm{Unif}\{\pm 1\}, b\sim[-\log^2 d, \log^2 d]}[\pi(v, b)\sigma(vz + b)]$$

$$= \sum_{k=0}^{\deg(\sigma_*)} s_k \mathbb{E}_{v\sim\mathrm{Unif}\{\pm 1\}, b\sim[-\log^2 d, \log^2 d]}\left[\frac{\pi'_k(v, b\log^{-2} d)}{\log^2 d}\log^{2k} d\sigma(vz + b)\right]$$

$$= \sum_{k=0}^{\deg(\sigma_*)} s_k \log^{2k-2} d\mathbb{E}_{v\sim\mathrm{Unif}\{\pm 1\}, b\sim[-1,1]}\left[\pi'_k(v, b)\sigma(\log^2 d(vz\log^{-2} d + b))\right]$$

$$= \sum_{k=0}^{\deg(\sigma_*)} s_k z^k = \sigma_*(z).$$

Therefore, it holds that

$$\mathbb{E}_{v \sim \text{Unif}\{\pm 1\}, b \sim [-\log^2 d, \log^2 d]}[\pi(v,b)\sigma(v \cdot g(\boldsymbol{x}) + b)]$$

$$= \mathbb{E}_{v \sim \text{Unif}\{\pm 1\}, b \sim [-\log^2 d, \log^2 d]}[\pi(v,b)\sigma(v \langle \beta, \boldsymbol{x} \rangle + v\delta + b)]$$

$$= \mathbb{E}_{v \sim \text{Unif}\{\pm 1\}, b \sim [-\log^2 d, \log^2 d]}[\pi(v,b)\sigma(v \langle \beta, \boldsymbol{x} \rangle + b)] + O(\delta \log^{2\deg(\sigma_*)-2} d)$$

$$= \sigma_*(\langle \beta, \boldsymbol{x} \rangle) + O(\delta \log^{2\deg(\sigma_*)-2} d).$$

Here, we used the fact that $|\langle \beta, \boldsymbol{x} \rangle| \leq \log^2 d$ with high probability.

$\square$

**Lemma 22.** *Under the condition of Lemma 21, there exists $\boldsymbol{a}' \in \mathbb{R}^m$ such that*

$$\left| \sum_{j=1}^m a_j' \sigma(v_j \cdot g(\boldsymbol{x}) + b_j) - \sigma_*(\langle \beta, \boldsymbol{x} \rangle) \right| = \tilde{O}(m^{-\frac{1}{2}}) + O(\delta \log^{2\deg(\sigma_*)-2} d)$$

*holds with high probability over $\boldsymbol{x} \sim (0, \boldsymbol{I}_d)$. Moreover, $\|\boldsymbol{a}'\|^2 = \tilde{O}(m^{-1/2})$ holds with high probability.*

*Proof.* Using $\pi(v,b)$ in Lemma 21, define $a_j' = m^{-1}\pi(v_j, b_j)$. Since $\sup_{v,b}|\pi(v,b)| = \tilde{O}(1)$, from Hoeffdng's inequality, it holds that

$$\left| m^{-1} \sum_{j=1}^m \pi(v_j, b_j)\sigma(v_j \cdot g(\boldsymbol{x}) + b_j) - \mathbb{E}_{v,b}[\pi(v,b)\sigma(v \cdot g(\boldsymbol{x}) + b)] \right| = \tilde{O}(m^{-1/2}),$$

with high probability. Hence we have

$$\left| \sum_{j=1}^m a_j' \sigma(v_j \cdot g(\boldsymbol{x}) + b_j) - \sigma_*(\langle \beta, \boldsymbol{x} \rangle) \right| = \tilde{O}(m^{-\frac{1}{2}}) + O(\delta \log^{2\deg(\sigma_*)-2} d).$$

As for the upper bound of the norm $\|\boldsymbol{a}'\|^2 = m^{-1} \cdot m^{-1} \sum_{j=1}^m \pi(v_j, b_j)^2$, Hoeffding's inequality yields

$$\left| m^{-1} \sum_{j=1}^m \pi(v_j, b_j)^2 - \mathbb{E}_{v,b}[\pi(v,b)^2] \right| = \tilde{O}(m^{-1/2}).$$

Since $\sup_{v,b}|\pi(v,b)| = \tilde{O}(1)$, we can say that $\mathbb{E}_{v,b}[\pi(v,b)^2] = \tilde{O}(1)$, which completes the proof.

$\square$

**Lemma 23.** *Let $\boldsymbol{a}^*$ be the parameter trained via Algorithm 1. Then there exists $\lambda_3$ such that*

$$\frac{1}{N_4} \sum_{t=N_1+N_2+N_3+1}^{N_1+N_2+N_3+N_4} |y_t - f_{\text{TF}}(\boldsymbol{x}_t, \boldsymbol{u}, \boldsymbol{v}^*, \boldsymbol{a}^*, \boldsymbol{b}^*)| = \tau + \tilde{O}(m^{-1/2}) + O(\delta \log^{2\deg(\sigma_*)-2} d)$$

*with high probability. Moreover, $\|\boldsymbol{a}^*\|^2 \leq \tilde{O}_p(m^{-1/2})$ is satisfied.*

*Proof.* Let $M = N_1 + N_2 + N_3$ and $\boldsymbol{a}'$ be the output parameter constructed in Lemma 22: from the equivalence between $\ell_2$-regularized and norm-constrained optimization algorithms, if we carefully choose $\lambda_3$, then we have

$$\left( \frac{1}{N_4} \sum_{t=M+1}^{M+N_4} |y_t - f_{\text{TF}}(\boldsymbol{x}_t, \boldsymbol{u}, \boldsymbol{v}^*, \boldsymbol{a}^*, \boldsymbol{b}^*)| \right)^2 \leq \frac{1}{N_4} \sum_{t=M+1}^{M+N_4} (y_t - f_{\text{TF}}(\boldsymbol{x}_t, \boldsymbol{u}, \boldsymbol{v}^*, \boldsymbol{a}^*, \boldsymbol{b}^*))^2$$

$$\leq \frac{1}{N_4} \sum_{t=M+1}^{M+N_4} (y_t - f_{\text{TF}}(\boldsymbol{x}_t, \boldsymbol{u}, \boldsymbol{v}^*, \boldsymbol{a}', \boldsymbol{b}^*))^2$$

$$\leq (\tau + \tilde{O}(m^{-1/2}) + O(\delta \log^{2\deg(\sigma_*)-2} d))^2$$

(recall that we assumed $\tau = \Theta(1)$) with high probability, which yields the first assertion. Moreover, from Lemma 22, we can see that

$$\|\boldsymbol{a}^*\|^2 \le \|\boldsymbol{a}'\|^2 \le \tilde{O}_p(m^{-1/2}),$$

which completes the proof. □

**Lemma 24.** *We fix $\boldsymbol{b}$ and $\boldsymbol{v}$. Let $\mathcal{F}_A = \{\boldsymbol{a} \mapsto \sum_{i=1}^m a_j \sigma(v_j \langle \boldsymbol{x}, \boldsymbol{u} \rangle + b_j) \mid \|\boldsymbol{a}\| \le A\}$ the set of transformers where the norm of the MLP parameter $\boldsymbol{a}$ is upper bounded. When $\|\boldsymbol{b}\| \le B$ and $v_j = \pm 1$, then it holds that*

$$\mathrm{Rad}_N(\mathcal{F}_A) = \tilde{O}\left( \frac{A(B + \sqrt{m})}{\sqrt{N_4}} \right).$$

*Proof.* Note that $\mathbb{E}[g(\boldsymbol{x})^2] = \mathbb{E}[\langle \boldsymbol{u}, \boldsymbol{x} \rangle^2] = O(\log d)$, considering $\|\boldsymbol{u}\| = 1$. Then, following the same argument as Lemma 25 in Nishikawa et al. (2025) yields the assertion. □

# G  PROOF OF THE THEOREM 1

Now, we are ready to give the proof of Theorem 1.

*Proof.* First note that

$$\mathcal{R}_{f_{\mathrm{TF}}}(\boldsymbol{u}, \boldsymbol{v}^*, \boldsymbol{a}^*, \boldsymbol{b}^*) - \tau = \frac{1}{N_4} \sum_{i=N_1+N_2+N_3+1}^{N_1+N_2+N_3+N_4} |y_i - f_{\mathrm{TF}}(\boldsymbol{x}_i, \boldsymbol{u}, \boldsymbol{v}^*, \boldsymbol{a}^*, \boldsymbol{b}^*)| - \tau$$

$$+ \mathcal{R}_{f_{\mathrm{TF}}}(\boldsymbol{u}, \boldsymbol{v}^*, \boldsymbol{a}^*, \boldsymbol{b}^*) - \frac{1}{N_4} \sum_{i=N_1+N_2+N_3+1}^{N_1+N_2+N_3+N_4} |y_i - f_{\mathrm{TF}}(\boldsymbol{x}_i, \boldsymbol{u}, \boldsymbol{v}^*, \boldsymbol{a}^*, \boldsymbol{b}^*)|.$$

From Lemma 23, the first two terms $\frac{1}{N_4} \sum_{i=N_1+N_2+N_3+1}^{N_1+N_2+N_3+N_4} |y_i - f_{\mathrm{TF}}(\boldsymbol{x}_i, \boldsymbol{u}, \boldsymbol{v}^*, \boldsymbol{a}^*, \boldsymbol{b}^*)| - \tau$ are bounded by $\tilde{O}(m^{-1/2}) + O(\delta \log^{2 \deg(\sigma_*)-2} d)$. Moreover, from Lemma 22 and the definition of $\boldsymbol{b}^*$, we have $\|\boldsymbol{a}^*\| = \tilde{O}(m^{-1/2})$ and $\|\boldsymbol{b}^*\| = \tilde{O}(\sqrt{m})$. Therefore, from Lemma 24, we have

$$\mathrm{Rad}_N(\mathcal{F}_A) = \tilde{O}(N_4^{-1/2}).$$

Using the standard technique yields(see Appendix D.3 in Oko et al. (2024))

$$|\mathcal{R}_{f_{\mathrm{TF}}}(\boldsymbol{u}, \boldsymbol{v}^*, \boldsymbol{a}^*, \boldsymbol{b}^*) - \tau| = \tilde{O}(N_4^{-1/2}) + \tilde{O}(m^{-1/2}) + O(\delta \log^{2 \deg(\sigma_*)-2} d),$$

with probability at least 0.995. Note that all the desired events occur with high probability, except for this event. Therefore, when $d$ is large enough, all the events occur with probability at least 0.99.

Finally, we rewrite the term $O(\delta \log^{2 \deg(\sigma_*)-2} d)$. From Lemma 20, $\delta = O(\sqrt{2\varepsilon \log d})$ holds. Also, $N_3 = \tilde{\Theta}(\frac{r\sqrt{r}}{\varepsilon} \log \frac{1}{\varepsilon})$, which means, when we ignore the term $\log \frac{1}{\varepsilon}$, $\varepsilon = \tilde{\Theta}(\frac{r\sqrt{r}}{N_3})$. Therefore

$$O(\delta \log^{2 \deg(\sigma_*)-2} d) = \tilde{O}\left( \sqrt{\frac{r\sqrt{r}}{N_3}} \right)$$

is satisfied. This completes the proof. □

# H  ADDITIONAL LEMMAS

**Lemma 25.** *Suppose $\boldsymbol{x} \sim \mathcal{N}(0, \boldsymbol{I}_d)$ and $\boldsymbol{y}$ is a vector satisfying $\|\boldsymbol{y}\| = C$ independent of $\boldsymbol{x}$. Then*

$$\langle \boldsymbol{x}, \boldsymbol{y} \rangle = O(C\sqrt{\log d})$$

*holds with high probability.*

**Lemma 26.** *Let $z_1, \ldots, z_n$ be i.i.d. random variables which satisfy $\|z_i\| \leq C$ with high probability and $\mathbb{E}[z_1^2] = O(d^\alpha)$, where $\alpha$ is a constant independent of $d$. When $n = \text{poly}(d)$,*

$$\frac{1}{n} \sum_{i=1}^{n} z_i - \mathbb{E}[z_i] = O\left(C\sqrt{\frac{1}{n}}\right)$$

*holds with high probability.*

*Proof.* Let $z_i' = z_i \mathbf{1}_{|z_i| \leq C}$. Applying the uniform bound argument, we may consider that $z_i = z_i'$ for all $i = 1, \ldots, n$ with high probability because our assumption $n = \text{poly}(d)$ yields

$$P(\max_i |z_i| \leq C) \geq 1 - O(nd^{-C_*}) \geq 1 - O(d^{-C_*'}),$$

where $C_*'$ is a constant determined appropriately. Then, using Hoeffding's inequality yields

$$\frac{1}{n} \sum_{i=1}^{n} z_i - \mathbb{E}[z_i'] = O\left(C\sqrt{\frac{1}{n}}\right).$$

We complete the proof by showing that $|\mathbb{E}[z_i] - \mathbb{E}[z_i']|$ is sufficiently small. This can be shown by

$$
\begin{aligned}
\mathbb{E}[z_i] - \mathbb{E}[z_i'] =& \mathbb{E}[z_i \mathbf{1}_{|z_i| > C_z}] \\
\leq& \mathbb{E}[\mathbf{1}_{|z_i| > C_z}]^{1/2} \mathbb{E}[z_i]^{1/2} \\
\leq& O(d^{\alpha - C_*}),
\end{aligned}
$$

where $C_*$ can be taken sufficiently large from the definition of high probability event. Since we assumed that $n = \text{poly}(d)$, this term is smaller than the main term, which completes the proof. $\square$

**Lemma 27** (Damian et al. (2023), Property 1). *Let $\alpha, \beta \in \mathbb{S}^{d-1}$, then*

$$\mathbb{E}_{\boldsymbol{x} \sim \mathcal{N}(0, \boldsymbol{I}_d)}[\text{He}_i(\langle \alpha, \boldsymbol{x} \rangle) \text{He}_i(\langle \beta, \boldsymbol{x} \rangle)] = \mathbf{1}_{i=j} \, i! \, \langle \alpha, \beta \rangle^i.$$

## H.1 PRETRAINED MATRIX

Let $\Gamma^*$ be a matrix obtained after pretraining. Then, from Nishikawa et al. (2025), $\Gamma^*$ can be written as

$$\Gamma^* = \frac{r\mathbb{E}_\beta[\beta\beta^\top] + \boldsymbol{N}}{\kappa\sqrt{r}}$$

for some matrix $\boldsymbol{N}$ satisfying $\|\boldsymbol{N}\|_F = O(1/\sqrt{d})$ and a number $\kappa = \Theta(\log^{C_\kappa} d)$. Also, from Nishikawa et al. (2025) and the assumption on the support of $\beta$,

$$\boldsymbol{U}\beta \sim \text{Unif}\{(\alpha_1, \alpha_2, \ldots, \alpha_r, 0, \ldots, 0) \mid \alpha_1^2 + \cdots + \alpha_r^2 = 1\}.$$

holds for some orthogonal matrix $\boldsymbol{U}$. Then, using $\boldsymbol{D} = \text{diag}(\underbrace{1, \ldots, 1}_{r}, \underbrace{0, \ldots, 0}_{d-r})$, we have that $r\mathbb{E}_\beta[\beta\beta^\top] = \boldsymbol{U}^\top \boldsymbol{D}\boldsymbol{U}$.

**Lemma 28.** *Let $\boldsymbol{x} \sim \mathcal{N}(0, \boldsymbol{I}_d)$, then*

$$\|\sqrt{r}\kappa\Gamma^*\boldsymbol{x}\| = \tilde{O}(\sqrt{r})$$

*holds with high probability.*

*Proof.* By the argument above, we can write that

$$\sqrt{r}\kappa\Gamma^*\boldsymbol{x} = (\boldsymbol{U}^\top \boldsymbol{D}\boldsymbol{U} + \boldsymbol{N})\boldsymbol{x}.$$

From rotational invariance, when we define $\boldsymbol{y}$ as $\boldsymbol{y} = \boldsymbol{U}\boldsymbol{x}$, $\boldsymbol{y} \sim \mathcal{N}(0, \boldsymbol{I}_d)$ is satisfied. From the definition of $\boldsymbol{D}$, the first $r$ components of $\boldsymbol{D}\boldsymbol{y}$ follow standard normal distribution i.i.d., and the other $(n - r)$ components are equal to zero. Since applying $U^\top$ to a vector does not change the norm of the vector, we obtain

$$\boldsymbol{U}^\top \boldsymbol{D}\boldsymbol{U}\boldsymbol{x} = \boldsymbol{U}^\top \boldsymbol{D}\boldsymbol{y} = \tilde{O}(\sqrt{r}),$$

with high probability. Finally, since $\|\boldsymbol{N}\|_F = O(1/\sqrt{d})$, $\boldsymbol{N}\boldsymbol{x} = \tilde{O}(1)$ holds. Hence we can ignore the term $\boldsymbol{N}\boldsymbol{x}$. $\square$

**Lemma 29.** *Let $x \sim \mathcal{N}(0, I_d)$, and $u$ be a vector independent of $x$ satisfying $\|u\| = 1$. then*

$$\langle \sqrt{r}\kappa\Gamma^* x, u \rangle = O(\sqrt{\log d})$$

*holds with high probability.*

*Proof.* As in the previous lemma, we know that

$$\sqrt{r}\kappa\Gamma^* x = (U^\top D U + N)x$$

holds. Again, from rotational invariance, when we define $y$ as $y = Ux$, $y \sim \mathcal{N}(0, I_d)$ is satisfied. From the definition of $D$, the first $r$ components of $Dy$ follow standard normal distribution i.i.d., and the other $(n - r)$ components are equal to zero. Also, when we define $u' = Uu$, we have $\|u'\| = 1$, which means $\sqrt{u_1'^2 + \cdots + u_r'^2} \leq 1$. Therefore, from Corollary 30 in Nishikawa et al. (2025), we have

$$\langle U^\top DU x, u \rangle = \langle Dy, u' \rangle = O_p(\sqrt{\log d}).$$

As for $Nx$, again from Corollary 30 in Nishikawa et al. (2025), we have

$$\langle Nx, u \rangle = O_p\left(\sqrt{\frac{\log d}{d}}\right).$$

This is smaller than the main term. $\qquad\square$

**Lemma 30.** *Let $x_1 \ldots x_n \sim \mathcal{N}(0, I_d)$, and $z_1, \ldots z_n$ are i.i.d. random variables which satisfy $|z_i| \leq C_z$ with high probability. When $n = \mathrm{poly}(d)$, it holds that*

$$\left\| \frac{1}{n} \sum_{i=1}^n (z_i \sqrt{r}\kappa\Gamma^* x_i) - \mathbb{E}[z_1 \sqrt{r}\kappa\Gamma^* x_1] \right\| = \tilde{O}\left(C_z \sqrt{\frac{r}{n}}\right),$$

*with high probability.*

*Proof.* We use the same proof strategy as Lemma 31 in Nishikawa et al. (2025). Let $z_i' = z_i \mathbf{1}_{|z_i| \leq C_z}$. First, we can confirm that

$$
\begin{aligned}
\mathbb{E}[z_1 \sqrt{r}\kappa\Gamma_* x_1] - \mathbb{E}[z_1' \sqrt{r}\kappa\Gamma_* x_1] &= \mathbb{E}[\mathbf{1}_{|z_i| > C_z} \sqrt{r}\kappa\Gamma_* x_1] \\
&\leq \mathbb{E}[\mathbf{1}_{|z_i| > C_z}^2]^{1/2} \mathbb{E}[\|\sqrt{r}\kappa\Gamma_* x_1\|^2]^{1/2} \\
&\leq O(d^{-C}).
\end{aligned}
$$

Where $C$ can be taken sufficiently large from the definition of high probability event. Because $n = \mathrm{poly}(d)$, by redefining $C$ if necessary, we can make this term smaller than $\tilde{O}(C_z \sqrt{\frac{r}{n}})$. As $\frac{1}{n} \sum_{i=1}^n (z_i \sqrt{r}\kappa\Gamma^* x_i) = \frac{1}{n} \sum_{i=1}^n (z_i' \sqrt{r}\kappa\Gamma^* x_i)$ with high probability, we continue to evaluate $\frac{1}{n} \sum_{i=1}^n (z_i' \sqrt{r}\kappa\Gamma_* x_i) - \mathbb{E}[z_1' \sqrt{r}\kappa\Gamma_* x_1]$. Let $y_i = Ux_i$. Then $y_i \sim \mathcal{N}(0, I_d)$ holds. $z_i' Dy_i$ is a sub-Gaussian vector, and $(Dy_i)_k = 0$ when $k > r$ from the definition of $D$. Then, applying a standard concentration bound for a sub-Gaussian vector to the $r$-dimensional vector $(Dy_i)_{1:r}$ yields

$$\frac{1}{n} \sum_{i=1}^n z_i' Dy_i - \mathbb{E}[z_1' Dy_1] = \tilde{O}\left(C_z \sqrt{\frac{r}{n}}\right).$$

As multiplying the orthogonal matrix $U^\top$ does not change the norm of the vector, we have

$$\frac{1}{n} \sum_{i=1}^n z_i' U^\top DU x_i - \mathbb{E}[z_1' U^\top DU x_1] = \tilde{O}\left(C_z \sqrt{\frac{r}{n}}\right).$$

Again from the standard concentration bound for a sub-Gaussian vector, we have

$$\frac{1}{n} \sum_{i=1}^n z_i' x_i - \mathbb{E}[z_1' x_1] = \tilde{O}\left(C_z \sqrt{\frac{d}{n}}\right),$$

with high probability. Since $\|\boldsymbol{N}\|_F = O(1/\sqrt{d})$, we have

$$\frac{1}{n}\sum_{i=1}^{n} z_i'\boldsymbol{N}\boldsymbol{z}_i - \mathbb{E}[z_1'\boldsymbol{N}\boldsymbol{x}_1] = \tilde{O}\left(C_z\sqrt{\frac{1}{n}}\right).$$

In summary, we arrive at

$$\frac{1}{n}\sum_{i=1}^{n}(z_i\sqrt{r}\kappa\Gamma^*\boldsymbol{x}_i) - \mathbb{E}[z_1\sqrt{r}\kappa\Gamma^*\boldsymbol{x}_1]$$

$$= \frac{1}{n}\sum_{i=1}^{n}(z_i(\boldsymbol{U}^\top\boldsymbol{D}\boldsymbol{U} + \boldsymbol{N})\boldsymbol{x}_i) - \mathbb{E}[z_1(\boldsymbol{U}^\top\boldsymbol{D}\boldsymbol{U} + \boldsymbol{N})\boldsymbol{x}_1] = \tilde{O}\left(C_z\sqrt{\frac{r}{n}}\right),$$

with high probability, which completes the proof. $\qquad\square$

**Lemma 31.** *We have that*

$$\sqrt{r}\kappa\Gamma^*\beta = \beta + O(1/\sqrt{d}).$$

*Proof.* First note that

$$\sqrt{r}\kappa\Gamma^*\beta = (\boldsymbol{U}^\top\boldsymbol{D}\boldsymbol{U} + \boldsymbol{N})\beta.$$

From the definition of $U$, we have $\boldsymbol{U}\beta = (\alpha_1, \alpha_2, \ldots, \alpha_r, 0, \ldots, 0)^\top$ for some $\alpha_1, \alpha_2, \ldots, \alpha_r \in \mathbb{R}$. Therefore, we obtain that

$$\boldsymbol{U}^\top\boldsymbol{D}\boldsymbol{U}\beta = \boldsymbol{U}^\top\boldsymbol{U}\beta = \beta.$$

Finally, by noticing $\|\boldsymbol{N}\|_F = O(1/\sqrt{d})$ and $\|\beta\| = 1$, we can see that $\boldsymbol{N}\beta = O(1/\sqrt{d})$ holds. $\quad\square$

**Lemma 32.** *We have that*

$$\sqrt{r}\kappa\Gamma^*(\sqrt{r}\kappa\Gamma^*\beta) = \beta + O(1/\sqrt{d}).$$

*Proof.* From lemma 31, we have $\sqrt{r}\kappa\Gamma^*\beta = \beta + \boldsymbol{d}$ where $\boldsymbol{d} = O(1/\sqrt{d})$. Therefore, it holds that

$$\sqrt{r}\kappa\Gamma^*(\sqrt{r}\kappa\Gamma^*\beta) = \sqrt{r}\kappa\Gamma^*(\beta + \boldsymbol{d}).$$

Again from Lemma 31, we see that $\sqrt{r}\kappa\Gamma^*\beta = \beta + O(1/\sqrt{d})$. Also we have $\sqrt{r}\kappa\Gamma^*\boldsymbol{d} = \boldsymbol{U}^\top\boldsymbol{D}\boldsymbol{U}\boldsymbol{d} + \boldsymbol{N}\boldsymbol{d}$, and since $\|\boldsymbol{U}^\top\boldsymbol{D}\boldsymbol{U}\|_2 = 1$ from the definition of $\boldsymbol{U}$ and $\boldsymbol{D}$, we have $\boldsymbol{U}^\top\boldsymbol{D}\boldsymbol{U}\boldsymbol{d} = O(1/\sqrt{d})$. Finally, since $\|\boldsymbol{N}\|_F = O(1/\sqrt{d})$, we have $\boldsymbol{N}\boldsymbol{d} = O(1/d)$. This indicates that $\sqrt{r}\kappa\Gamma^*\boldsymbol{d} = O(1/\sqrt{d})$. $\qquad\square$

# I   EXPERIMENTAL DETAILS

The GPT-2 model architecture we used in Section 4 originates from Garg et al. (2023). Given the $(N + 1)$-length prompt $\{(\boldsymbol{x}_i, y_i)\}_{i=1}^{N+1}$, we first construct the embedding as

$$\boldsymbol{E} = [\boldsymbol{x}_1, \tilde{\boldsymbol{y}}_1, \ldots, \boldsymbol{x}_{N+1}, \tilde{\boldsymbol{y}_{N+1}}] \in \mathbb{R}^{d\times(2N+2)},$$

where $\tilde{\boldsymbol{y}}_i = [y_i, 0, \ldots, 0]^\top$. Next, the read-in layer transforms this embedding into $\tilde{\boldsymbol{E}} \in \mathbb{R}^{D\times(2N+2)}$, where $D = 128$. This mapped embedding $\tilde{\boldsymbol{E}}$ goes through a 6-layer GPT-2 backbone with 4 attention heads, following the configuration by Garg et al. (2023). Finally, the output of GPT-2 backbone is transformed by the read-out layer into the vector $[z_1, z_2, \ldots, z_{2N+1}, z_{2N+2}]$. Here, $z_{2i-1}$ is the prediction of $y_i$ given the context $(\boldsymbol{x}_1, y_1, \ldots, \boldsymbol{x}_{i-1}, y_{i-1}, \boldsymbol{x}_i)$. We used Adam optimizer (Kingma & Ba, 2017) with a learning rate of 0.0001. To reduce the pretraining cost, we adopted the curriculum learning strategy, which is also used in Garg et al. (2023). The training started with the dimension $d = 4$, and the dimension was increased by two until it reached the target dimension.

For the model with TTT, we used the same base model as ICL evaluation. We introduced low-rank adaptation (LoRA) to the attention projection layers (c_attn and c_proj) and the feedforward layer (c_fc). The rank of LoRA was set to 4, and the parameters LoRA_alpha and LoRA_dropout were set to 8 and 0.1, respectively. The LoRA parameters were updated 10 times for each query.

The inference-time learning rate was 0.01 and 0.05 in the experiment in Figure 1a and Figure 1b, respectively. To prevent information leakage when evaluating the prediction of $y_i$, the model used only the preceding data $(\boldsymbol{x}_1, y_1, \ldots, \boldsymbol{x}_{k-1}, y_{k-1}, \boldsymbol{x}_k)$ to update the weight. The LoRA weights were reset before proceeding to the prediction of the next target, $y_{i+1}$.

For the additional experiment (shown in Figure 2), we reused the TTT results from the in-distribution setting (Figure 1a) for the $r = d = 16$ case. For the $r = 4, d = 16$ case, the inference-time learning rate was also set to 0.01, consistent with the $r = 16$ setting.

The test loss was averaged over 256 runs, with each run containing 256 independent queries.

