# OpenReview forum: "Test time training enhances in-context learning of nonlinear functions"
_ICLR.cc/2026/Conference — Submitted to ICLR 2026_

### Official Review · Reviewer_n2jk · 2025-10-23

**Soundness:** 3
**Presentation:** 2
**Contribution:** 2
**Rating:** 4
**Confidence:** 3

**Summary:**

The authors consider a theoretical analysis of single-layer transformers trained to do ICL of single-index models with test-time training.

**Strengths:**

The authors consider the analysis of an interesting problem: ICL under test-time training.
- The results found seem reasonable and I believe most of the authors' claims about their model are well-supported.
- To my knowledge, the work is original and considers new topics
- While I have some reservations (below), the results in the paper are interesting and I believe that this is an important line for continuing research. In particular, the ideas the authors raise about distribution shift of the ICL mapping function are relevant to LLM phenomena.

**Weaknesses:**

I have a few reservations about the paper:
- The presentation is difficult to follow, and builds on a technical body of work. It would be useful if the authors could provide more intuition for their results
- It is not clear to me how to make an adequate comparison between theory and experiment from the information presented, and what comparison I can make leads me to suspect that theory & experiment do not agree.
    - No theoretical curves for the predicted test error are presented in Figure 1 in order to make comparison possible.
    - On line 430, the authors write "Contrary to our initial expectation, the pretrained model’s ICL ability is powerful enough to adapt to the varying link function, while TTT does not lead to significant improvement of the prediction." If I am understanding this correctly, the authors are saying that their theoretical results in fact do not agree with their experiments, and that they would not have expected from their theory that ICL alone could even learn the task.
    - On line 437, the authors write that "TTT can adapt to distribution shifts, while ICL fails to do so." From my understanding of Figure 1, this is not true. While TTT does outperform ICL on distribution shifts in the authors' experiments, ICL shows at least some ability to adapt to distribution shifts.
    - Taken together, these observations leave me with reservations about how well the authors' theoretical analysis reflects reality. If the authors could take steps towards unifying theory & experiment (even in the setting considered where the transformer learns single-index models), I would be happy to update my review.

**Questions:**

- What aspects of the analysis are leading to the prediction that ICL alone cannot learn a distribution of the link functions $\sigma$'s? How might the analysis be updated?
- It is unclear to me what aspects of the main theorem hold implications for real models trained on real data. Could the authors please clarify this?
- Could the authors please clarify what is meant by the statement that ICL fails to adapt to distribution shifts on line 437?

---

> ### Author Response · Authors · 2025-11-19
>
> Thank you for your constructive feedback.  We address your concerns as follows.
>
> **1.The numerical experiment**
>
> Based on your comments, we updated the experiment section.  Here are the key features:
>
> ・Reconciling the theory and the experiment: You correctly pointed out that our experiments showed ICL adapting to varying link functions, contrary to our theoretical prediction for single-layer models. This discrepancy stems from the architectural difference.  Our theory analyzes a single-layer transformer consisting of an attention layer followed by an MLP layer, where the MLP layer must capture the nonlinearity of the link function.  Since parameters are fixed during test-time in the ICL-only setting, a single-layer ICL model cannot adapt to varying link function \sigma.  However, our experiment utilized a 6-layer GPT-2 model, which has MLP layer followed by an attention layer.  As shown in recent work (Oko et al. 2024), such a combination allows the model to dynamically fit varying link functions.  We suppose that this is the reason why ICL actually learned the task.   Furthermore, regarding the absence of theoretical curves in Figure 1, please note that our theoretical result gives an upper bound on the predictive loss, not the exact amount of it. Since the bound is an inequality rather than an equality, plotting it directly on the experimental results would not yield a tight fit or a valid visual comparison.
>
> ・Clarification on “ICL fails to adapt”: We agree that the statement "ICL fails to adapt" was too strong given the empirical results in Figure 1.  Therefore, we have rephrased this to: “This highlights an important benefit of TTT: its superior ability to adapt to distribution shifts compared to ICL.”  We acknowledge that ICL adapts to the distribution shifts to some extent, but it is significantly outperformed by TTT in out-of-distribution scenarios.
>
> ・Importantly, Figure 1 also suggests that TTT does not degrade performance even in regimes where ICL is already effective. This indicates that TTT is a robust addition that improves resilience to shifts without compromising in-distribution performance.
>
> **2.Implications for real models**
>
> Regarding your question on real-world implications: Our convergence rate assures that a transformer can learn a single-index model with enough data.  This implies that for real-world high-dimensional data, transformers via TTT can efficiently learn hidden important direction and fit nonlinearity.  Moreover, our sample complexity does not depend on d, suggesting the transformer can adapt to hidden low-dimensionality.
>
>
> We are happy to address any concerns and questions you have during the discussion period.

---

> ### Author Response · Authors · 2025-11-29
>
> We would like to confirm if our response has resolved your concerns.  The discussion between the authors and the reviewers may become impossible soon, but if you have any remaining concerns, we are happy to have a “last minute discussion” before we will not be able to have a discussion.

---

### Official Review · Reviewer_b1ir · 2025-10-31

**Soundness:** 4
**Presentation:** 3
**Contribution:** 3
**Rating:** 6
**Confidence:** 2

**Summary:**

This work provides a theoretical understanding for why transformers using in-context learning (ICL) can further improve their performance approximating non-linear functions under distribution shift using test time training (TTT). TTT has proven to be an effective approach empirically, but a strong understanding of several properties of this approach, such as its sample complexity, were unknown. The paper compares with previous work, and demonstrates that their new results provide convergence rates in more relevant variables, such as the dimensionality of the task-related subspace and number of training examples. Furthermore, the paper’s guarantees also allow for the non-linearity in the family of single-index models that they analyze to change at test time, providing a more interesting and relevant setting for TTT. Finally, the paper provides results for synthetic experiments demonstrating that TTT specifically allows for adaptation to a changing link-function, something that ICL cannot handle on its own.

**Strengths:**

- The techniques being analyzed (ICL and TTT) are very relevant right now to empiricists and theorists alike.
- The proof sketch provided makes a great job at not only explaining the reasoning in the proof but also demonstrating how pretraining with ICL and TTT are both necessary for this setting.
- Extensive comparisons with previous literature are provided, setting the stage clearly.
- Synthetic experiments towards the end of the paper provide satisfying evidence that TTT specifically helps with adapting to changes in the non-linear aspect of the single index model.
- The setting provided (link-function changing between test and train) seems more relevant to TTT than when the link-function is static, making the theoretical results more impactful.

**Weaknesses:**

- [W1] Some things in Algorithm 1 are not clear. See [Q1], [Q2].
- [W2] It is not always clear what the contributions of this work are, specifically in comparison to Nishikawa et al. (2025). See [Q3], [Q4].
- [W3] In real applications of TTT, examples are still provided in-context after TTT (Akyurek et al. (2025)). This seems to demonstrate that using information in context is still important for the performance of transformers using TTT. However, the final predictive function $f_{TF}$ does not use in-context examples, somewhat separating this work from real applications.
- [W4] (Minor) Some typos (line 127, “polylogamistic” -> polylogarithmic?).

**Questions:**

- [Q1] What is $N_{new}$? Based on line 142/143, I was under the impression that the only sizes of TTT data we had access to were $N_1, N_2, N_3, N_4$.
- [Q2] Where are the projected data in algorithm lines 7 and 16 used? It’s made clear that this operation is meant to put the data in the subspace spanned by the $\beta$s seen during pretraining, but they aren’t used in stage 2 or stage 3 of the algorithm.
- [Q3] In 3.1.1 and 3.1.4, it is stated that Nishikawa et al. proved the important properties of the pre-trained attention matrix, as well as how to recover the link-function with a ReLU MLP, respectively. Is the novelty in this approach in the weak and strong recovery steps? How does this new approach differ from how Nishikawa et al. recover $\beta$, if they do so?
- [Q4] On lines 333-339, Nishikawa et al.’s sample complexity bound is said to be tighter, but that the new bound still has two key benefits. Are these two benefits not also shared with the bound provided by Nishikawa et al.?

---

> ### Author Response · Authors · 2025-11-19
>
> Thank you for your constructive feedback.  We address your concerns as follows.
>
> **1. Clarification on Algorithm 1 (regarding [Q1] and [Q2])**
>
> We appreciate you pointing out the ambiguity. We have revised the paper to clarify these definitions.
>
> [Q1] Definition of N_{new}: $N_{new}$ is distinct from $N_1,N_2,N_3$ and $N_4$.  This represents the number of the auxiliary vectors $w_i  \overset{\text{i.i.d.}}{\sim} \mathcal {N}(0,\mathbf{I_d})$.  In TTT stage I, we do not use the label corresponding to the query $\mathbf{x}$.  Therefore, instead of using labeled data, we sample new random vectors $w_i$ to facilitate the training process.
>
> [Q2]Usage of Projected Data: The projected data is indeed essential for computing $f_{TF}$ and $g$.  To make this explicit, we have updated the notation to $f_{TF}(\Gamma, \mathbf{X}, \mathbf{y}, \mathbf{x})$, clearly indicating its dependency on the projected data generated in Lines 7 and 16.
>
> **2.The comparison with Nishikawa et al. (regarding [Q3] and [Q4])**
>
> We make the following remarks.
>
> [Q3]: Yes, our novelty basically lies in the weak and strong recovery steps.  While Nishikawa et al. recover $\beta$ via ICL, their estimation lacks accuracy, resulting in the absence of a convergence rate with respect to the dimension $d$. Furthermore, they do not prove risk convergence with respect to $N_{test}$. In contrast, our TTT approach allows for a more accurate prediction of $\beta$, yielding an explicit convergence rate with respect to $N_{test}$.
> In addition, while Nishikawa et al. train the MLP layer during pretraining, we train it at test-time, which allows our model to dynamically adapt to the task-specific link function—a key advantage of our approach.  The way of training the MLP layer is the same, but the timing is different.
>
> [Q4] Clarification on Bounds: We agree that the tightness of the bound is a shared trait. We have updated the paper to strictly separate "shared properties" from our "unique contributions" (nonlinear TTT analysis, adaptivity and convergence guarantees) to avoid overclaiming.
>
> **3. Usage of In-Context Examples (Response to [W3])**
> We acknowledge that not using context after TTT differs from practical applications (e.g., Akurek et al., 2024). However, this is a deliberate architectural choice for our theoretical setting.  As discussed in Remark 4, relying on softmax attention to extract information on $\beta$ in this specific setting introduces prediction errors. Therefore, for learning single-index models theoretically, avoiding the continued use of softmax attention yields better guarantees. Our results highlight the limitations of standard ICL in this specific regime and demonstrate how TTT compensates for them.
>
> We are happy to address any concerns and questions you have during the discussion period.

---

> ### Author Response · Authors · 2025-11-29
>
> We would like to confirm if our response has resolved your concerns.  The discussion between the authors and the reviewers may become impossible soon, but if you have any remaining concerns, we are happy to have a “last minute discussion” before we will not be able to have a discussion.

---

### Official Review · Reviewer_Kk7X · 2025-11-02

**Soundness:** 3
**Presentation:** 3
**Contribution:** 2
**Rating:** 4
**Confidence:** 3

**Summary:**

The paper discusses the efficiency of Test Time Training (TTT) in contrast with In-Context Learning (ICL) as a means of adapting to the prompt/context of an LLM.

The paper establishes bounds on the expected risk in terms of the number of samples provided in the context.

The improvement of TTT over ICL is demonstrated in a synthetic setting.

**Strengths:**

The difference between ICL and TTT shown in the experiment, and that it requires a sufficiently hard problem (in this case, a distribution shift) is interesting.

**Weaknesses:**

There are several strong modeling assumptions in the paper.
1. The “Regression for ICL” model
2. The single index model.
3. Pretraining consists of only one step of GD.
4. The context samples are used in a specific way (first to learn the direction of $\beta$ weakly, and then to learn $\beta$ properly).
5. The “input” weights of the MLP are frozen (and random), only the output weight is trained.


Arguably, points 1 and 2 are now common enough in the literature to be considered benign. There are several works that consider the type of GD that is considered here (point 3 and 5), an early example I remember is on feature learning in 2 layer ReLUs, I consider this to be a significant deviation from practice but refrain from ultimate judgement. Point 4 is novel to this setting afaik. I am not sure if the context is actually split in this way to learn some ground truth in stages. Can this be elaborated upon?

**Questions:**

Are $N_{\text{new}}$ and $N_2$ the same? Is $w_i$ the $i$th sample in the $\mathbf{X}_{N_2}$ set?

What is $g$?

What is the point of lines 15-16 in the algorithm? It doesnt seem like those x are used again. Also arent all the covariates projected onto $\Gamma$? Might be easier to just say that in line 6-7 if so.

For random feature based learning I would expect the size of the hidden layer to be quite large. Here we have m = poly(d). What does $d$ need to be explicitly? I couldnt tell from Appendix G (may be this is made in explicit in some of the related work).

---

> ### Author Response · Authors · 2025-11-19
>
> Thank you for your constructive feedback.  We address your concerns as follows.
>
> **1. Clarifications on Algorithm 1 and Notation**
>
> We apologize for the confusion regarding the notation.
>
> ・$N_{new}$ and $\mathbf{w_i}$: $N_{new}$ is distinct from $N_2$. Specifically, $\mathbf{w_i}$ are not test-time data points ($\mathbf{x_i}$) but are auxiliary vectors sampled from $\mathcal {N}(0,\mathbf{I_d})$ specifically in our algorithm, and $N_{new}$ is the number of sampled vectors $\mathbf{w_i}$. Since we do not need access to the ground truth label for the query $\mathbf{x}$ during the TTT stage I, we sample these new vectors $\mathbf{w_i}$ to serve as queries.
>
> ・The definition of g: Thank you for pointing this out.  We have explicitly defined g in the revision.  It denotes the attention layer output of the original model $g(\Gamma_*,\mathbf{X_{N1}},\mathbf{y_{N_1}},\mathbf{x}) = \frac{N_1^{-1}\sum_{i=1}^{N_1}y_i\mathrm{e}^{y_i/\rho}\mathrm{e}^{\mathbf{x_i}^\top\Gamma_*\mathbf{x}}}{N_1^{-1}\sum_{i=1}^{N_1}\mathrm{e}^{y_i/\rho}\mathrm{e}^{\mathbf{x_i}^\top\Gamma_*\mathbf{x}}}.$
>
> ・The meaning of the projection:  The projected data is indeed used for the model predictions $f_{IC}$ and $g$. We have updated the notation to $f_{IC}(\Gamma, \mathbf{X}, \mathbf{y}, \mathbf{x})$ to show explicitly that the context data affects the output. Note that projections are applied to specific subsets ($\mathbf{X_1}, \mathbf{X_2}$), while $\mathbf{X_3}$ and $\mathbf{X_4}$ remain unprojected.
>
> **2. The size of hidden layer**
>
> As for your last question, we do not require $d$ to grow to control $m$.  We treat the input dimension $d$ as a fixed, sufficiently large constant, and the width $m$ scales relative to the desired error rate, rather than $d$ directly.  The condition $m=poly(d)$ assures that the estimation error with respect to $d$ is small enough to be controlled.
>
> **3.Context Splitting (Modeling Assumption 4)**
>
> The splitting of the context into distinct groups is a deliberate theoretical choice to ensure statistical independence between updates. By using fresh samples for weak recovery (Stage I), strong recovery (Stage II), and MLP training (Stage III), we avoid complex correlations that would arise from reusing data, allowing us to derive rigorous convergence bounds. While practical implementations might pool data, this splitting is crucial for the theoretical validity of the proof.  On the other hand, we also believe that the sample splitting could be omitted by carefully handling a uniform bound with sharp concentration inequalities. However, we chose a simpler theoretical analysis instead of unnecessarily messing up the proof. We would like to defer this point to an important future work.
>
> We are happy to address any concerns and questions you have during the discussion period.

---

> ### Author Response · Authors · 2025-11-29
>
> We would like to confirm if our response has resolved your concerns.  The discussion between the authors and the reviewers may become impossible soon, but if you have any remaining concerns, we are happy to have a “last minute discussion” before we will not be able to have a discussion.

---

### Official Review · Reviewer_ke57 · 2025-11-02

**Soundness:** 2
**Presentation:** 1
**Contribution:** 2
**Rating:** 2
**Confidence:** 4

**Summary:**

This paper proposes an algorithm for test-time scaling and validates its effectiveness through theoretical analysis and experiments. The authors derive an upper bound that depends on the feature subspace dimension, the test-time context length, and the network width, suggesting that Transformers can adapt to low-dimensional subspaces. The paper also includes numerical experiments on synthetic data to support the theoretical findings.

**Strengths:**

The underlying problem the authors investigate is important, given the growing power and influence of LLMs. Understanding the mechanism behind test-time training, which has shown strong empirical performance, is both timely and valuable. Also, the paper makes a clear effort to exposit the structure of the proof, helping readers follow the theoretical reasoning and connect the analytical results to the underlying intuition.

**Weaknesses:**

The paper has issues in terms of writing quality and overall presentation, placing it below the acceptance threshold.

Firstly, the authors fail to clearly compare their theoretical results with prior work, making it hard to understand what the exact novelty and contribution of this paper are. While it is true that the paper provides theoretical results, one key questions remain:

1. The order of the bounds in the theoretical results appears standard and not particularly impressive. For a theoretical paper, the authors should explicitly highlight how their bounds improve upon existing ones, what technical challenges were involved, and how these were addressed. It seems that the authors assume the reader have read the paper Nishikawa et al. (2025), and therefore a lot of comparison are missing. Even after reading 327-352, I am still not sure what are the difference of the setups, assumptions, and technical improvements.

Moreover, there are many small issues in terms of writing, which really impacts the readability

1. line 074 $N_{pt}$ and $T_{pt}$ are not defined, and the author put this in the statement of a theorem.

2. section 2.3 is introduced without any transition. It is about a student model, but the author never mentioned why we need a student model for TTT. At least a smooth transition for the motivation is required. For instance, if I replace "student model" with "one-layer transformer", then the paper would still read fine.

3. it is mentioned earlier that the motivation is for developing the theory is that when $r < d$ the sample complexity is much better. But in numerical experiment, the authors set $r = d = 16$.

In general, when reading the paper, I am quite confused about the setup and motivation.

**Questions:**

See weakness above

---

> ### Author Response · Authors · 2025-11-19
>
> Thank you for your constructive feedback.  We address your concerns as follows.
>
> **1.The comparison with Nishikawa et al.(2025)**
>
> We acknowledge that the comparison with prior work needed more explicit detailing. We clarify the main difference of our work compared to Nishikawa et al. (2025) from three key perspectives:
>
> ・Problem Setting (Generalization): The task of learning a nonlinear single-index model is the same, except for the link function: While Nishikawa et al. assume a fixed link function across all tasks, our setting is more challenging and realistic as we allow task-specific link functions. This generalization requires the model to adapt dynamically, which is difficult under their fixed framework.
>
> ・Methodology (TTT vs. Static): The core algorithmic difference lies in the use of Test-Time Training (TTT). Nishikawa et al. rely solely on pretraining for parameter updates (updating parameters only during the pretraining phase). In contrast, we explicitly modify parameters using test data.  In particular, we train the MLP layer at test time, rather than pretraining it.   This allows our model to dynamically adapt to the aforementioned task-specific variations, which is a key advantage of our approach.
>
> ・Theoretical Contribution (Convergence): Regarding your concern about the bounds: The novelty lies in the establishment of the convergence guarantee itself, rather than the order.  The recovery precision of $\beta$ by Nishikawa et.al. is insufficient, offering no convergence assurance with respect to $N$. By leveraging TTT via weak and strong recovery steps, we achieve precise predictions of $\beta$, enabling us to provide an explicit convergence rate with respect to $N$ that did not exist before. This is a qualitative leap, proving that TTT is essential for convergence in this generalized setting.
>
>
> **2. Readability issues**
>
> We apologize for the lack of clarity and the definitions. We have revised the paper to address your specific points:
>
> ・Definitions of $N_{pt}$ and $T_{pt}$: We have updated the statement of the theorem (informal) and Section 2.1 to explicitly state that $N_{pt}$ denotes the number of context samples and $T_{pt}$ denotes the number of distinct datasets used during pretraining.
>
> ・Motivation for the "Student Model" (Section 2.3): We added an explicit transition explaining why a concrete student architecture is required for the theoretical analysis.
>
> ・Experimental Setup ($r=d$ vs. $r < d$): We acknowledge your valid point regarding the inconsistency between our theoretical motivation ($r \ll d$) and the initial experimental setting ($r=d=16$). Our original intent was to isolate the algorithmic comparison between TTT and standard ICL. However, we agree that verifying the sample efficiency in the low-rank regime is essential. Following your suggestion, we conducted a new experiment comparing $r=4$ (low-rank) and $r=16$ (full-rank), while fixing $d=16$.  The result, which we added to the revised manuscript, shows that the model in the low-rank setting ($r=4$, purple line) converges significantly faster and achieves a lower final predictive error than the full-rank setting ($r=16$, orange line).  This new result provides direct empirical evidence that our TTT framework successfully adapts to the intrinsic dimensionality ($r$) of the task, which was precisely the theoretical motivation for our analysis.  We thank the reviewer for this constructive feedback, which has led to a much stronger empirical validation of our work.
>
> We are happy to address any concerns and questions you have during the discussion period.

---

> ### Author Response · Authors · 2025-11-29
>
> We would like to confirm if our response has resolved your concerns.  The discussion between the authors and the reviewers may become impossible soon, but if you have any remaining concerns, we are happy to have a “last minute discussion” before we will not be able to have a discussion.

---

### Author Response · Authors · 2025-12-01
**Summary of Revisions and Response to Reviewer Concerns**

Dear new Area Chair,

We understand that you have been assigned to this submission recently. Based on the constructive feedback from the reviewers, we have significantly revised the manuscript to improve clarity and added new experimental results. Here, we summarize how we have addressed the major concerns raised in the reviews:

**1. Clarifications on Algorithm 1**

We have resolved the ambiguities regarding the notation and definitions in Algorithm 1.

・**Projection and Notation**: We updated the notation to $f_{IC}(\Gamma, \mathbf{X}_N, \mathbf{y}_N, \mathbf{x})$ and $g(\Gamma, \mathbf{X}_N, \mathbf{y}_N, \mathbf{x})$ to explicitly show that the model predictions depend on the projected vectors $\mathbf{x}_i$.

・**Definition of $w_i$ and $N_{new}$**: In the explanation of the algorithm(Lines 306-311), we clarified that $N_{new}$ is distinct from $N_1, \dots, N_4$. It represents the number of auxiliary vectors $\mathbf{w}_i$ sampled from $\mathcal{N}(0, \mathbf{I}_d)$ used during the TTT stage, which are distinct from the test inputs $\mathbf{x}_i$.

・**Definition of $g$**: We have included the explicit mathematical definition of $g$ in the main text (Lines 308-309).

**2.The numerical experiments**

・**Low-rank validation**: To address the concern regarding the setting of $r=d$, we conducted a new experiment comparing a low-rank setting ($r=4$) with the full-rank setting ($r=16$). The results confirm that our model converges significantly faster in the low-rank regime, empirically validating our theoretical motivation that TTT adapts to the hidden low dimensionality.

・**Theory vs. Experiment Discrepancy**: Reviewers noted that ICL performed better in experiments than our single-layer theory predicted. We clarified that this is due to the architectural difference between the theoretical model (single-layer) and the experimental model (6-layer GPT-2). We revised the text to explain that while deep ICL shows some adaptability to distribution shifts, TTT significantly outperforms it in out-of-distribution settings. Furthermore, we emphasized that TTT maintains competitive performance even in in-distribution settings where ICL is effective.

**3.Comparison with prior work**

We have explicitly clarified the key differences between our work and Nishikawa et al. (2025) as follows:

・**Problem Setting**: We tackle a generalized setting with task-specific link functions, whereas they assume a fixed link function.

・**Methodology**: Our approach uses Test-Time Training (TTT) to update the MLP layer dynamically, while they rely solely on pretraining for parameter update.

・**Theoretical Guarantee**: We provide an explicit convergence rate with respect to $N$ by leveraging weak and strong recovery steps via TTT, which was not achieved in their work.

・**Clarification on Sample Complexity**: The reviewers pointed out that the near-optimal sample complexity is already achieved by Nishikawa et al. (2025). However, we would like to emphasize that the underlying nature of these bounds are essentially different. That is, Nishikawa et al.'s result only guarantees that the loss is $o_d(1)$, meaning their bound holds only in the asymptotic limit where $d \to \infty$. In contrast, we prove not only convergence of loss but also its rate with respect to $N_{test}$, ensuring our results hold for large but *fixed* $d$. We have revised the "Main Result" section (Page 7) to clearly distinguish our unique contributions from this shared property to avoid overclaiming.

We believe these revisions have resolved the reviewers' concerns and strengthened the paper's contribution.

Unfortunately, the reviewers have not responded to our rebuttals, likely due to the recent system disruptions, and their scores may not reflect these clarifications. We respectfully ask the new Area Chair to consider these points and evaluate our work based on the revised manuscript and our responses to the misunderstandings in the initial reviews.

---

### Meta-Review · Area_Chair_ySWW · 2025-12-30

**Summary:**

The authors provide a theoretical analysis of the generalization error of a transformer with a nonlinear MLP and softmax attention layer, in the setting of test-time training (TTT): first, pretraining occurs on diverse tasks, then at test time, additional training occurs before making predictions.   They consider families of single-index models sigma_*(<beta,x>) and characterize the number of pretraining tasks and test-time tasks, and samples per task, that are needed to guarantee small excess risk.

The reviewers were in agreement that the problem is important, but their consensus was leaning towards rejection.  A main concern, which I share, is with the splitting of the context into four distinct groups and then using different stages of training: Kk7x and b1ir both had issues with this.  There was also the fact that the final predictor f_{TF} ignores the original in-context examples which feels unusual.  A number of reviewers also raised concerns about the improvement over Nishikawa et al.  Although many of the techniques from this paper are clearly derived from that work, the setting (TTT) is qualitatively different than the one considered there (pretraining for ICL only), but I agree that the technical novelty from this paper is more limited given that work.

In its current form, I don't think the paper has strengths that would outweigh the lack of support among the reviewers to recommend acceptance.  I think the main improvement that could be done would be to work towards an analysis of a TTT algorithm that is more clearly aligned with how TTT is actually implemented, with only one stage ideally (perhaps two if absolutely necessary, but three feels over the top).

**Reviewer Concerns:**

I believe the comparison with Nishikawa et al was, largely, addressed.  The settings are different; multiple link functions can be covered; and there are non-asymptotic results here.

The experimental questions (r=4 and r=16 LORA experiments, the usage of 6 layers in experiments etc) were addressed

However, the reviewers' concerns regarding the architecture/algorithms were not addressed (and in no way could be addressed during a rebuttal).  I think the authors need to come up with a way to include the in-context examples in the final prediction, and to use fewer than three stages of TTT, in order for this work to be accepted.

**Reviewer Scores:**

ke57 - I think the reviewer may have changed it to a 4 but I'm not sure.  I don't think the reviewer's severe negative score was merited from their review's text.

b1ir - probably maintained at 6, perhaps upgraded to an 8

Kk7x - equally likely to have maintained at 4 or raised to a 6; they didn't have very significant concerns, but weren't enthusiastic

n2jk - slightly more likely to have raised to a 6 than maintained at 4; their concerns were primarily regarding experiments

---

### Decision · Program_Chairs · 2026-01-26

Reject